# Land-use change undermines the stability of avian functional diversity

Thomas L. Weeks[1,2], Patrick A. Walkden[1,2], David P. Edwards[3], Alexander C. Lees[4], Alexander L. Pigot[5], Andy Purvis[1,2] & Joseph A. Tobias[1✉]

Land-use change causes widespread shifts in the composition and functional diversity of species assemblages. However, its impact on ecosystem resilience remains uncertain. The stability of ecosystem functioning may increase after land-use change because the most sensitive species are removed, which leaves more resilient survivors[1–3]. Alternatively, ecosystems may be destabilized if land-use change reduces functional redundancy, which accentuates the ecological impacts of further species loss[4,5]. Current evidence is inconclusive, partly because trait data have not been available to quantify functional stability at sufficient scale. Here we use morphological measurements of 3,696 bird species to estimate shifts in functional redundancy after recent anthropogenic land-use change at 1,281 sites worldwide. We then use extinction simulations to assess the sensitivity of these altered assemblages to future species loss. Although the proportion of disturbance-tolerant species increases after land-use change, we show that this does not increase stability because functional redundancy is reduced. This decline in redundancy destabilizes ecosystem function because relatively few additional extinctions lead to accelerated losses of functional diversity, particularly in trophic groups that deliver important ecological services such as seed dispersal and insect predation. Our analyses indicate that land-use change may have major undetected impacts on the resilience of key ecological functions, hindering the capacity of natural ecosystems to absorb further reductions in functionality caused by ongoing perturbations.

Anthropogenic land-use change is the primary driver of biodiversity decline and turnover[6]. At a global scale, natural and semi-natural habitats are undergoing complex and accelerating changes, including agricultural expansion, industrial development and urbanization[7,8]. These landscape transformations are a defining feature of the Anthropocene and have led to substantial shifts in the composition of species assemblages[9,18,19]. However, the impacts of these compositional changes on ecosystem function are difficult to measure or predict[13].

A standard approach to inferring changes to ecosystem function involves estimating the diversity of functional traits in species assemblages. This is based on strong evidence that species traits provide information about functional roles[14–17]. A growing number of studies have also shown a correlation between trait diversity and ecological processes (Supplementary Information), which provides support for the widespread use of functional diversity (FD) metrics to assess the impacts of land-use change on ecosystem function[9,18,19]. Such analyses often conclude that land-use change has relatively minor effects on FD after accounting for species turnover[20,21] and that high levels of functionality are therefore retained in human-modified landscapes[22,23]. Nonetheless, most studies focus on overall trait diversity of assemblages, an approach with two main limitations. First, FD estimated for whole assemblages does not provide information about the integrity of particular ecological functions, some of which are less resilient than others (Fig. 1). Second, standard FD metrics reflect a snapshot in time and do not tell us anything about the stability of ecosystem function in the face of further environmental perturbation[5], which suggests that the long-term impacts of land-use change may be underestimated.

Functional redundancy—and its flipside, functional uniqueness—are dimensions of FD that focus on the supply of species to deliver each function. Redundancy metrics achieve this parameter by estimating the number of co-occurring species with overlapping functionality[24]. If multiple species in an assemblage provide similar functions, surplus species are functionally redundant[25]. In ecological terms, functional redundancy is a positive attribute[26] because surplus species increase resilience and stability, which facilitates the continuity of ecological processes when conditions change[4,5,27,28]. This 'insurance effect' is widely reported in empirical studies[29–31], suggesting that functional redundancy is a core feature of resilient ecosystems. In general, assemblages with many surplus species that perform similar roles will have increased levels of functional resistance—one of the two major components of overall resilience[32] (Fig. 1)—therefore ensuring that ecological functionality is maintained when species are lost from the assemblage[33–35].

Under random species loss, functional redundancy is equivalent to functional resistance. However, the effects of land-use change

[1]Department of Life Sciences, Imperial College London, Silwood Park Campus, Ascot, UK. [2]Department of Life Sciences, Natural History Museum London, London, UK. [3]Department of Plant Sciences, University of Cambridge, Cambridge, UK. [4]Department of Natural Sciences, Manchester Metropolitan University, Manchester, UK. [5]Centre for Biodiversity and Environment Research, Department of Genetics, Evolution and Environment, University College London, London, UK. ✉e-mail: j.tobias@imperial.ac.uk

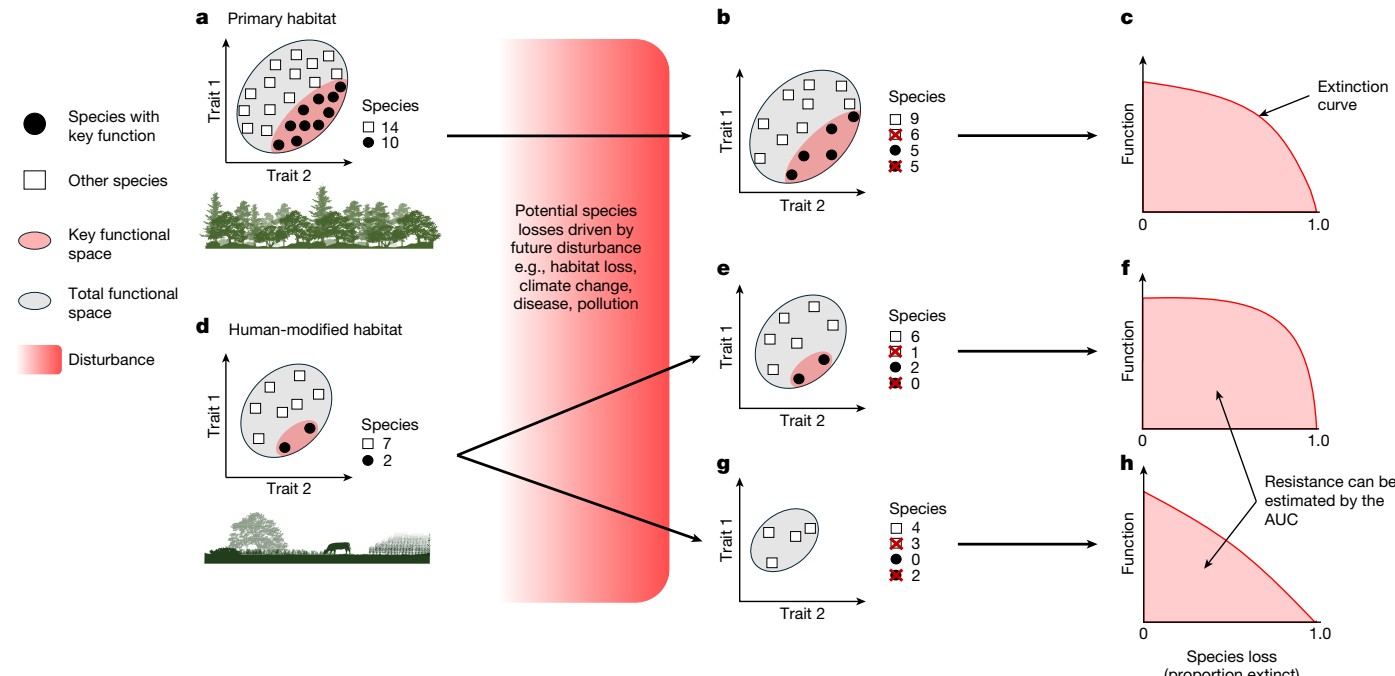

**Fig. 1 | Impacts of environmental disturbance on the functional stability of species assemblages. a–h,** Conceptual diagram illustrating how species loss may influence FD and resilience. Resilience (the ability to withstand disturbance) is a function of resistance (the amount of change after a disturbance) and recovery (the ability to return to equilibrium after a disturbance). Note that functional resistance—the potential for assemblages to absorb extinctions without declines in FD—is boosted by redundancy. That is, declines in FD are slower when there are more surplus species with similar function. **a,** FD of an intact species assemblage visualized as the volume of functional trait space occupied by all species present, with the red subset showing FD of a key function (for example, pollination). **b,** FD of the same assemblage after various potential disturbances, illustrated by a colour gradient from weak to strong red, reflecting the intensification of anthropogenic pressures on the environment.

These hypothetical future disturbances will remove sensitive species from the assemblage (X indicates extinction). Note that standard FD assessments may overlook changes to stability because surplus species can be lost with minimal effect on the total area of occupied trait space. **c,** Functional resistance is more clearly expressed by an extinction curve that describes FD loss until species richness declines to zero. **d,** Human-modified habitats may have lower FD with uncertain effects on functional resistance. **e,f,** Hypothetically, if most sensitive species of pollinators (red subset) have already been filtered from the assemblage (**e**), functional resistance may increase if surviving species are more tolerant to disturbance, thereby slowing FD loss (**f**). **g,h,** Alternatively, functional resistance may be undermined if functionally unique species are not disproportionately tolerant (**g**), thereby leading to rapid losses of key functions (**h**). AUC, area under the extinction curve.

are non-random because species with particular combinations of traits are more extinction-prone and tend to be filtered from the new environment[36,37]. Moreover, these sensitive species are distributed non-randomly, often clustering in distinct functional groups. These groups may undergo increased rates of local extinction, which in turn lead to increased risks of ecological collapse and vulnerability of ecological processes[38–40]. Indeed, if land-use change drives non-random species gains in some tolerant functional groups in parallel with species losses in more sensitive groups, the functional stability of an assemblage can be impaired despite no overall loss of functional redundancy. This may occur, for instance, when ecological specialists are replaced by disturbance-tolerant or generalist species in anthropogenic habitats[2,3]. We are left with a key conundrum: whether anthropogenic land-use change leads to new species assemblages that are more resilient to future shocks (because sensitive species are already lost and resilient species increase in abundance) or to assemblages that become more fragile and sensitive to further collapse.

To examine this question, we quantified the impacts of land-use change on functional trait diversity and redundancy of bird assemblages (Extended Data Fig. 1a–d). As redundancy and resistance can be decoupled if highly sensitive species provide unique functions in the assemblage, we also quantified the vulnerability of each assemblage to functional losses. Previous trait-based analyses have made progress in identifying which response traits predict species sensitivity to land-use change[36,37], with an emphasis on the first element of standard response–effect frameworks[41]. In this study, we shifted the focus onto the second element—effect traits—to estimate the impacts of species loss on the

functioning of future ecosystems[42]. Birds provide an ideal opportunity to quantify the functional effects of environmental change with high resolution because they have been intensively surveyed. Moreover, comprehensive trait data with well-established links to key ecological and trophic processes[16,43] are now available for all bird species[44].

In total, we examined 3,696 bird species in 1,281 focal assemblages worldwide, sampled across land-use gradients from primary vegetation to urban habitats (Fig. 2a and Supplementary Table 1). For each species, we compiled 8 morphometric traits from AVONET[44], representing averages calculated from a mean of 11 individuals per species, then used a two-step principal component analysis (PCA) to account for collinearity among traits (Methods and Extended Data Fig. 2). We estimated the FD of each assemblage as functional richness, which was defined as the total volume of the occupied trait space in a probabilistic hypervolume generated from the PCA axes and dietary information for all species in the assemblage[45] (Fig. 1 and Methods). We first compared 177 assemblages in primary vegetation—including 152 (86%) in forests and 25 (14%) in non-forest vegetation (mainly grasslands and shrublands)—with 1,104 assemblages in nearby human-modified landscapes. Anthropogenic land-uses drove the removal of species with larger body size, lower dispersal ability and narrower geographical, climatic and dietary niches (Extended Data Fig. 3), a result that is in line with previous studies[36,37,46]. In tandem, low-intensity human activity drove minor but significant increases in FD, as detected in disturbed primary vegetation ($\hat{\beta} = 0.150$, $P = 0.006$). Mature secondary forests recovered similar levels of FD to intact primary vegetation ($\hat{\beta} = -0.35$, $P = 0.144$). However, substantial reductions in FD were consistently observed

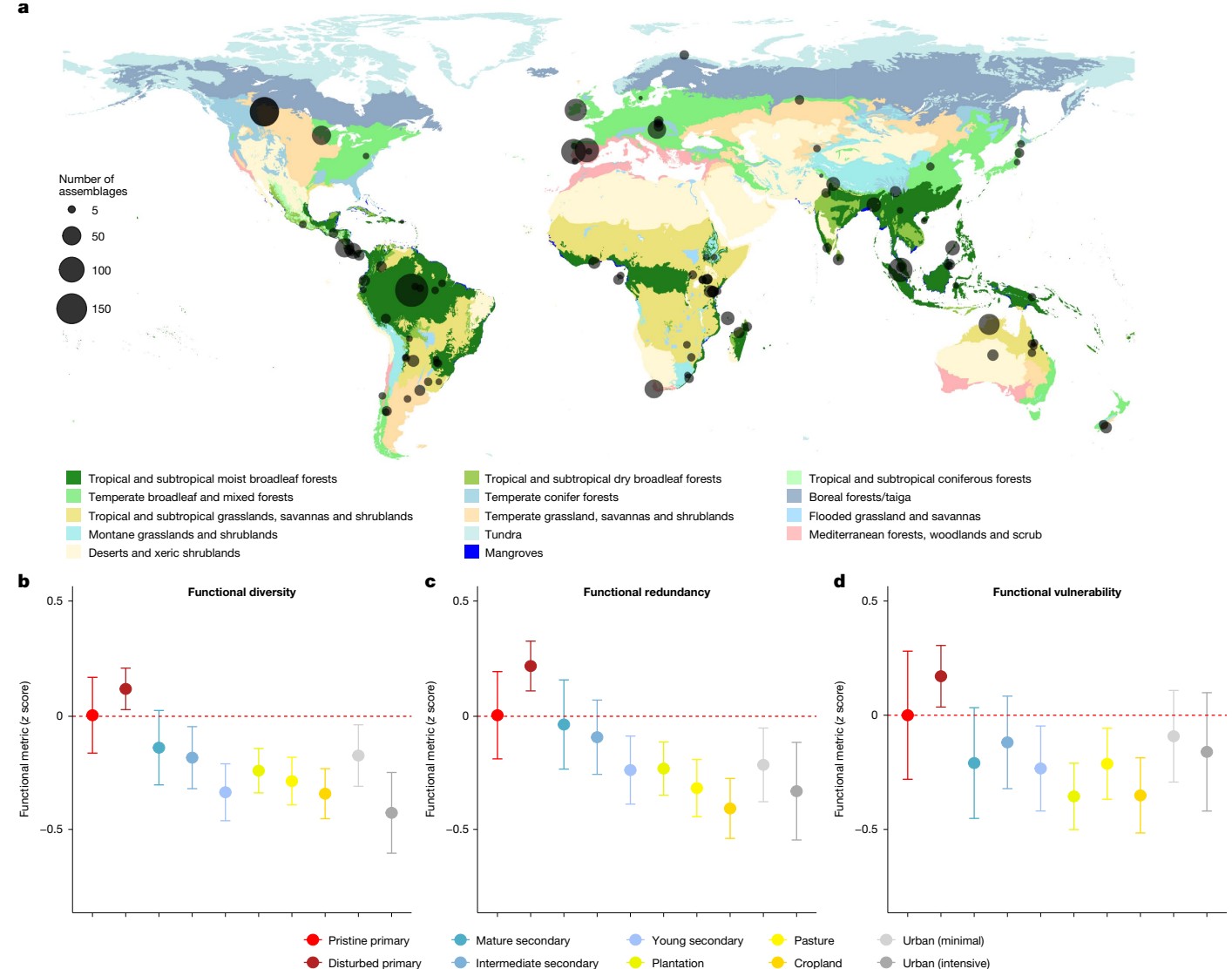

**Fig. 2 | Sampling and impacts of land-use change on avian assemblages.**
**a**, Circles show the geographical location of 98 field surveys that provided data
for 1,281 avian assemblages. The circle size is proportional to the number of
assemblages surveyed in each study landscape. Colours indicate major biomes[69]
projected onto a world map. **b**–**d**, Outputs from univariate mixed-effects
models assessing the impact of land-use change on three assemblage-level
metrics calculated from all assemblages ($n$ = 1,281): FD (measured as functional
richness) (**b**); functional redundancy (**c**); and functional vulnerability (**d**).
All metrics are compared with a pristine primary vegetation baseline
(including forests, grasslands, shrublands and wetlands; dashed red line).

Functional vulnerability was calculated using Spearman's rank correlation
coefficient between species-level redundancy and trait-based sensitivity
scores (Methods). To aid comparison, response variables were converted to
$z$ scores by square-root transformation before analysis and then scaled by their
standard deviation. Points shown are $z$ score estimates for each functional
metric and error bars indicate 95% confidence intervals. Note that a negative
functional vulnerability implies greater resistance to further loss of FD. World
map outline in **a** adapted from Natural Earth (https://www.naturalearthdata.com)
and ArcGIS under a Creative Commons licence CC BY 4.0.

across other, more heavily disturbed land-use types (Fig. 2b), particu-
larly in highly urbanized landscapes ($\hat{\beta}$ = −0.463, $P$ < 0.001).

## Functional redundancy

To evaluate these patterns in FD from the perspective of ecosystem
resilience, we estimated functional redundancy for each assemblage
using the position of all species in the trait hypervolume. We also esti-
mated their intraspecific variation generated using a standard kernel
density estimator[47] (Methods). These values of intraspecific variation
provide an estimate of niche breadth in the context of trait space, which
enabled us to calculate functional redundancy as the average number
of species that could be removed from each cell of the hypervolume
without reducing the functional volume occupied by the assemblage

as a whole (Supplementary Information). The results revealed that in
addition to its effects on FD, land-use change altered assemblage-level
functional redundancy (Fig. 2c), with roughly matching patterns
reported from plant communities[48]. Specifically, trait redundancy
initially increased after the switch from pristine to disturbed primary
vegetation ($\hat{\beta}$ = 0.200, $P$ = 0.010), with minor, nonsignificant reduc-
tions in both mature ($\hat{\beta}$ = −0.087, $P$ = 0.394) and intermediate-age
($\hat{\beta}$ = −0.145, $P$ = 0.089) secondary vegetation. However, more intensive
land-use showed significant decreases in redundancy, with particularly
sharp declines in cropland ($\hat{\beta}$ = −0.506, $P$ < 0.001) and intensively
urbanized landscapes ($\hat{\beta}$ = −0.386, $P$ < 0.001). Results were similar
when we calculated intraspecific variation based on direct measure-
ments of multiple individuals per species (Methods and Extended Data
Fig. 4).

To disentangle the effects of land-use change on different ecological processes regulated by trophic interactions, we modelled changes in FD and functional redundancy within trophic guilds (Extended Data Fig. 5). In dietary generalists and granivores, FD and redundancy either remained constant or increased in agricultural and urban landscapes, which reflected an influx of open-country and urban-tolerant species, some with distinctive traits[12]. By contrast, FD and redundancy declined steeply in frugivores (which are involved in seed dispersal) and invertivores (with roles in controlling insect populations). For these analyses, we used a standard classification that defines trophic specialists as species that consume the relevant food type across most (>60%) of their diet[44]. However, trophic generalists may contribute to the same ecological functions; therefore we ran sensitivity analyses with broader trophic guilds (>25% of diet; Methods). The results were similar, which suggests that declines in FD and redundancy are much steeper in components of avian diversity that contribute to seed dispersal and insect predation (Extended Data Figs. 5 and 6). These findings indicate that whole-assemblage FD and redundancy should be treated with caution because they are averaged across multiple ecological processes with widely diverging sensitivity to land-use change. Specifically, when diversity increases in disturbance-tolerant guilds, this can obscure substantial declines in disturbance-sensitive guilds and mask the reduced capacity of anthropogenic assemblages to maintain important ecosystem functions.

## Functional vulnerability

Although functional redundancy patterns imply that land-use change can limit the capacity of ecosystems to withstand further species losses, the link between functional redundancy and stability is not clear-cut. Assemblages with low redundancy can be stable if the remaining species are well adapted to human-modified landscapes. Moreover, ecological functions can be unstable even in highly redundant assemblages if many species are densely packed into only a few functional groups to leave other areas of trait-space under-represented[49]. In such a scenario, the delivery of rarer functions can be unstable if the species responsible are disproportionately sensitive to land-use change or persist in small population sizes (Methods).

To examine the question of stability more closely, we devised two metrics of functional vulnerability that incorporated the amount of unique function provided by species, along with their probable sensitivity to anthropogenic pressures. Specifically, we calculated a species-level redundancy value based on the relative contribution of each species to total assemblage functional redundancy and a general sensitivity score estimating the vulnerability of each species to future threats. We estimated sensitivity based on general response traits (Supplementary Table 2) or population size (rarity) to represent the likelihood that each species would undergo local extinction in response to a broad range of disturbances (Methods and Supplementary Information). To generate functional vulnerability values, we calculated the covariance between the sensitivity scores for all species occurring in the assemblage and their functional redundancy. High functional vulnerability values indicate a negative covariance between sensitivity and redundancy, which implies that species with increased extinction risk also provide a large proportion of unique function.

Our global-scale models revealed that functional vulnerability is reduced in all anthropogenic land-use types whether the sensitivity of species to disturbance is estimated as a function of general response traits (trait-based functional vulnerability; Extended Data Fig. 5i–l) or abundance (rarity-based functional vulnerability; Extended Data Fig. 5m–p). However, the only significant declines were detected for trait-based functional vulnerability in young secondary vegetation and agricultural landscapes. These findings support the hypothesis that intensive land-use change removes the most sensitive species, which results in lower assemblage vulnerability because most of the species surviving in and colonizing anthropogenic landscapes tend to be less prone to extinction[2,50].

## Functional stability

By showing that functional redundancy and functional vulnerability both decline after land-use change, our analyses suggest that human impacts have opposing effects on ecosystem stability. In anthropogenic environments, ecological functions are delivered by a reduced set of species. However, these species have lower extinction risk because they are more tolerant of further anthropogenic pressures. To disentangle these effects, we calculated two functional-resistance values for each assemblage under realistic species-loss scenarios that targeted the most extinction-prone or rarest species. We simulated these scenarios by removing species in order of sensitivity (high to low) using the same trait-based and rarity-based sensitivity scores devised to calculate functional vulnerability values (Fig. 3a and Methods). This approach enabled us to track the rate at which FD declines when species are sequentially removed from the assemblage (Supplementary Fig. 1), thereby quantifying how land-use change may influence the functional stability of assemblages undergoing future stressors[33,34].

We did not find evidence that lower functional vulnerability in disturbed habitats acts as a buffer to initial losses of function. Instead, under both trait-based and rarity-based extinction, functional resistance for whole assemblages followed a similar pattern to functional redundancy. That is, substantial declines occurred in human-modified landscapes, particularly in agricultural and urban settings (Fig. 3b,c). The trait-based scores we used in simulated extinctions were generated from a basket of general response traits that reflect sensitivity to unknown future threats, which provides little direct insight into the likely interaction between land-use change and other explicit stressors, such as climate change[51]. Moreover, we actively removed the most vulnerable species at each time step until species richness dropped to zero, which may not reflect the longer-term persistence of disturbance-tolerant species in anthropogenic environments.

To evaluate the robustness of our results in the context of methods, we ran three sensitivity analyses (Methods). First, we selected a different set of response traits associated with species sensitivity to climate change. Second, we generated extinction curves with passive (probability-weighted) species loss, wherein tolerant species with lower sensitivity scores were allowed to remain in the assemblage for longer periods. Third, we re-ran our main analyses with an alternative functional resistance metric, the half-life ($t_{1/2}$) of each extinction curve, defined as the proportion of species that need to be removed for FD to decline by 50%[5] (Extended Data Fig. 1e and Supplementary Fig. 1). These analyses produced similar results (Extended Data Fig. 7), which suggests that functional stability consistently declines with increasing land-use intensity. In effect, any additional stability conferred by lower functional vulnerability in disturbed habitats seems to be counteracted by an absence of surplus species across the whole assemblage, leading resistance to decline because there is less redundancy per species (Extended Data Fig. 8). Our simulations also support the view that land-use change creates species assemblages with reduced resilience to synergistic threats, including climate change[51] (Extended Data Fig. 7a). Finally, removal of all spatially autocorrelated studies from our analyses did not change the results. This finding indicates that our conclusions are not influenced by spatial autocorrelation (Extended Data Fig. 9).

## Interpreting variation in resilience

Functional stability calculated at the assemblage level reflects the combined stability of different trophic groups with varying responses to land-use change[36,37] (Extended Data Fig. 5). Accordingly, functional

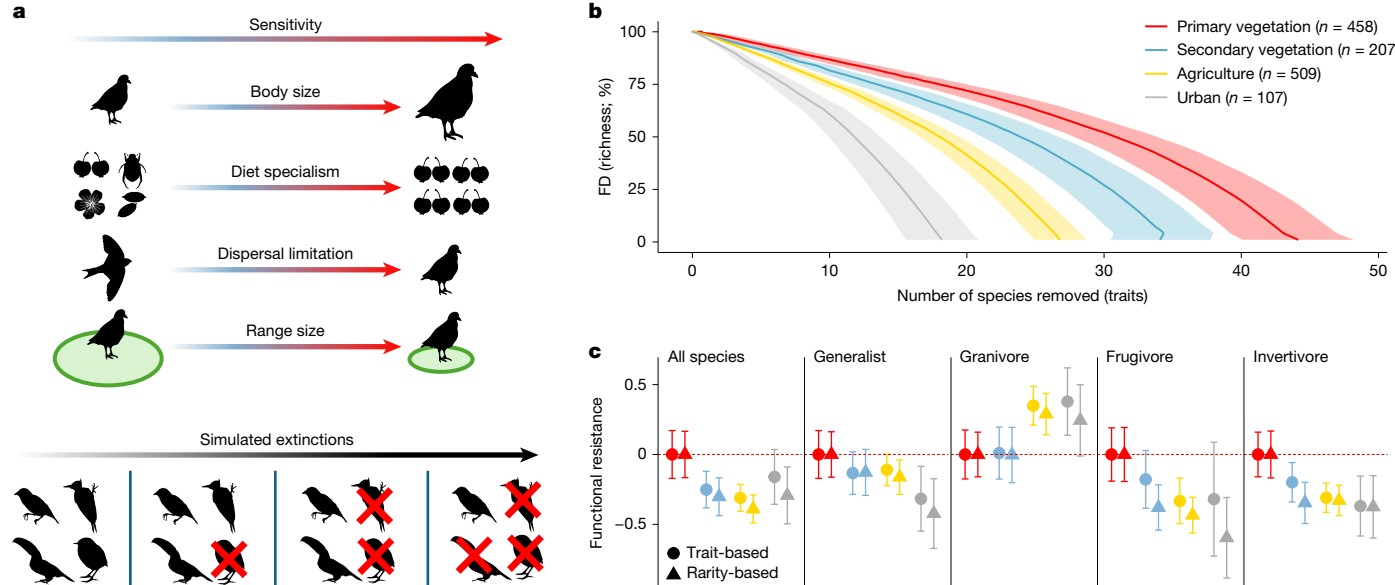

**Fig. 3 | Land-use change reduces functional stability. a**, Schematic of the procedure used to quantify functional stability of assemblages ($n = 1,281$). As a first step, all species in an assemblage were ranked by extinction risk based on four general response traits (Methods). Arrows are coloured to illustrate the gradient from low sensitivity (blue) to high sensitivity (red). Using these sensitivity gradients, simulated extinction curves were generated by removing species sequentially in order of their sensitivity score (high to low). The impact of species loss was then quantified by calculating functional trait diversity as a proportion of the starting FD before any species was removed, which provides an index of stability (functional resistance). **b**, Using this approach, we plotted average extinction curves for each land-use type predicted using a cubic smooth spline algorithm. The shaded region shows 95% confidence intervals. To aid visualization, we use the total number of species removed, whereas our functional vulnerability and resistance analyses use the proportion of species remaining in the assemblage (to avoid our results being driven by species richness). **c**, The impacts of land-use change on functional resistance in different trophic groups were visualized by calculating the predicted change

in functional resistance as the AUC (Supplementary Fig. 1). Dashed red lines indicate the standardized predicted functional resistance (set to 0) for pristine primary vegetation (forests, shrublands, grasslands and wetlands). Metrics were calculated and compared across five subsets: all species ($n = 1,281$ assemblages); trophic generalists ($n = 1,281$ assemblages); and the three key trophic guilds granivores ($n = 1,271$ assemblages), frugivores ($n = 944$ assemblages) and invertivores ($n = 1,274$ assemblages). Points shown are coefficient estimates from five separate linear mixed-effects models and error bars indicate 95% confidence intervals. Silhouettes are adapted from PhyloPic (https://www.phylopic.org) under a CC0 1.0 Universal Public Domain licence, unless otherwise stated. *Campephilus magellanicus* created by Edwin Price; *Chionis minor* created by Alexandre Vong; *Geranium maculatum* created by Mason McNair; *Malus pumila* created by T. Michael Keesey under a Public Domain Mark 1.0 licence; *Ploceidae* created by lucy_the_bob_man under a Public Domain Mark 1.0 licence; *Popillia japonica* created by Andy Wilson; *Riparia riparia* created by Bruno Maggia; *Xenicus gilviventris* created by Ferran Sayol; *Ramphastos* created by Edwin Price under a Creative Commons licence CC BY 4.0.

resistance in assemblages of dietary generalists and granivores tended to remain stable or even to increase across human-modified landscapes. Such changes reflected the proliferation of different food resources such as domestic waste, carrion and seed-bearing grasses in the borders of agriculture and human settlements. By contrast, functional resistance was highly unstable in more sensitive guilds, including frugivores and invertivores (Fig. 3c and Extended Data Figs. 5 and 6). Declines in ecosystem stability after land-use change are therefore unevenly distributed, with the largest losses concentrated in key trophic guilds that mediate ecological services, such as seed dispersal and pest control, which are vulnerable to rapid future collapse in human-modified landscapes. In particular, the substantial increase in functional vulnerability of frugivores in disturbed habitats (Extended Data Figs. 5 and 6) suggests that few surplus species survive in the most at-risk regions of functional space. This scenario implies that there would be accelerated declines in seed dispersal services under further species loss. The high sensitivity of key trophic guilds was found in all species-removal simulations (Extended Data Fig. 7), a result consistent with previous studies showing that frugivorous and insectivorous birds are susceptible to local extirpation in disturbed tropical forests[40,52]. Indeed, the response of tropical seed dispersers and insect predators to land-use change across all extinction scenarios drove a more general pattern of accentuated declines in FD and redundancy at lower latitudes (Extended Data Fig. 10), where many bird species are sensitive to habitat loss and fragmentation[46,53].

## Implications for land-use management

The widespread decline we observed in avian FD is consistent with numerous studies reporting similar patterns in response to agricultural expansion, land-use intensification and urbanization[9,12,54,55]. This outcome reflects the loss of species maladapted to highly modified environments, including ecological specialists that occupy unique regions of functional trait space[56]. It can be argued that the reduction in FD simply reflects reduced ecological demand for services provided by these functionally unique species in agricultural and urban assemblages. Nonetheless, a wide variety of trophic interactions are still required for human-modified ecosystems to function efficiently, and a diverse baseline of predators, pollinators and seed dispersers is needed to maintain the potential for ecosystem recovery and restoration[40,57] (Supplementary Information).

In addition to lower FD, we detected substantial reductions in trait redundancy and assemblage-level functional vulnerability. This finding indicates that human-modified assemblages are dominated by fewer, typically generalist species as landscapes become more intensively transformed[3]. The lower functional vulnerability of post-disturbance assemblages implies that they are more resilient, perhaps because extinction filters have removed the most sensitive species[1,50]. However, the results of simulated extinctions suggest the opposite (Fig. 3). Instead, as redundancy declines, the insurance effect provided by the rich diversity of undisturbed bird assemblages is eroded, which

accentuates the adverse impacts of further species losses[5,33]. In other words, the minor positive effects of land-use change on functional vulnerability are outweighed by reductions in trait redundancy, which potentially leaves ecosystems susceptible to much larger declines in functionality if further species are lost.

High functional vulnerability detected in pristine habitats reflects an increased number of disturbance-sensitive species with unique trait combinations. This finding highlights the role of intact ecosystems as safe harbours for rare, functionally distinct and extinction-prone species[58-60]. It may seem logical to conclude that ecosystem functionality is least stable in natural primary vegetation where so many disturbance-sensitive species are important to ecological function[61]. However, we showed that greater instability arises from widespread reductions in trait redundancy that is occurring throughout the entire assemblage in moderately to heavily disturbed environments, consistent with theoretical predictions and experimental evidence[29-31]. Undisturbed habitats support much higher levels of redundancy throughout the entire assemblage, thereby promoting functional stability. Notably, well-developed secondary vegetation and lightly disturbed habitats had similar levels of redundancy to those found in pristine primary vegetation. This result highlights the importance of retaining and restoring semi-natural and disturbed vegetation to boost the resilience of ecosystem functions[62].

## Caveats and limitations

Our analyses are subject to multiple limitations and unavoidable sampling biases. The use of space-for-time comparisons to estimate impacts of land-use change introduces uncertainty, not least because most primary habitats sampled have a long history of human disturbance. Widespread defaunation of ecosystems worldwide, including Pleistocene megafaunal extinctions, mean that even primary vegetation supports much-depleted levels of FD compared with a historical baseline[63,64]. Moreover, any dataset derived from bird surveys is prone to error because survey-detection probabilities vary across species and land-use types, with rates of detection increasing in open or disturbed habitats where birds are visible at longer range (Supplementary Information). Our dataset nonetheless provides a reasonable estimate of recent land-use-change impacts given that most bird species are identifiable and relatively detectable even in dense habitats because of their songs and other acoustic signals. Moreover, the main effect of shifting baselines and imperfect detection is to reduce estimates of species richness, abundance and redundancy in primary habitats (Supplementary Information). Therefore, improved detection rates would most probably accentuate our main results by boosting FD values in undisturbed landscapes and steepening the estimated decline in resilience after land-use change.

Another source of uncertainty lies in our simulation of future extinctions. Although it is not possible to know which species will drop out of a local assemblage and over what time frame, we used species traits and rarity to define the most likely sequence of extinctions. We also resampled many extinction sequences, under varying levels of extinction probability, to provide an estimate of uncertainty. Future studies should explore different ways of simulating extinctions, with refined estimates of species sensitivity and turnover. Finally, the ecological trait diversity of bird assemblages can only provide limited insight into the functioning of whole ecosystems. For example, our analyses did not consider variation in activity patterns and physiological rates nor the extent to which particular functions are replaceable by mammals, insects and other taxa. Further research is needed to integrate additional species traits across a wider set of taxonomic groups and to quantify the connection between FD and ecosystem resilience. Nonetheless, the abundance and trait diversity of birds provides a useful starting point for understanding the impacts of environmental change. That is, a global framework for estimating trophic processes and associated energy flows that can be strongly mediated by birds even at the ecosystem scale[40,65].

## Conclusions

By integrating species traits into biodiversity metrics, our analyses revealed that land-use change drives pervasive declines in functional resistance and stability of bird species assemblages. The impacts were most severe in heavily modified environments and concentrated in key ecological groups with prominent roles in seed dispersal (frugivores) and pest control (invertivores). Overall, anthropogenic landscapes support fewer surplus species in these regions of trait space, exposing them to future dysfunction if additional species are removed. The consistent pattern detected in birds confirms and extends the findings of local-scale studies showing reduced functional resistance in invertebrate assemblages[66,67]. An important implication of these findings is that standard approaches to estimating the effects of land-use change on ecosystem function may underestimate longer-term impacts. Specifically, they may suggest that species assemblages in human-modified landscapes are more resilient, whereas a detailed appraisal using trait hypervolumes reveals that they are actually more fragile and primed for further declines in functionality if biodiversity losses continue unchecked[68]. Conservation efforts should therefore focus on maintaining and restoring the functional resilience of species assemblages to reduce the risk of future ecological collapse.

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

## Methods

### Survey data

To assess impacts of land-use change on bird diversity, we began by collating surveys from the PREDICTS database, a repository of species occurrence and abundance data sampled across multiple land-use types[70]. We removed 30 datasets because they lacked abundance data ($n = 12$) or were incomplete ($n = 18$), usually because sampling was limited to particular guilds or methods, such as camera traps (Supplementary Information). This process produced a baseline of 72 datasets that we augmented by conducting a systematic literature review using Web of Science to identify further published bird surveys that targeted land-use gradients (Supplementary Information). After contacting authors for data, we received 29 suitable datasets that we added to the PREDICTS database. Our sample in this study contains data from these 29 surveys, along with 5 additional datasets released in the latest version of PREDICTS[71]. Finally, to improve sampling in megadiverse regions, we integrated further independent datasets generated by intensive surveys in Bornean[72] and Amazonian rainforests[73]. Geographical location and sources for all published surveys used in our analyses are presented in Fig. 2, Supplementary Table 1 and Supplementary Data 1.

Most survey data in the PREDICTS database are organized as a hierarchy. Survey sites are nested in study blocks, and blocks are nested in study landscapes. Study blocks are spatially segregated but not always temporally defined[70]. We found 16 datasets in PREDICTS that contain surveys sampled in different years or seasons; therefore, we subdivided the data into separate study blocks partitioned by location and time of survey. To ensure consistency, we also collapsed 17 surveys in PREDICTS into 7 studies by combining all data extracted from the same original source publications. We then partitioned these seven studies into distinct study blocks representing geographical and temporal subsets. After restructuring, our final dataset consisted of 98 study landscapes (Fig. 2), each with numerous survey sites clustered into study blocks. Subdividing our data in this way enabled us to account for spatial, annual and seasonal effects across studies by including study block as a random effect in our models (Supplementary Information).

We converted survey data into species assemblages to enable comparisons across land-use types. In some study landscapes, species assemblages reflect the total number of species identified in a study site, usually pooled across a series of transects or point-counts conducted at intervals between dawn and midday. Other published studies focused at finer resolution, sometimes defining each point-count as a separate survey site. Single point-counts generally undersample species richness, and neighbouring sites may be very close together, which caused problems for our analyses. In these cases, to facilitate calculation of functional metrics and to minimize the risk of pseudo-replication, we grouped species into larger assemblages by aggregating survey sites with similar land-use types in the same study block (Supplementary Information). Species assemblages were therefore defined as all species encountered in a restricted, spatially and temporally segregated area, largely confined to the same land-use type. We do not use the term community because we do not have direct data confirming species interactions[74].

### Land-use classification

To classify land-use types for each survey site, we used PREDICTS data to estimate the predominant type and stage of vegetation and the intensity of human use. First, we assigned sites to one of six vegetation classes: primary vegetation, secondary vegetation, plantation forests, pasture, cropland and urban. We then classified lightly or intensively used primary vegetation sites as disturbed primary vegetation, whereas minimal-use primary vegetation sites were classified as pristine primary vegetation (a proxy for undisturbed natural vegetation). Based on previous analyses showing reduced avian FD in intensely urbanized areas[55], we also split minimal-use urban sites from sites with more intensive

urbanization. Finally, to account for the effects of vegetation structure at different successional stages[75], we partitioned secondary vegetation according to age class (mature, intermediate, young, indeterminate; Supplementary Information). Indeterminate age secondary vegetation was removed from our dataset.

Our final dataset consisted of 98 study landscapes distributed across 6 continents (Fig. 2a and Supplementary Table 1), representing a total of 1,281 avian assemblages in 10 distinct land-use types: pristine primary vegetation ($n = 177$); disturbed primary vegetation ($n = 281$); mature secondary vegetation ($n = 44$); intermediate age secondary vegetation ($n = 77$); young secondary vegetation ($n = 86$); plantation forest ($n = 218$), pasture ($n = 184$); cropland ($n = 107$); and urban, including both minimal-use ($n = 46$) and intense-use ($n = 61$) urban landscapes.

### Functional trait data

Species traits can provide information about sensitivity to perturbations (response traits) and the impacts of species presence or absence on ecological function (effect traits). In both cases, data availability is often patchy for major taxonomic groups at a global scale[76]. We obtained morphometric measurements for all 3,696 species reported in our study landscapes from the AVONET trait database[44]. Species means were compiled for seven traits: beak length (culmen), beak length (tip-to-nares distance), beak depth, beak width, tail length, tarsus length and wing length (Supplementary Data 1). These traits have been shown to predict a range of key ecological niche axes, including diet and foraging strategy[16] (Supplementary Table 3). We also included data on the hand–wing index (HWI), a metric of wing elongation that predicts aerial lifestyle and dispersal distance in birds[77]. HWI is widely used as a proxy for dispersal ability[78]. Species mean values for all traits used in our study were calculated from an average of 11 individuals per species (41,515 individual birds measured in total).

Avian morphological traits are often intercorrelated because of an underlying association with body size[16]. Accordingly, all traits in our dataset were strongly correlated with the body size axis (Extended Data Fig. 2a), apart from HWI ($R = 0.22$). Following previous studies[40,79], we removed the association with body size through a two-step PCA, which reduced our seven linear morphometric traits into three niche axes related to ecological functions (Extended Data Fig. 2b). We performed two separate PCAs on trophic traits (related to beak morphology) and locomotory traits using all species in our dataset. In both cases, the first principal component (PC) was strongly correlated with body size; therefore, we used the second PC to represent the dominant axis of variation, which is effectively independent from body size (Extended Data Fig. 2c). We then performed a third PCA on the first PC scores from both the trophic and locomotory PCAs, taking the resultant first PC to represent the body size axis.

We use this metric of body size because it correlates strongly with body mass while also reflecting trophic niche differences. As the body size axis is extracted from linear measurements of beak, wing, tail and tarsus, it more closely reflects trophic and locomotory niches than conventional body-mass estimates. For example, our estimates of body size will distinguish between hummingbirds with equal body mass but different beak lengths, thereby reflecting associations with different foraging niches linked to pollination. Finally, we supplemented these three derived trait axes (trophic, locomotory and size) with a fourth morphological trait axis consisting of the log-transformed HWI (related to dispersal ability).

### Taxonomic matching

Bird species names and classifications vary over time and between different taxonomic treatments. This is problematical for global datasets based on published field surveys because different authors use a variety of taxonomic approaches, including English or local names. We converted all species names into a single taxonomy using published cross-walks[44] and verified taxonomic assignments with geographical

range maps[80] (Supplementary Information). This enabled accurate alignment with species trait data. Some taxa reported in survey data were impossible to assign directly to species because they were only identified to the genus level. Deleting these taxa would result in missing data, which can reduce the accuracy of FD estimates[81]. Instead, we created pseudo-species representative of the genus. Given that avian life history and morphological traits tend to be highly conserved within genera[44], we assigned trait values to pseudo-species by averaging the trait values of all congeners potentially occurring at the locality. To generate trait data for averaging, we used geographical range maps to provide a list of all members of the focal genus with geographical distributions overlapping the site location (Supplementary Information). We synthesized data for 73 pseudo-species in 133 of our 1,281 study assemblages.

### Dietary data

Birds mediate a wide range of ecological processes and services depending on their trophic interactions, including seed dispersal by frugivores and pest control by invertivores[82,83]. The morphological trait dataset used in this study was strongly correlated with avian diets and associated foraging behaviours[16,43]. However, the connection between morphology and dietary niche was noisy and weak in some taxonomic groups (Supplementary Information). Therefore, we also included standard diet classifications in functional metric calculations. We used published estimates of the proportion of species diets across nine major resource types: herbivore (aquatic), herbivore (terrestrial), nectarivore, granivore, frugivore, invertivore, vertivore (aquatic), vertivore (terrestrial) and scavenger. The data were extracted from a previous publication[16] and were primarily based on the EltonTraits dataset[84] with extensive updates and reorganization based on subsequent literature.

To define dietary groups for analyses, we used published data that classified species into trophic guilds according to their primary food source, with any species obtaining >60% of their diet from a single food type defined as a trophic specialist[44]. Species that obtained resources more equally across different food types were classed as omnivores[16,85]. Our main analyses focused on trophic guilds rather than omnivores because bird species with more specialized diets have a higher certainty of contributing to particular ecological roles and services[40]. However, generalists are often abundant, which suggests that they may contribute substantially at the population level to ecological processes such as seed dispersal and insect predation. Thus, non-specialist omnivores may help to stabilize ecosystems by providing additional redundancy. To assess whether our results were sensitive to trophic guild classification and inclusion of generalists, we generated broader dietary groupings containing all species that obtained >25% of their diet from a single food source. We then repeated our main analyses on these expanded groups (Extended Data Fig. 6 and Supplementary Information).

### Calculating FD and redundancy

To calculate functional metrics, we created trait probability densities (TPDs) using the TPD package in R[86]. The TPD approach uses species mean trait-values and intraspecific trait variation to calculate probabilistic hypervolumes in which the potential position and extent of occupancy for each species can be predicted along multiple trait axes. By using axes of trait variation to define ecological niche axes, TPD hypervolumes represent a Hutchinsonian niche[45]. We constructed species hypervolumes on the basis of their diet proportion data across nine major resource types and their distribution along four derived morphological trait axes (locomotory, trophic, dispersal and size). We then estimated FD for each assemblage as the total cumulative volume occupied in trait space by all species in the assemblage. Functional redundancy was calculated as the proportion of this total volume shared by multiple species, weighted by the relative abundance of species occupying the same regions of trait space[47] (Supplementary Information).

To create TPDs, we first calculated distance matrices using the R package gawdis, which is specifically designed to combine compositional data (such as our proportional diet data) into a single axis of variation[87] (Supplementary Information). We calculated distance matrices for each of our 98 study landscapes using diet and morphological data for all species present. Following previously described methods[88], we back-transformed our distance matrices into three-dimensional coordinates representing the relative position of each species in functional trait space (Supplementary Information).

Calculation of TPDs requires the estimated position of species means in trait space and the square-root of intraspecific variability. The latter is required to generate a probability kernel around each species-mean position. This intraspecific variation kernel (IV kernel) is taken as the niche of each species, represented in functional trait space. Ideally, the IV kernel is estimated by directly comparing measurements of many conspecific individuals and calculating the standard deviation across each dimension in functional trait space. However, as bird diet data are only available as species-mean estimates, we were unable to use this method for our main analysis. Thus, following previous studies[47,89,90], we approximated the IV kernel using a bandwidth estimator, which calculates an equally sized density kernel around each species mean based on the distances between co-occurring species (Supplementary Information). As the volume covered by the IV kernel can vary between assemblages, we calculated the dimensions of the kernel for each assemblage and took the square-root of the mean dimensions as our common IV kernel for each species (Supplementary Information). For each of our 98 study landscapes, we calculated a landscape-level TPD using the TPDsMean function, which generates a TPD using the species-mean positions in each assemblage, alongside the IV kernel dimensions. The functional trait space of each TPD was divided into 125,000 equally sized grid cells, with a value reflecting the likelihood of occupancy.

For each assemblage in each study landscape, we calculated FD and functional redundancy by running the in-built REND and redundancy functions from the TPD package across respective landscape-level TPDs[47]. Both functions filter the landscape-level TPDs for the species that occur in each assemblage. FD was then calculated as the number of grid cells with a likelihood of occupancy >0. Redundancy was calculated as the average number of species that could be removed from each grid cell without reducing the total occupied area of functional trait space[47] (Supplementary Information). By setting a minimal threshold for occupancy (>0), the FD value reflects the volume of occupied trait space independent of species abundance. Conversely, functional redundancy calculations are shaped by abundance and highly sensitive to the number of species that share a similar area of trait space (Supplementary Information).

To analyse the effect of land-use change on specific ecological roles, we calculated functional metrics across five species subsets to assess how functional trait structure changed within different trophic guilds in each assemblage. We focused on the entire assemblage ($n = 1,281$) and all generalist species ($n = 1,281$), as well as three key dietary guilds sampled across all land-use types: granivores ($n = 1,271$), frugivores ($n = 944$) and invertivores ($n = 1,274$). Sample sizes varied between guilds as TPDs cannot be created when three or fewer members of the guild are recorded in a given study landscape.

We did not standardize functional metrics by the number of species present and instead allowed both FD and redundancy to correlate with species richness. We assumed that each additional species added to an assemblage will either increase the number of trophic processes performed or increase the probability that multiple species deliver a particular function, or both[5]. We therefore allowed increased species richness to drive greater FD and redundancy values in larger assemblages. To examine how this decision influenced our results, we also conducted a supplementary analysis in which we modelled the relationship between functional resistance and a species richness

standardized redundancy metric (Extended Data Fig. 8 and Supplementary Information).

To address whether our method of estimating the dimensions of our IV kernel affected our conclusions, we recreated our TPDs using intraspecific variation calculated from multiple measurements of conspecific individuals, using morphological measurements extracted from AVONET[44] (Supplementary Information). The results of the sensitivity analyses based on these revised TPDs were similar, which showed the same general patterns of decline in FD or redundancy with land-use change (Extended Data Fig. 4).

As overall assemblage redundancy reflects the total amount of shared trait space in the assemblage, declines in redundancy are driven by either species losses or reduced amount of niche overlap per species (that is, niche differentiation). To decipher whether the effect of redundancy losses on functional resistance was driven by declines in species richness or reduced niche overlap per species, we also calculated the relative redundancy for each assemblage using in-built functions in the TPD package (Extended Data Fig. 8).

## Calculating sensitivity scores

To calculate functional vulnerability and functional resistance metrics, we began by scoring sensitivity to disturbance, which reflects the likelihood that a particular species would be removed from the assemblage by future perturbations. For each species in each assemblage, we generated two forms of sensitivity score: (1) trait-based and (2) rarity-based. Our main trait-based sensitivity score was based on four general response traits associated with extinction risk: geographical range size, body size, diet specialism and dispersal limitation[91,92] (Supplementary Information). This is a broad bandwidth score that uses traits associated with any form of disturbance (for example, fire, storms, drought, habitat loss, pollution or human exploitation) to reflect uncertainty regarding the source and severity of future perturbations. To provide a more explicit test of sensitivity associated with a known stressor, we calculated a secondary trait-based score using response traits associated with sensitivity to climate change: higher elevational distributions, narrower temperature niche, longer generations and dispersal limitation[93]. To calculate our rarity-based sensitivity scores, we extracted the inverse (that is, negative) abundance of each species in the assemblage based on the assumption that rarer species are more likely to be removed from an environment by population fluctuations[94]. All sensitivity scores were scaled by their standard deviation and centred to have a mean of zero.

## Functional vulnerability

Previous studies have associated functional stability with the distribution of unique functional traits[95] and have linked assemblage vulnerability to the variation in disturbance sensitivity of species traits[60]. We devised a new metric of functional vulnerability to encapsulate both these concepts. To estimate the relationship between trait redundancy and species sensitivity, we quantified the amount of redundancy provided by each species. Species redundancy was estimated as the change in assemblage redundancy after removal of the focal species. To calculate this value, we separately removed each species from the full assemblage and recalculated the assemblage redundancy. We then calculated two functional vulnerability scores as the covariance between species redundancy and either trait-based or rarity-based sensitivity scores using Spearman's rank correlation coefficient. We reversed the direction of the covariance so that a high positive correlation indicates that the most sensitive species in the assemblage tend to be the least redundant (that is, most unique).

## Functional resistance

To estimate functional resistance, we ran species extinction simulations using sequential removals and quantified the associated decline in FD. The sensitivity of species to environmental change is strongly influenced by their functional response traits, with certain trait combinations predicting the likelihood of local extinction[36,41]. Moreover, species abundances are often a strong indicator of local extinction risk, as rarer species are typically more sensitive to environmental perturbations[96,97]. Therefore, for each assemblage, we ran two extinction simulation scenarios by sequentially removing each species according to both sensitivity scores (trait-based and rarity-based).

Following previous methods[47,98–100], we plotted extinction curves to quantify how FD declines as the proportion of species occurring in the original assemblage is reduced to zero. Using a standard approach, we then measured the AUC as an estimate of functional resistance[99,100]. If an assemblage maintains constant FD when species are removed, the AUC remains large, which indicates high levels of functional resistance. Conversely, when extinctions drive declines in FD, the AUC decreases (Extended Data Fig. 1 and Supplementary Fig. 1). For each extinction curve, the AUC was measured using the MESS package in R[101].

As we were specifically interested in quantifying the pace of FD decline, we standardized the extinction curves by scaling the FD values between 1 and 0, where 1 equals the FD calculated for the full assemblage before any species was removed. This standardization prevented the magnitude of the FD value from influencing the AUC, which ensured that AUC values reflect variation in the shape of the extinction curve only[98]. For each assemblage, we calculated our functional resistance metric across the same five subsets of species as in the preceding FD and redundancy analyses: all species, all dietary generalists, granivores, frugivores and invertivores.

One drawback of the AUC approach is its sensitivity to the order in which species are lost from an assemblage. For example, when a single morphologically unique species is lost before other more redundant species, this causes a steep initial decline in FD. Therefore, following the same methods used to generate functional vulnerability values, we also estimated functional resistance as the half-life ($t_{1/2}$) of each extinction curve (Supplementary Information).

In our main simulations, we implemented extinction scenarios based on a set of traits associated with general disturbance. To assess the robustness of our results to methods and choice of traits, we conducted a series of sensitivity analyses. First, we calculated extinction curves generated using trait-based sensitivity scores based on a different combination of response traits specifically related to climate change tolerance. Second, we re-ran analyses under passive (probability-weighted) species loss scenarios, in which 0–2 species were removed at each time step, and the probability of a species being removed at each step was equal to its sensitivity score (Supplementary Information). Under this procedure, highly tolerant species were allowed to remain in the assemblage longer, and assemblages with a high proportion of tolerant species did not necessarily lose species at each time step, thereby increasing the AUC. Results of these sensitivity analyses were similar to our main analyses (Extended Data Fig. 7).

## Statistical analyses

To assess how the distribution of key functional traits were affected by land-use change, we subdivided assemblages ($n = 1,281$) into four categories: (1) primary vegetation, (2) secondary vegetation, (3) agriculture (including plantation forests) and (4) urban. We then constructed a set of univariate linear mixed-effects models with land-use as a single predictor variable and nine separate response variables covering all response and effect traits analysed in this study (see Supplementary Information for the rationale). Models were assessed by comparing coefficient estimates for each land-use type against primary vegetation. When coefficient estimates were positive, land-use change was inferred to filter species with low functional trait values.

To assess the impact of land-use on functional structure, we first constructed three univariate linear mixed-effects models. In all three models, land-use was the sole predictor variable and each model analysed the effect of land-use change on FD, functional redundancy or

functional vulnerability. Each model was conducted across all species and separately across four different data subsets related to dietary guild (generalists, granivores, frugivores and invertivores), and two subsets related to climatic region: tropical study landscapes and non-tropical study landscapes. For these models, land-use was split into ten distinct categories: pristine-primary vegetation, disturbed primary vegetation, mature secondary vegetation, intermediate age secondary vegetation, young secondary vegetation, plantation forests, pasture, cropland, minimal urban and intense urban.

To address how land-use change affects the functional resistance of bird assemblages, we ran six additional sets of univariate mixed-effects models. These analyses modelled the effects of land-use change on the functional resistance of each assemblage under both of our main species-loss scenarios (general trait-based AUC and rarity-based AUC) and our four alternative functional stability (general trait-based $t_{1/2}$, rarity-based $t_{1/2}$, climate trait-based AUC and passive AUC). For these final analyses, we also split land-use into four categories: (1) primary vegetation, (2) secondary vegetation, (3) agriculture (including plantation forests) and (4) urban. In line with FD and redundancy analyses, we conducted our functional resistance models separately across all species and our four dietary guild subsets (generalists, granivores, frugivores and invertivores).

All models were conducted using the lme4 package in R[102]. We added study landscape to account for among-study differences in sampling methods and study block to account for confounding variables related to temporal and geographical distinctions in each study. Models were interpreted by comparing the change in estimated regression coefficients for each land-use type using the functional metric value calculated for the least disturbed land-use category as our reference. Our hierarchical modelling approach did not require all land uses to be present in each study landscape to produce standardized regression coefficients. For most models, our least disturbed category was termed pristine primary vegetation, which accounted for 177 (13.8%) of our survey sites and was present in 50 out of our 98 study landscapes (51%). Although no primary vegetation landscape is entirely unaffected by human disturbance, this term is used to differentiate between more heavily disturbed vegetation types. For our functional stability models, we combined all primary vegetation sites into a single land-use type and used this broader classification as our least-disturbed land-use category.

To account for seasonal effects, we ensured that assemblages in the same study block were surveyed in the same season. Therefore, by including study block as a random effect, we removed detection biases arising from variation in sampling season. Moreover, incomplete sampling owing to undetected rare or cryptic species was reduced by aggregating survey sites into larger species assemblages to increase overall sampling depth (Supplementary Information).

We assessed all models for normality of residuals and heteroskedasticity and did not find that any of our models violated the assumptions of a linear model. Owing to the hierarchical structure of our data, it was difficult to incorporate covariance structures that account for spatial autocorrelation between local survey sites in the same study landscape into our global models. However, following a previously described method[11], we assessed spatial autocorrelation in study landscapes and study blocks separately using Moran's I and found that it did not affect our results (Extended Data Fig. 9).

### Reporting summary

Further information on research design is available in the Nature Portfolio Reporting Summary linked to this article.

### Data availability

All data are available at Zenodo (https://zenodo.org/records/17184411)[103]. Original survey datasets are available from PREDICTS (https://doi. org/10.5519/JG7I52DG). Bird traits for all study species are available from AVONET at Figshare (https://figshare.com/s/b990722d72a26b5b-fead)[104].

### Code availability

The code to conduct analyses and replicate figures is available at Zenodo (https://zenodo.org/records/17184411)[103].

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

**Acknowledgements** This research was supported by grants from the Natural Environment Research Council UK (NE/I028068/1 and NE/K016385/1 to J.A.T., NE/Z50404X/1 to D.P.E. and NE/X015262/1 to A.C.L.) and the UKRI Global Challenges Research Fund (ES/P011306/1 to J.A.T.). T.L.W. and P.A.W. were funded by Natural Environment Research Council studentships through the Science and Solutions for a Changing Planet Doctoral Training Programme. We thank J. Borer, S. Contu, X. Huang and G. Needler for compiling and processing survey data.

**Author contributions** This study was conceived and developed by J.A.T. and T.L.W., with input from A.P. and A.L.P. T.L.W. integrated datasets and ran all analyses with support from P.A.W. and A.P. D.P.E. and A.C.L. provided additional datasets. T.L.W. wrote the manuscript and designed all figures with extensive input from J.A.T. All authors contributed to subsequent drafts and gave final permission for publication.

**Competing interests** The authors declare no competing interests.

**Additional information**
**Correspondence and requests for materials** should be addressed to Joseph A. Tobias.

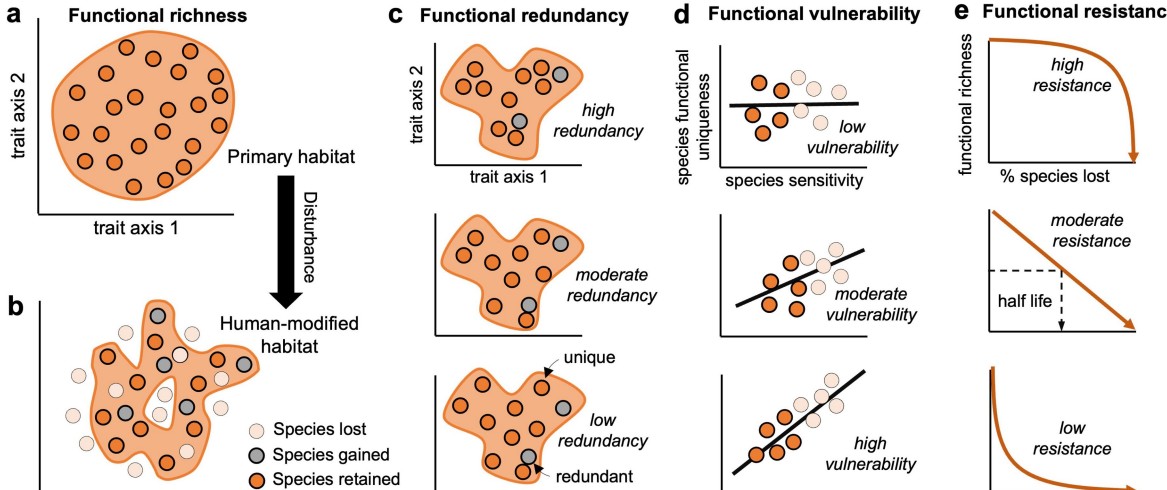

**Extended Data Fig. 1 | Hypothetical impacts of land-use change on functional structure of species assemblages.** Panels show diagrams clarifying concepts and terms used in this study. Disturbance drives non-random shifts in functional diversity (FD) because species with traits ill-suited to the modified environment are removed (pale orange circles), more tolerant species are retained (dark orange circles), and species with advantageous trait combinations are gained (grey circles). This turnover alters the functional trait space (orange polygon) and the assemblage's internal functional structure. The total volume of this space (functional richness) tends to decline as primary habitats (**a**) are converted into human-modified ones (**b**). To assess the effects of species loss and turnover, we use two primary metrics. Functional redundancy (**c**) reflects the amount of functional trait space shared by multiple species; high redundancy is identified when species are clustered in trait space. In contrast, functional vulnerability (**d**) is based on the relationship between how sensitive and how unique a species is within the assemblage. Assemblages are highly vulnerable when numerous sensitive species occupy unique areas of trait space. Functional resistance (**e**) describes the rate at which functional richness declines as species are removed from the assemblage. Assemblages with high redundancy can absorb the loss of more species with minimal effect on the total functional trait space, whereas low vulnerability means the most sensitive species are unlikely to provide a unique function. Both high redundancy and low vulnerability contribute to high resistance, allowing the assemblage to absorb more species losses with a slower decline in overall function. We quantify functional resistance as either the Area Under the Curve (AUC) or half-life of the extinction curve (see Supplementary Fig. 1).

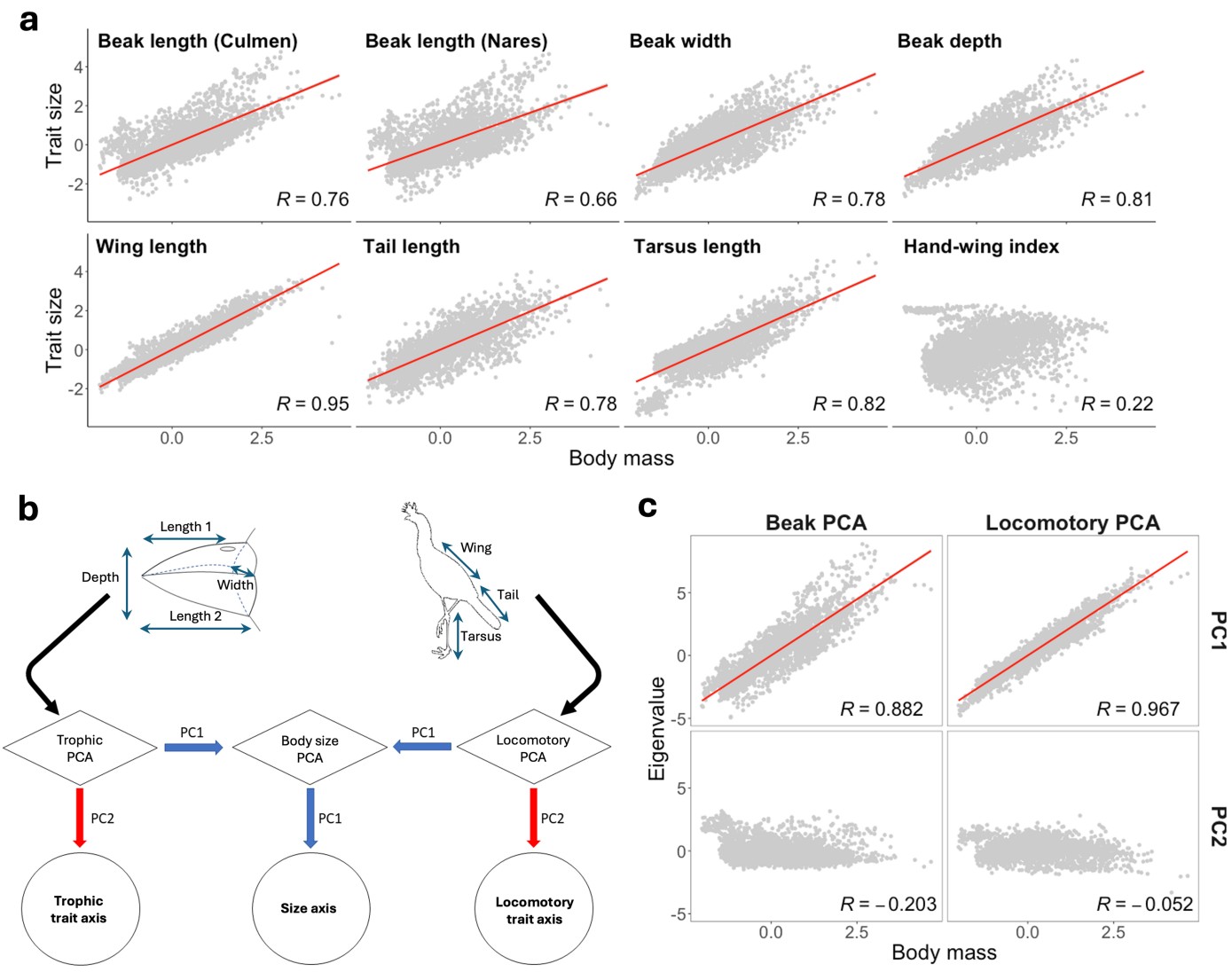

**Extended Data Fig. 2 | Using traits to generate independent niche dimensions.** Raw measurements of trait dimensions reflect overall size of the measured specimen and are strongly correlated to body mass (**a**). Only hand-wing index is uncorrelated with body mass as it is calculated as a ratio of two separate traits[44]. To avoid functional metrics simply reflecting variation in overall body mass, we use a 2-step principal component to generate three distinct trait axes (**b**). We separated seven morphological traits correlated with body mass into a trophic axis based on beak measurements (culmen length, tip-to-nares, depth, width) and a locomotory axis (wing length, tail length, tarsus length). We then performed two separate PCAs on trophic and locomotory traits using all species in the dataset. The first principal components (PCs) are correlated with body mass, so we used the second principal component to represent the dominant axis of variation, independent of body mass, for beak shape and locomotion, respectively (**c**). After taking these second PCs as our trophic and locomotory trait axes, we performed a third PCA using both first PCs of the original PCAs, then used the first PC of this final PCA to reflect the size axis. Traits were extracted from AVONET[44] for all species in this dataset (n = 3696), log-transformed and scaled to their standard deviation. R-values are Pearson's correlation coefficients; red line indicates the slope of the correlation coefficient estimate. Beak illustration was created by the authors. Bird outline adapted from PhyloPic (https://www.phylopic.org). *Cariama cristata* created by George Edward Lodge (vectorized by T. Michael Keesey) under a Public Domain Mark 1.0 licence (Supplementary Table 4).

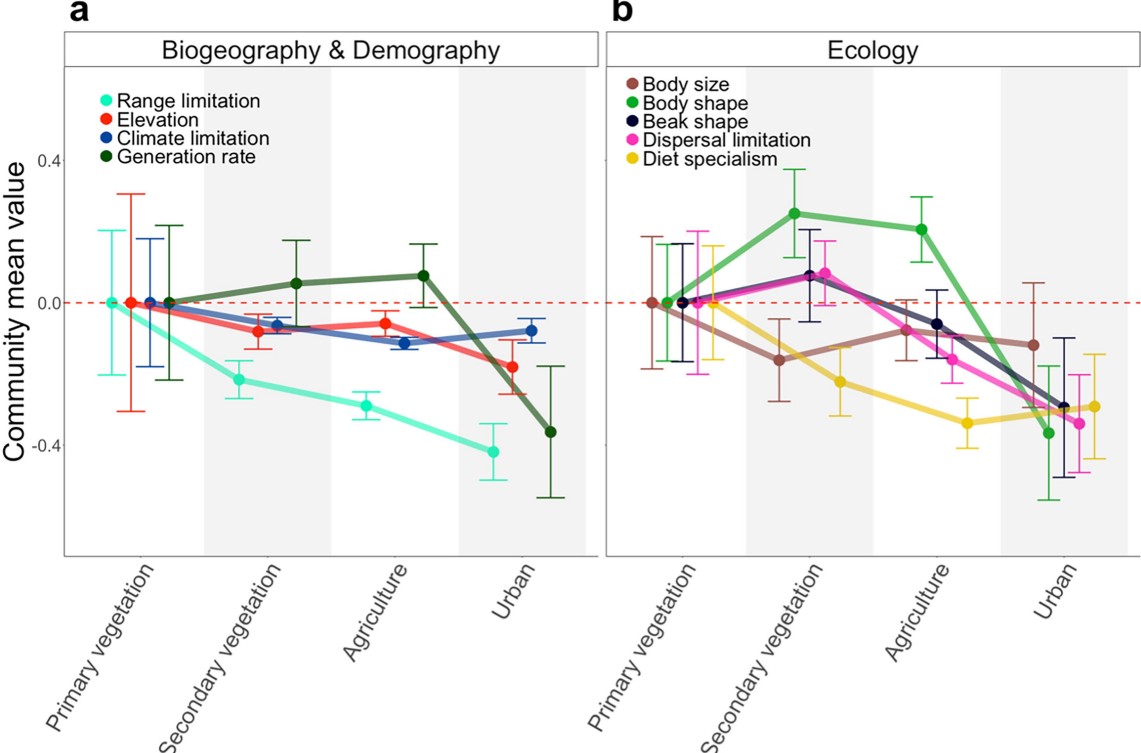

**Extended Data Fig. 3 | Impacts of land-use change on species traits associated with sensitivity to disturbance.** Results of global models ($n$ = 3696 bird species; $n$ = 1281 assemblages). **a**, Estimated change in community mean (CM) trait-values for response traits related to biogeography and demography: Range limitation = inverse (negative) total geographical area (km²) of breeding and non-breeding ranges; Elevation = minimum elevation recorded per species[105]; Climate limitation = inverse (negative) mean annual temperature seasonality across the species range[106]; Generation rate = age of first breeding, longevity and adult survival rate[107]. **b**, Estimated change in CM for ecological traits: body size, body shape and beak shape calculated from morphological traits[44] (see Methods; Extended Data Fig. 2a); dispersal limitation = inverse (negative) Hand-wing

Index; diet specialism = proportion of diet obtained from the primary food-source. Higher values for body shape indicate longer wing/tail and shorter tarsus; for beak shape indicate thinner, more elongated beaks; for diet specialism indicate greater limitation to single food source. Rationale for trait selection in Supplementary Tables 2–3. CMs for each land-use category are compared against a standardized baseline value (set to 0, dashed red line) calculated for pristine primary vegetation. To aid comparison, all traits were standardized by their standard deviation prior to analyses. Points shown are coefficient estimates from nine separate linear mixed effects models; error-bars indicate 95% confidence intervals.

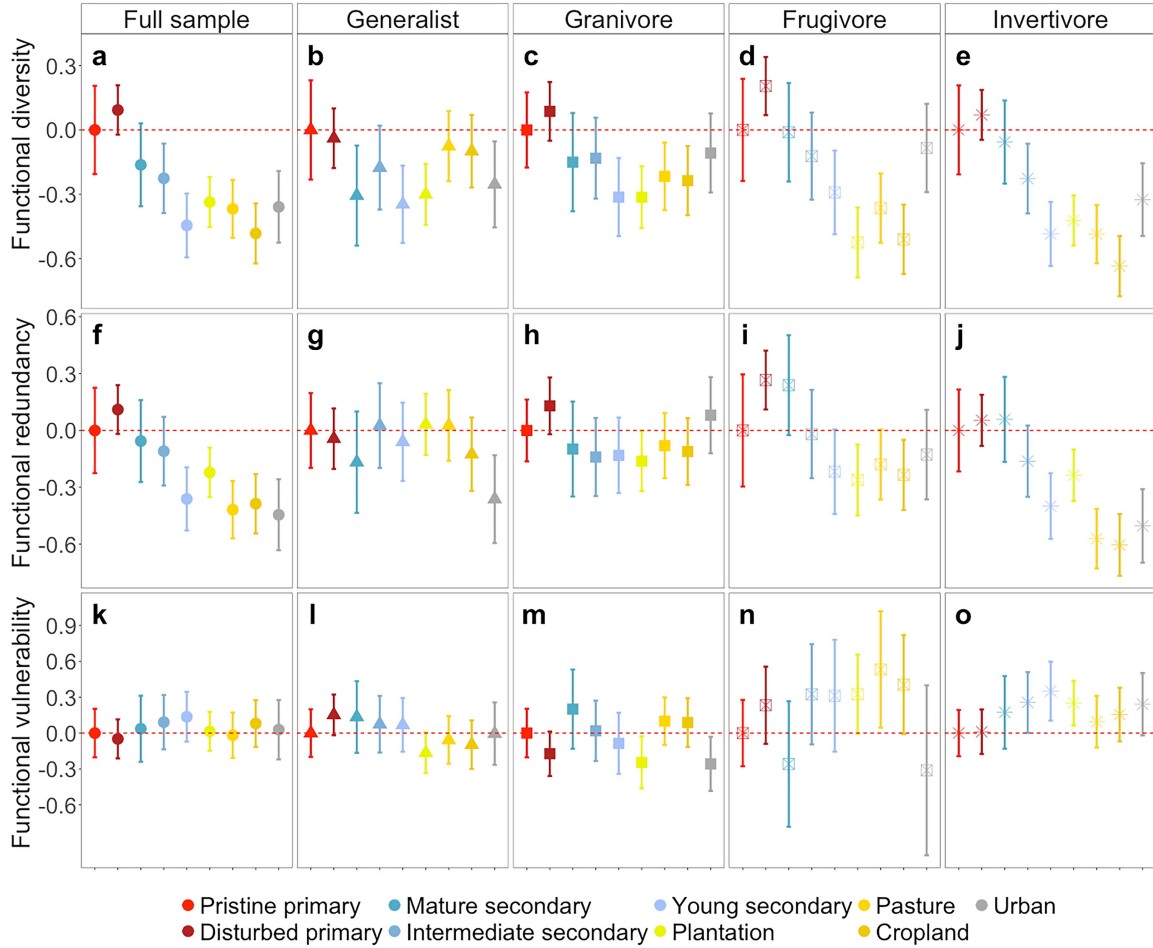

**Extended Data Fig. 4 | Functional metrics of bird assemblages using alternative method for calculating intraspecific variation.** We estimated intraspecific variation using individual-level measurements for species traits extracted from AVONET[44]. Panels show outputs from univariate mixed effects models estimating shifts in functional metrics: (**a-e**) functional diversity (FD; measured as functional richness); (**f-j**) functional redundancy (FRed); and (**k-o**) Functional vulnerability (FV; calculated using general sensitivity scores). Models were run separately on the full sample (*n* = 1281, 1281) and dietary subsets of generalists (*n* = 1281, 1185), granivores (*n* = 1271, 775), frugivores (*n* = 944, 506), and invertivores (*n* = 1274, 1224). Two sample sizes are given because FD and FRed are calculated for all assemblages, whereas FV can only be measured when >1 species of a single guild is present. Species are assigned to a trophic guild if they consume >60% of their diet from a single food source; all other species are defined as generalists. We omit dietary data from these calculations due to a lack of individual-level estimates. FV is calculated using Spearman's rank correlation coefficient between species-level redundancy and general trait-based sensitivity scores (see Methods). Functional metrics are compared against a standardized baseline value (set to 0, dashed red line) calculated for pristine primary vegetation (forests, grasslands, shrublands). Points shown are coefficient estimates; error-bars indicate 95% confidence intervals. Response variables were squareroot transformed prior to analysis and scaled by their within-group standard deviation to aid comparison.

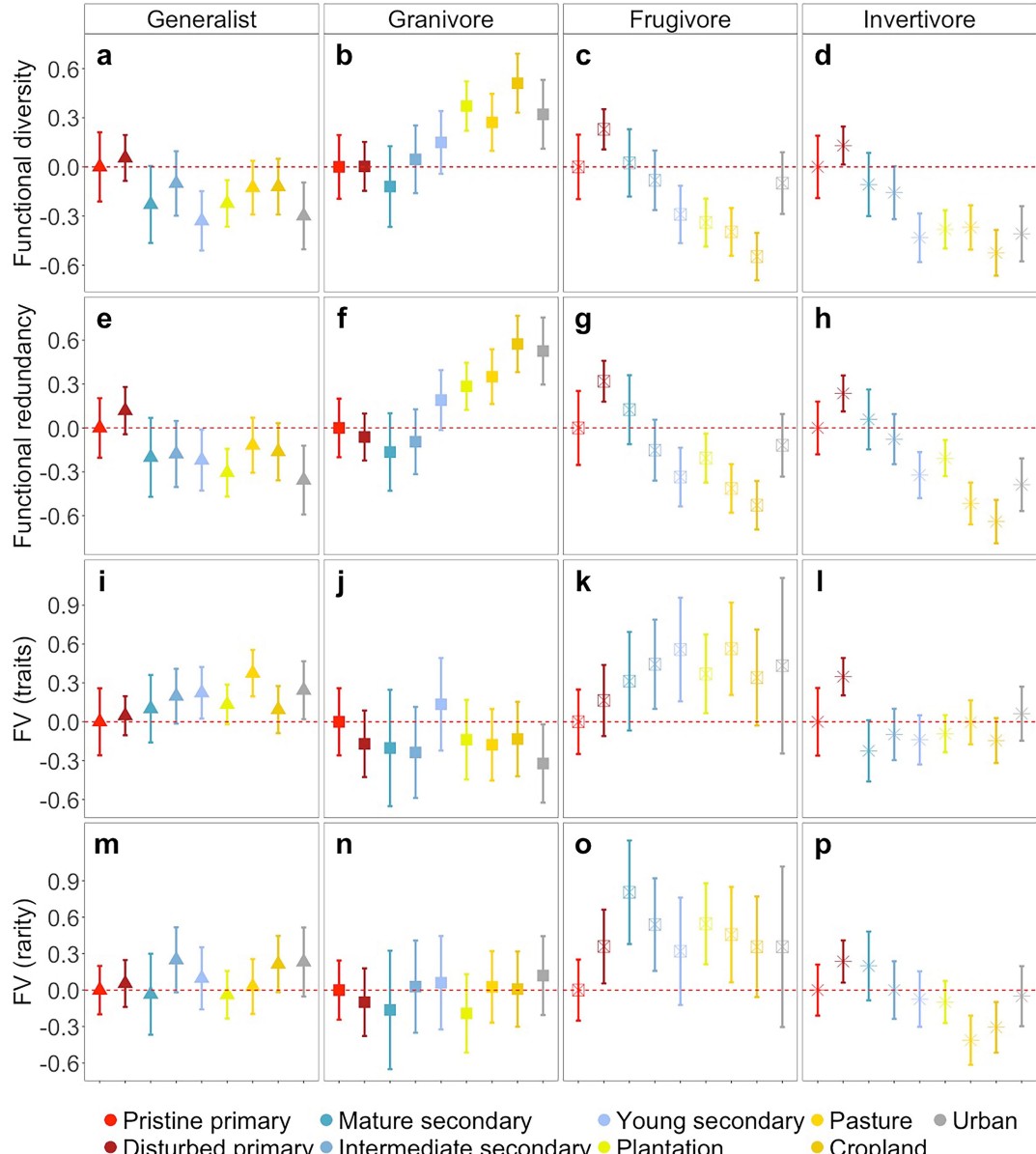

**Extended Data Fig. 5 | Land-use change alters functional diversity and redundancy of avian assemblages.** Panels show outputs from univariate mixed effects models estimating shifts in functional metrics: (**a-d**) functional diversity (FD; measured as functional richness); (**e-h**) functional redundancy (FRed); and functional vulnerability (FV) calculated using general trait-based sensitivity scores (**i-l**) and rarity-based sensitivity scores (**m-p**). Models were run separately on samples of generalists (*n* = 1281, 1185), granivores (*n* = 1271, 775), frugivores (*n* = 944, 506), and invertivores (*n* = 1274, 1224). Two sample sizes are given because FD and FRed are calculated for all assemblages, whereas FV can only be measured when >1 species of a single guild is present. Species are assigned to a trophic guild if they consume >60% of their diet from a single food source; all other species are defined as generalists. Functional metrics are compared within trophic guilds against a standardized baseline value (set to 0, dashed red line) calculated for pristine primary vegetation (including forests, grasslands and shrublands). Points shown are coefficient estimates; error-bars indicate 95% confidence intervals. Response variables were squareroot transformed prior to analysis and scaled by their within-group standard deviation to aid comparison.

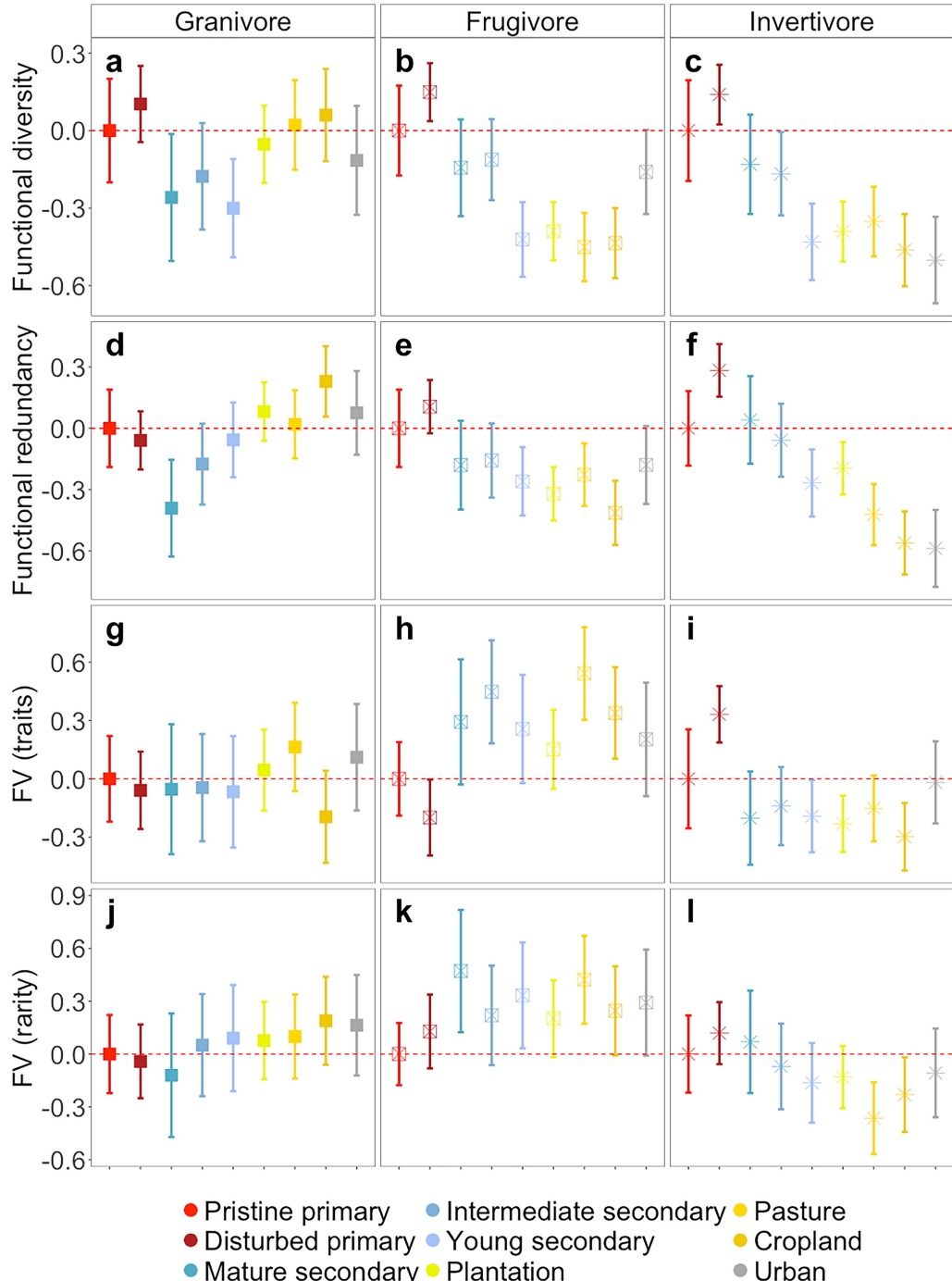

**Extended Data Fig. 6 | Impacts of land-use change on expanded trophic guilds associated with key ecological functions.** Panels show outputs from univariate mixed effects models estimating shifts in functional metrics after land-use change: (**a-c**) functional diversity (FD; measured as functional richness); (**d-f**) functional redundancy (FRed); and functional vulnerability (FV) calculated using general trait-based sensitivity scores (**g-i**) and rarity-based sensitivity scores (**j-l**). Models were run separately on samples of granivores ($n = 1271, 1095$), frugivores ($n = 1271, 1033$), and invertivores ($n = 1274, 1269$). Two sample sizes are given because FD and FRed are calculated for all assemblages, whereas FV can only be measured when >1 species of a single guild is present. Samples are larger than in Extended Data Fig. 4 because a wider set of species are assigned to a trophic guild if they consume >25% of their diet from a single food source. Functional metrics are compared within trophic guilds against a standardized baseline value (set to 0, dashed red line) calculated for pristine primary vegetation (including forests, grasslands and shrublands). Points shown are coefficient estimates; error-bars indicate 95% confidence intervals. Response variables were squareroot transformed prior to analysis and scaled by their within-group standard deviation to aid comparison.

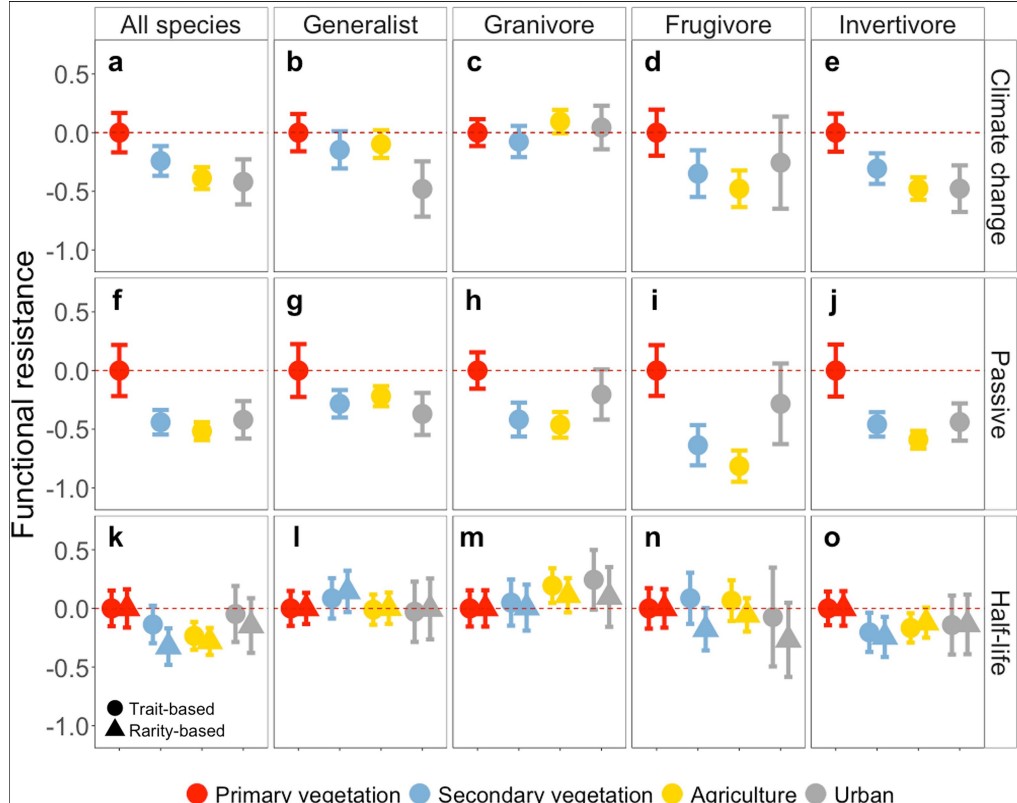

**Extended Data Fig. 7 | Functional resistance of bird assemblages under alternative projections of future extinction.** Results show predicted functional resistance of species assemblages for each land-use type in response to different extinction scenarios and calculations: climate change (**a-e**), passive (probability-weighted) species loss (**f-j**), half-life (**k-o**). The climate change scenario uses a set of response traits linked to climate sensitivity; the passive scenario assigns a probability of extinction based on general response traits (see Methods). In both simulations, functional resistance is calculated as the area under the extinction curve (AUC; Extended Data Fig. 1e; Supplementary Fig. 1) for each assemblage. Functional resistance calculations calculated using the half-life ($t_{1/2}$; Extended Data Fig. 1e; Supplementary Fig. 1) is measured using extinction curves generated under trait-based (circles) and rarity-based (triangles) species-loss scenarios. Under the trait-based scenario, species are removed based on general response traits; under the rarity-based scenario, species are removed in reverse order of abundance (see Methods). Functional resistance is compared within all species ($n$ = 1281 assemblages) and generalists ($n$ = 1281 assemblages), as well as for trophic guilds defined as standard dietary specialists (see Methods): granivores ($n$ = 1271 assemblages), frugivores ($n$ = 944 assemblages), and invertivores ($n$ = 1274 assemblages). Dashed red line indicates the standardized predicted functional resistance to species losses (set to 0) for assemblages in pristine primary vegetation (forests, grasslands, shrublands). Points shown are coefficient estimates, and error-bars indicate 95% confidence intervals.

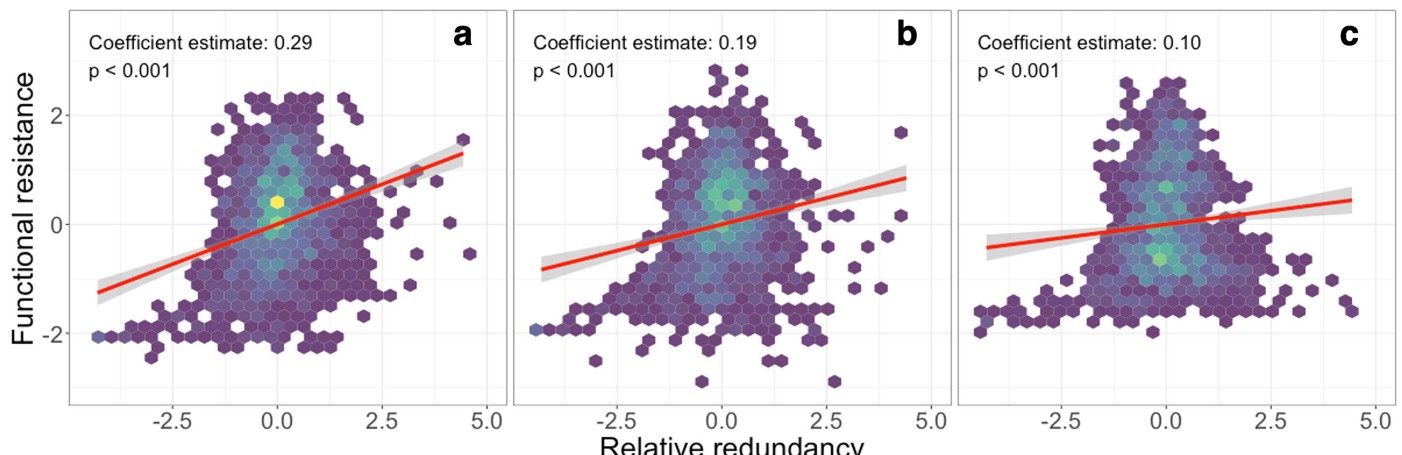

**Extended Data Fig. 8 | Functional resistance as a product of functional trait space overlap per species.** To understand whether functional resistance arises from total species richness or from the degree of trait overlap among species, we decomposed functional redundancy into two components: species richness and the average trait space overlap per species (relative redundancy). To test whether niche overlap alone can explain variation in assemblage resistance, we modelled the relationship between relative redundancy and functional resistance. **a-c**, Panels show the results of linear models assessing the relationship between relative redundancy and functional resistance when extinction curves are generated by general trait-based sensitivity scores (**a**), climate sensitivity scores (**b**), and through a passive (probability-weighted) species loss procedure (**c**; see Methods). Under scenarios **a** and **b**, a species is removed at each time step; in **c**, this procedure is relaxed so that species with low sensitivity can survive indefinitely. Each point represents a species assemblage (*n* = 1281); colour scale indicates level of overlap from low density (purple) to high density (yellow). Results shown are the coefficient estimate from three separate linear models (red line) alongside the 95% confidence intervals (shaded area). Functional resistance and relative redundancy are positively related under each scenario suggesting that functional resistance is driven by relative redundancy per species rather than species richness alone.

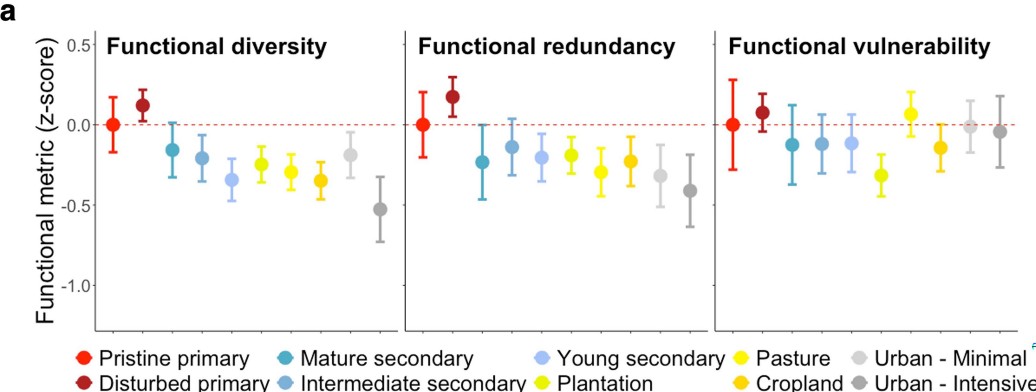

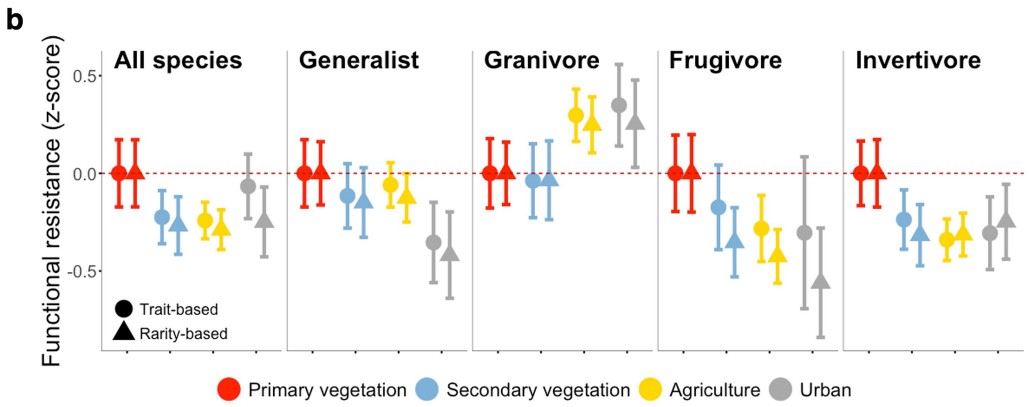

**Extended Data Fig. 9 | Functional metrics of bird assemblages after removal of spatially autocorrelated data.** Within each study-landscape we measured the spatial auto-correlation across survey sites for each functional metric using Moran's I (see Methods). Results are outputs from univariate mixed effects models exploring effects of land-use change on each of our assemblage-level functional metrics (**a**) after removing study landscapes that showed clear spatial autocorrelation between functional metrics calculated at different survey sites. Sample sizes vary as the number of spatially auto-correlated study landscapes is differs according to functional metric; Functional diversity (measured as functional richness; n = 1177 assemblages), functional redundancy ($n$ = 899 assemblages) & functional vulnerability (n = 973 assemblages). We repeated this process for the functional resistance across different dietary-groups (**b**) calculated as the Area Under the Curve (AUC; see Methods) of extinction curves generated from two different species-loss scenarios: trait-based (circles; removal on basis of species response traits) and rarity-based (triangles; removal in reverse order of abundance). Results are shown for the entire-assemblage ($n$ = 1269 assemblages), trophic generalists ($n$ = 1242 assemblages), and three separate trophic guilds: granivores ($n$ = 1025 assemblages), frugivores ($n$ = 630 assemblages), and invertivores ($n$ = 1247 assemblages). Dashed red line indicates the standardized predicted functional resistance (set to 0) for assemblages in pristine primary vegetation (forests, grasslands, shrublands). Results shown are coefficient estimates from two separate linear mixed-effects models with 95% confidence intervals. All response variables were squareroot transformed prior to analysis and then scaled by their standard deviation to aid comparison.

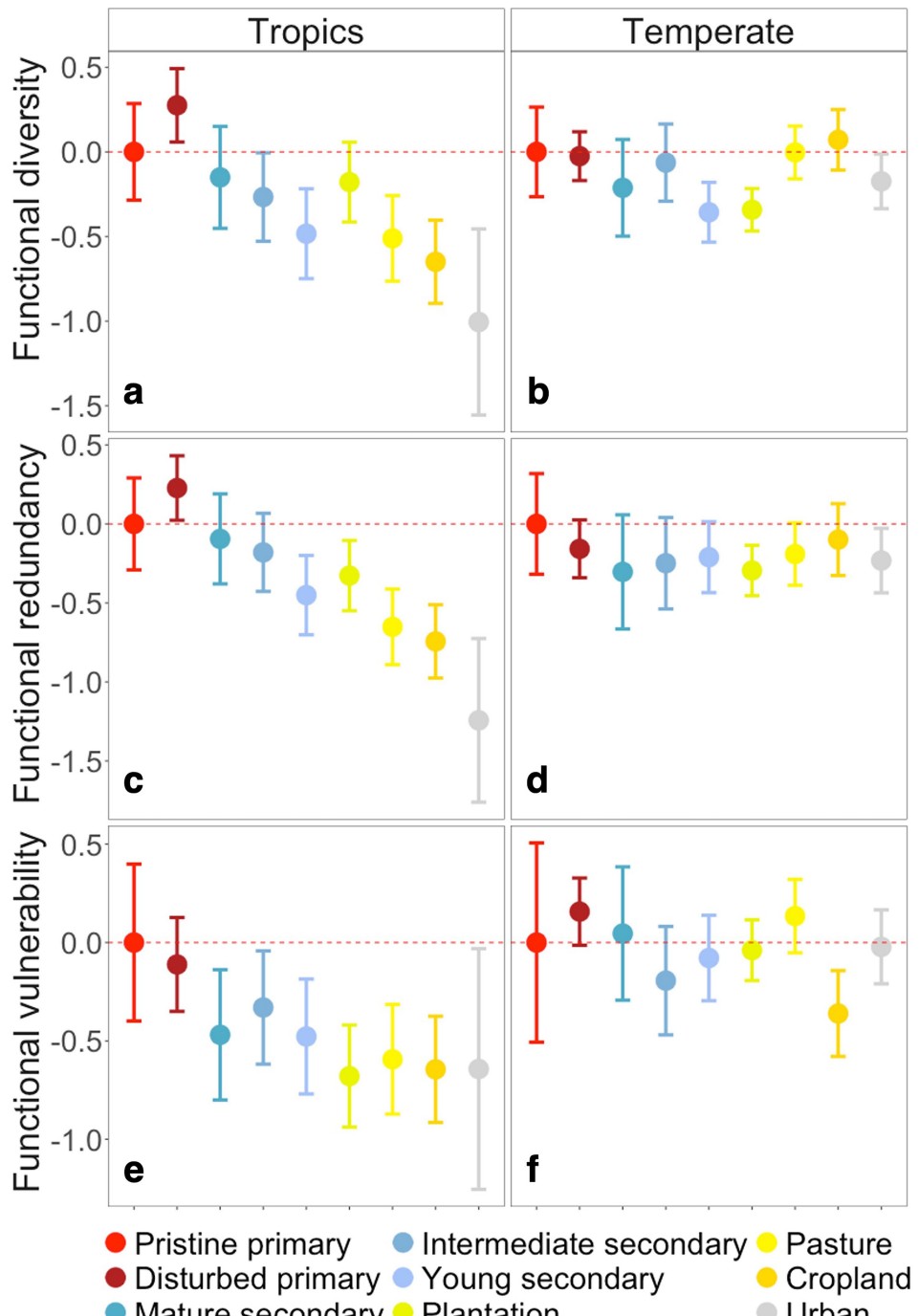

**Extended Data Fig. 10 | Impacts of land-use change on structure and function of avian assemblages vary by latitude.** Results are outputs from univariate mixed effects models exploring effects of land-use change on (**a**, **b**) assemblage functional diversity (FD; measured as functional richness); (**c**, **d**) functional redundancy (FR), and (**e**, **f**) functional vulnerability (FV). Metrics are calculated and analyzed separately for Tropical ($n$ = 613), and Non-tropical ($n$ = 668) survey sites. FV is calculated using Spearman's rank correlation coefficient between species-level redundancy and trait-based general sensitivity scores (see Methods). In all cases, metrics are compared to the baseline value (dashed red line) for pristine primary vegation (forests, grasslands, shrublands and wetlands). Points shown are coefficient estimates, and error-bars indicate 95% confidence intervals. Response variables were squareroot transformed prior to analysis and then scaled by their standard deviation to aid comparison (the transformation step was not possible for FV).

# Reporting Summary

## Statistics

For all statistical analyses, confirm that the following items are present in the figure legend, table legend, main text, or Methods section.

| n/a | Confirmed | |
|---|---|---|
| ☐ | ☒ | The exact sample size (*n*) for each experimental group/condition, given as a discrete number and unit of measurement |
| ☒ | ☐ | A statement on whether measurements were taken from distinct samples or whether the same sample was measured repeatedly |
| ☐ | ☒ | The statistical test(s) used AND whether they are one- or two-sided<br>*Only common tests should be described solely by name; describe more complex techniques in the Methods section.* |
| ☐ | ☒ | A description of all covariates tested |
| ☐ | ☒ | A description of any assumptions or corrections, such as tests of normality and adjustment for multiple comparisons |
| ☐ | ☒ | A full description of the statistical parameters including central tendency (e.g. means) or other basic estimates (e.g. regression coefficient) AND variation (e.g. standard deviation) or associated estimates of uncertainty (e.g. confidence intervals) |
| ☐ | ☒ | For null hypothesis testing, the test statistic (e.g. *F*, *t*, *r*) with confidence intervals, effect sizes, degrees of freedom and *P* value noted<br>*Give P values as exact values whenever suitable.* |
| ☒ | ☐ | For Bayesian analysis, information on the choice of priors and Markov chain Monte Carlo settings |
| ☐ | ☒ | For hierarchical and complex designs, identification of the appropriate level for tests and full reporting of outcomes |
| ☐ | ☒ | Estimates of effect sizes (e.g. Cohen's *d*, Pearson's *r*), indicating how they were calculated |

*Our web collection on statistics for biologists contains articles on many of the points above.*

## Software and code

Policy information about availability of computer code

| Data collection | No code was used for data collection. Data was collected by contacting reasearchers for their survey data. Data wrangling of AVONET and PREDICTS datasets used tidyverse (Version 2.0.0) in R (Version 4.4.2). All code is available at: 10.5281/zenodo.17184411 |
|---|---|
| Data analysis | Data analysis was undertaken using R (Version 4.4.2). We used the TPD package (version 1.1.0) to calculate functional diversity, redundancy and vulnerability metrics. MESS (Version 0.5.12) to calculate SUC resistance metric. We used lme4 (Version 1,1-36) for linear mixed effects models. Gawdis (Version 0.1.5) to generate distance matrices. Ks (Version 1.15.1) for plug-in density kernels. No custom codes were used for analysis. All code is available at: 10.5281/zenodo.17184411 |

For manuscripts utilizing custom algorithms or software that are central to the research but not yet described in published literature, software must be made available to editors and reviewers. We strongly encourage code deposition in a community repository (e.g. GitHub). See the Nature Portfolio guidelines for submitting code & software for further information.

## Data

Policy information about availability of data

All manuscripts must include a data availability statement. This statement should provide the following information, where applicable:
- Accession codes, unique identifiers, or web links for publicly available datasets
- A description of any restrictions on data availability
- For clinical datasets or third party data, please ensure that the statement adheres to our policy

> All data are available at 10.5281/zenodo.17184411. Original survey datasets are available from PREDICTS (https://doi.org/10.5519/jg7i52dg). Bird trait database is available from AVONET (https://figshare.com/s/b990722d72a26b5bfead).

## Research involving human participants, their data, or biological material

Policy information about studies with human participants or human data. See also policy information about sex, gender (identity/presentation), and sexual orientation and race, ethnicity and racism.

| | |
|---|---|
| Reporting on sex and gender | NA |
| Reporting on race, ethnicity, or other socially relevant groupings | NA |
| Population characteristics | NA |
| Recruitment | NA |
| Ethics oversight | NA |

Note that full information on the approval of the study protocol must also be provided in the manuscript.

# Field-specific reporting

Please select the one below that is the best fit for your research. If you are not sure, read the appropriate sections before making your selection.

☐ Life sciences   ☐ Behavioural & social sciences   ☒ Ecological, evolutionary & environmental sciences

For a reference copy of the document with all sections, see nature.com/documents/nr-reporting-summary-flat.pdf

# Ecological, evolutionary & environmental sciences study design

All studies must disclose on these points even when the disclosure is negative.

| | |
|---|---|
| Study description | We calculate various functional metrics for 1281 bird assemblages globally. We test how land-use change effects the functional diversity, functional redundancy, functional vulnerability & ultimately functional resistance using hierarchical mixed effects linear models. We compare the functional metrics in least disturbed habitats to those in disturbed/human-modified habitats. |
| Research sample | 1281 bird assemblages. This was selected to generate a global sample. The majority of data is extracted from the PREDICTS database (https://data.nhm.ac.uk/dataset/release-of-data-added-to-the-predicts-database-november-2022 & https://data.nhm.ac.uk/dataset/the-2016-release-of-the-predicts-database) . We also add 2 studies from the Amazon (https://doi.org/10.1073/pnas.2202310119) and Bornean rainforests (https://doi.org/10.1890/14-0010.1) to ensure we represent the most undisturbed areas. |
| Sampling strategy | Selection from the PREDICTS database to ensure consistency amongst studies. We removed various unacceptable studies which did not target the entire assemblage or contain abundance information. |
| Data collection | All data is recorded by on-the-ground samplers from 98 different published studies. Details can be found in the supplementary data. All have been accessed from the following 4 sources: PREDICTS database (https://data.nhm.ac.uk/dataset/release-of-data-added-to-the-predicts-database-november-2022 & https://data.nhm.ac.uk/dataset/the-2016-release-of-the-predicts-database). We also add 2 studies from the Amazon (https://doi.org/10.1073/pnas.2202310119) and Bornean rainforests (https://doi.org/10.1890/14-0010.1) |
| Timing and spatial scale | We selected datasets from 1990-present. This was to be in keeping with the land use harmonization project which can be used to project our data into the future. This also allowed us to be in keeping with other studies from the same dataset. Spatial scale was global sets of assemblages, from different published studies. Each study varied in geographical extent ranging from less than 1km² to 125,000km² |
| Data exclusions | Exclusion criteria was based on whether or not the sampling design targeted the entire assemblage and whether the data was published with abundances rather than presence-absence. |

| Reproducibility | All attempts to repeat analyses were successful |
|---|---|
| Randomization | Species were grouped into assemblages. Assemblages were based upon predefined studies and study-blocks. A single study-block can contain multiuple assemblages, but all assemblages surveyed within same study block were surveyed in the sampling season and are in close proximity. We used hierarchical modelling with study and study block used as random effects to ensure assemblages are effectively compared within study blocks. |
| Blinding | Blinding was not used as data had already been acquired by the PREDICTS project. To maximize sample size we selected all appropriate studies from the PREDICTS database. This allows us a global sample. |

Did the study involve field work? ☐ Yes ☒ No

# Reporting for specific materials, systems and methods

We require information from authors about some types of materials, experimental systems and methods used in many studies. Here, indicate whether each material, system or method listed is relevant to your study. If you are not sure if a list item applies to your research, read the appropriate section before selecting a response.

## Materials & experimental systems

| n/a | Involved in the study |
|---|---|
| ☒ | ☐ Antibodies |
| ☒ | ☐ Eukaryotic cell lines |
| ☒ | ☐ Palaeontology and archaeology |
| ☒ | ☐ Animals and other organisms |
| ☒ | ☐ Clinical data |
| ☒ | ☐ Dual use research of concern |
| ☒ | ☐ Plants |

## Methods

| n/a | Involved in the study |
|---|---|
| ☒ | ☐ ChIP-seq |
| ☒ | ☐ Flow cytometry |
| ☒ | ☐ MRI-based neuroimaging |

## Plants

| Seed stocks | NA |
|---|---|
| Novel plant genotypes | NA |
| Authentication | NA |

