## [Peer Review File · Nature]

Land-use change undermines the stability of avian functional diversity

Corresponding Author: Professor Joseph Tobias

Version 0:

Reviewer comments:

Referee #1

(Remarks to the Author)

This study combines two large datasets AVONET, for bird traits and the land use biodiversity impacts dataset PREDICTS to great effect, and shows that land use change causes major shifts to the functional diversity and composition of bird assemblages. It also shows, via some rather advanced statistical analyses and simulations, that these changes could affect the stability of the functions that these communities provide.

The paper is impressive and the results are important, but as currently presented the emphasis is a little heavily statistical in presentation with not too much of the underlying biology discussed. I agree with the tight logical flow of arguments presented in the paper, these nicely underpin the analysis choices taken, but this does not prevent the paper from being based upon a series of contingent assumptions where fairly abstract metrics are related to each other e.g. that a simulated extinction will lead to a certain loss of trait space that might in turn affect function. This criticism does not mean that I am unsupportive of the paper but I do think it would benefit from two main things.

The first would be a more simple and fundamental description of which traits are changing in response to land use and why. This simple result – a description of the general trait changes imposed on bird communities worldwide by land use would be interesting in its own right, and enough to merit publication in a high impact journal, given the generality of the results. Further, it would make a good basic foundation for the more advanced analysis that could then follow. By going back to the basic natural history of these birds a bit it would also make the work less abstract and grounded in ecological reality.

The second would be a less certain wording when interpreting the results of the more advanced and assumption rich analyses. While the logical flow of arguments is strong the paper does rely upon a series of complex stacked assumptions that requires a few 'leaps of faith by the reader'. The link between trait measures and real ecosystem level functioning measures is not clear or well established. It is a bit of a jump to go from a beak length distribution to an actual function such as the rate of invertebrate consumption or frequency of seed dispersal. How big an impact that these changes in functional composition would have on ecosystem functioning therefore semi-speculative and so wording should be less certain to reflect this. e.g.- 'stability likely reduced'.

Another key concern I have considering the true biological meaning of the results is whether the lost functioning of the community is still needed to maintain ecosystem functions in the new environment? For example, does the loss of a fruit eating seed disperser matter if the new system is a city or monoculture that contains no naturally planted fruit producing trees? It would be nice to see some discussion of this. The paper is generally missing this type of biological discussion, particularly at the function level. Adding more information and insight in this area would make the paper more accessible to a wide readership, and thus increase its impact.

Finally, while the intro displays excellent grasp of and use of ecological theory, and does a good job of navigating a series of complex concepts in a few lines, it may benefit from adding the distinction of effect and response traits, as it's a somewhat

key concept. E.g. in explaining why redundant species at the effects level can still stabilize function due to differences in response traits. While I understand the necessity of keeping things simple in a paper of this kind there are a few places where adding this distinction would make the arguments flow more logically. The concept does come in later (e.g. line 170, so maybe good to set it up from the onset).

Overall, this is an accomplished study, that is well written, well thought through, and important. A few changes would make it more plausible, grounded and accessible. I feel these comments should be addressable and hope these and the more specific ones below are useful in revising the paper.

Specific comments

33- resistant or resilient? (or disturbance tolerant). Given the papers focus on stability these terms from the stability literature need to be used carefully.

43- likely destabilize

46- reveal>indicate

56- Le Provost 2021, Nat Comms may be a more appropriate reference from this project- more taxa, more directly linked to diversity responses

67-69- is this argument essential to the paper?

84-88- This makes logical sense but how much empirical evidence is there for it, especially when direct measures of function are considered?

93- effects groups?

104- really like this question

209- This might be a good place to discuss whether the lost functioning is 'needed' in the remaining ecosystem. Perhaps not in some cases, and this should be acknowledged, but it reduces options for sure- e.g. if future restoration to the original state was required.

246- Allan et al 2015 did not study functional diversity

253- resilient to what? Again, resistance and resilience terms are not clear

276- ecosystem

298- Would be good to give a 1-2 line overview of predicts here to readers unfamiliar with it

341- are these all effects traits? the classification of later traits as response suggests so, but it's not entirely clear.

363- I guess the net result is that all the measures are 'relative to body size'. Would be nice to have a little explanation here

367-368- might be nice to say a little more about what functioning these axes relate to

410-417- This is a particularly complex and difficult to understand section

421- these are not functions, but community properties that may be related to them- this distinction is not clear in the paper in general

487- Thus

520-523- This seems like cherry picking of data- why does its variability make it invalid, maybe it's just a variable community type?

535- was pristine vegetation present in all cases?

Fig 1. Right side of figure is small and hard to read

Figure 3. (and others) 'coefficient estimate' is not an intuitive label – it could be anything, so the reader has to rummage around to find out what they are looking at. Please present a more intuitive label e.g. extinction-FD slope steepness. It took a few reads to get this figure, if I did at all. Important to get clear as it's the main results figure.

Extended 1. Coefficient estimate could also be more intuitively labelled

Extended 2- again, could axis be something like functional vulnerability (standardized coefficient)

(Remarks on code availability)

Referee #2

(Remarks to the Author)

This is a very interesting manuscript that aims to understand the degree to which land use change affects the functional structure of avian communities worldwide. The central question is fundamental for conservation, the use of a complete trait database including all bird species is very compelling, as is the methodologically robust approach in which several aspects of functional structure and of its resistance to species losses are evaluated. The results are clear, and they are discussed in a convincing way (but note that I am not an expert in avian biology or ecology). In the methodological front, I would like to congratulate the authors for devising the functional vulnerability index, which I found very interesting; in particular, the estimation of redundancy at the species level looks like a very nice idea. Overall, I think that the manuscript is likely to be influential.

However, I have identified a series of potential methodological issues that I explain below and would like the authors to clarify.

 1. My main concern is related to the potential effects of species richness being the underlying driver of some (perhaps all?) functional patterns. Tables S3 and S2 strongly point this way, with redundancy being only slightly better than species richness as a predictor of resistance. All the functional diversity indices that the authors have chosen are related to a higher or lower degree (in general, I would say rather high) to species richness. The authors are aware of this, and rightly state that using null models to get corrected metrics that are not affected by species richness is not an ideal solution. I am however skeptical that including the study structure (study and study block) as random effects is enough to solve this issue. The authors state that "we include study and study block as random effects to ensure that functional metrics are exclusively

compared among nearby assemblages within the same study”, but this doesn’t rule out that differences in functional structure among sites in the same study/study block are not due exclusively to differences in species richness: there are other factors beyond the regional species pool and sampling effort that are likely to affect species richness. In short, within the same study, an agricultural site generally will have lower functional richness than a primary vegetation site simply (or at least partly) because it has lower species richness, regardless of the traits of the species present in either site. A further complication is that the effect of species richness on the different functional metrics is likely to be non-linear, so that adding or removing a species in a species-poor assemblage is likely to have a larger impact on its functional structure than in a species-rich assemblage. Unfortunately, I am not sure about what the best solution would be here, but perhaps some complimentary analyses using null models would help (maybe restricting the randomizations to the pool of species within each study?). I would also estimate the same kind of mixed models for species richness, to show that the functional patterns are somewhat different than the results including only species richness.

In the same line, regarding the correlation between redundancy and resistance, redundancy and resistance to extinction should be mathematically related, through their shared relationship with species richness. Because of this, I fail to see how this result is not somewhat trivial. I think that a more useful estimation of resistance would be one that compares the expected trajectory of FD with species losses (trait-based or rarity-based) with a set of random trajectories for the same community (i.e. when species are lost in a random order). This way perhaps you could get a more robust indicator of FV. For example, one possibility would be to estimate worst-case and best-case extinction scenarios in which species losses are ordered following the species redundancy ranking (worst case: first species that are lost are the less redundant ones; the opposite for the best case. Perhaps you could recalculate species redundancy after each extinction to have more realistic worst and best cases). Then you can estimate FV_worst, FV_best and compare your observed FV value with them.

 2. I had problems understanding what has been done in the TPD analyses (lines 414-427 in the main text). It is probably my fault, but the "reduced into smaller site-level TPD subsets" is a bit confusing to me. I was also worried about what is written in lines 423-427: Four species (the number of dimensions plus one) would be needed to estimate the bandwidth, but not to estimate the TPD function of a site if you have already estimated the TPD of individual species. This looked worrisome to me, because if you are estimating a TPDs function for each site, then it is likely that you are using a different bandwidth estimation for each site. However, after superficially checking the script ("1. Simulations_kde_abundance_all.R"), it seems as if you have done the calculations correctly. But perhaps you should reconsider how these parts are written? Nevertheless, I think your choice of bandwidth estimation is not optimal, since you have considered the whole pool of species in the study (I think; Line 100 in the script). This would be the optimal approach if you were ultimately interested in estimating the TPD function for the full set of species (as in a "Global spectrum" approach). However, here you are interested in estimating the TPD functions of individual assemblages. Using all species at once is likely to result in bandwidths that are too narrow for the scale of interest (the individual sites), so that the TPD functions of the sites will be too "peaky", the overlap between species will be too small, and things like functional richness and functional redundancy will tend to mimic species richness much more strongly. In general, the approach to estimate TPD functions that makes sense to me would be the following:

--Step 1: Choose a common bandwidth for all species. You have done this using the Hpi.diag function, but applying it to the whole set of species, which is not optimal. I think the best way to solve this issue is to use Hpi.diag to estimate a bandwidth for each site (i.e. 1,281 bandwidths), then estimate an average of those bandwidths.

--Step 2: Combining the (square root) of that average bandwidth and the mean position of each species in the three-dimensional space, estimate a single TPD function for EACH INDIVIDUAL SPECIES, using the TPDsMean function (by the way, the function's name is misspelled in the main text); you have done this in L117 of the script, and it seems correct.

--Step 3: For each community (site), aggregate the (abundance-weighted) TPD functions of the the species that are present. This can be done in the TPD package with the TPDc function; you have done this in Line 189, and it also seems correct.

Other minor comments:

-Lines 126-127: "we calculated functional redundancy for each assemblage as the amount of shared niche overlap between co-occurring species." I think this is not the best definition of functional redundancy under the TPD framework (see Box 2 in Carmona et al. 2016 TREE). FR does not only consider what proportion of functional space is occupied by multiple species, but also by HOW MANY SPECIES. This detail is important, since an assemblage that includes five species occupying the space in the same way should have higher redundancy than one including two species. Your description of redundancy here (and also in ED, Fig. 4) does not seem to account for this aspect. Rather, I would say something like "we calculated functional redundancy for each assemblage as the average number of species (across all parts of the functional space occupied by the assemblage) that could be removed without reducing the functional volume occupied." Just a proposal; I am sure there is more elegant way and shorter way to write this.

Carlos P. Carmona

(Remarks on code availability)

I have reviewed parts of the code, which is very extensive. I haven't ran it, but I have been able to understand most of what I have seen. I think it will be a very useful resource for the community. My only concern is that some of the scripts use code from other authors, adapted in some cases, and I think this should be acknowledged in some way (either in the header of the script, or in the manuscript itself)

Referee #3

(Remarks to the Author)

Thank you for giving the opportunity to review "Land use change undermines the stability of avian functional diversity" for

consideration in Nature. There are many things to love about this manuscript. The ideas are original, the dataset is large and thus adequate to address the questions, the writing is quite clear, the findings are very interesting, and the topic would be of interest to a broad swath of scientists and practitioners. I particularly like the alternative hypotheses that the authors pose for why disturbed land uses may be more (or less!) resilient to future species loss. I haven't seen these hypotheses before and was really excited by the way the authors chose to address this topic. The finding that disturbed land uses lack functional redundancy and thus may be even more vulnerable to future biodiversity loss is an important and novel contribution to the ecological literature.

That said, I do have a number of concerns with this manuscript, articulated in depth below. Though I bring up a number of issues, my biggest concern centers around the way that the authors simulate future extinctions to evaluate the relative importance of functional redundancy versus functional vulnerability. In short (and as I articulate below), I do not believe that this approach allows for the mechanism underpinning their second hypothesis to operate and thus think that their core finding from that effort is an artifact of the way their analysis was conducted.

As mentioned, I have articulated all my thoughts below in a point-by-point manner. Again, thanks for giving me the opportunity to review this manuscript. I really enjoyed it and learned a lot from the authors' great work.

Introduction

- Lines 66-70: I'm curious why the authors do not explicitly reference the Etard et al 2022 paper that they cite in other places, both here and more explicitly in reference to their findings. This paper uses the same dataset (PREDICTS) but different traits and a broader array of taxa to look at the effects of land use on functional diversity. Unlike the statement here (line 66-70), Etard et al 2022 do find strong declines in FD with land use, a finding that is echoed in Fig 2b. I would explicitly reference this paper when you talk about your findings regarding functional diversity (Lines 144-122), saying that your work aligns with prior analyses of the same dataset.
- Lines 106-110: Here and throughout the authors suggest that morphological diversity has well established links to ecological processes and functions in birds. As a result, the authors make strong statements about ecosystem functions and processes; for example, at the end of the abstract: "Our analyses reveal that land-use change may have major undetected impacts on the stability of key ecological functions, hindering the capacity of natural ecosystems to absorb further declines in functionality caused by ongoing perturbations." The evidence they cite to connect morphological diversity to ecosystem functions/processes is often Pigot et al 2020. The Pigot paper connects morphology quite strongly to other functional traits (i.e., trophic position, diet, etc.) and therefore morphology does seem to be a good surrogate for species' ecological roles in ecosystems. However, I do not see evidence in that paper or others cited here that functional diversity has been linked to actual ecosystem processes/functions in birds. Consider insect predation. One meta-analysis showed that a variety of functional diversity measures correlate with insect predation in tropical agroforests. However, no FD measure performed better than species richness and, ultimately, the authors concluded that sampling effects are likely the driver, with a few key bird species providing the bulk of pest removal services (Philpott et al 2009 Ecological Applications). All this is to say that I do not believe the authors have established a firm connection between functional diversity and actual ecological processes in birds. That doesn't mean such a connection is absent- it would make intuitive sense for it occur. I would just recommend that the authors be more circumspect about this in the writing. I would also include a statement in a limitations section saying that the connection between functional diversity and actual ecosystem functions/services in birds has yet to be firmly established (that is, of course, unless the authors can cite papers firmly linking functional diversity/morphological diversity to measurements of ecosystem functions/processes).
- I am not sure that the conceptual diagram (i.e., Fig 1) is that helpful. There is a lot going on in this figure and it takes much more time to try to understand the figure than it does to understand the core concepts being communicated. For example, for panels b and c, bars need to be compared between the far left and far right sides of the graph to determine how land use transformations effect functional diversity/redundance. The reader intuitively wants to compare adjacent bars, but these are FD vs. FR and thus not intended to be compared. If the authors would like to keep a conceptual figure, I would consider simplifying it to only visually depict the two alternative hypotheses about how land use affects functional stability.
- Lines 110-112: Please clearly define what an assemblage is upon first mention. Is it a list of species present at a given site? From one survey? Multiple surveys? Can multiple sites be aggregated (e.g., two point counts in the same area) or is it just one location?

Results and Discussion

- Fig 2: I would recommend running some post-hoc tests to identify differences among land-use categories and then denote significance between them with letters in the graphs. As is, the graph clearly shows differences relative to a 'primary vegetation' baseline. However, many readers might also be interested in knowing, for example, if there are significant differences between minimally and intensively used urban areas.
- Lines 125-140: I appreciate the author's explanations in the extended data for why they did not choose to use species-richness corrected measures of FD/FR. I agree that doing so would mask important effects on ecosystem function. That said, I'm still left feeling really curious about how doing so would affect the patterns in FD/FR across land uses. Were the most functionally unique species lost first from land-use change, causing declines in FD after accounting for changes in richness? Is it that the least unique species were lost first, causing a decline in FR but not FD after accounting for richness? Is it more of a random pattern of loss, such that FD and FR do not change after accounting for changing in richness? I'd suggest the authors include such an analysis as an addition to the manuscript, not to make statements about ecological functioning, but rather to make some statements about the pattern of functional diversity/redundancy loss with land-use change.
- Lines 141-152: I wanted to bring up a potential issue with dividing birds into generalists, granivores, insectivores, and frugivores guilds. Are you not eliminating some key functional redundancy inherently by relegating birds into the generalist category? For insect predation, both generalists and specialist insectivores eat insects. They may do so in functionally redundant (or distinct) ways. So why would you not consider the generalists as well if you are interested in the particular

ecosystem function of insectivory? The same statement could be made for seed dispersal and seed predation. Instead, to analyze redundancy/diversity in insectivory (for example), I'd subset to the fraction of community that eats insects (you could impose some sort of threshold; say, 25% or more of their diets is insects). Then conduct the analyses. The generalists would appear in multiple analyses but that's ok- they would still be there to provide (or not provide) the function. Also, the patterns exhibited here are entirely consistent with the observation that you often see strong declines in the richness of insectivores/frugivores but not generalists/granivores. Again, I'd be curious if the most functionally unique/redundant birds are lost first (a question that could be answered with richness-adjusted measures).

- Lines 169-171: I have a few suggestions about additional analyses that could be done to robustly examine patterns of subsequent species loss.
 - o First, it would be nice to do a little bit of sensitivity analyses surrounding the response traits you are choosing. As the authors are well aware from their own work (i.e., Hatfield et al 2017 Eco Apps), response traits can be especially fickle in their ability to predict species responses to disturbances across databases. It would thus be nice to see that the patterns exhibited here are robust to different choices about which traits they include. For example, beyond the four traits examined here, Etard and Newbold 2023 outline a few others that showed predictive ability in dictating species responses to land-use change (e.g., habitat breadth and specialization on natural habitats).
 - o Second, I might also be explicit what sort of 'future disturbances' you are considering that could induce future species losses. As is, the disturbances are really vague and could represent anything. I see the value in that (we don't know what the next disturbance will be) but I also think an additional analysis of a more concrete stressor that could cause further extirpations would be interesting. One possibility could be climate change. You could use some traits that have been shown to be at least moderately predictive of species abilities to weather climate change and implement those as the ones to determine species sensitivity. Some such traits are present in Pacifici et al 2017 Nature Climate Change, for example.
- Lines 176-178: Just to make it extra clear to the reader, I'd add a clause saying that a decline in Functional Vulnerability with land use is, essentially, a good thing. That is, land use is causing the assemblages to be less functionally vulnerable. It took me a couple reads to get that.
- Lines 178-180: While the two metrics do produce broadly similar trends, there are some key differences. The response trait version shows a much more confident decline in FV for intensive urban and plantation sites, whereas the rarity-based one shows stronger trends for pasture and cropland. I'd point out this difference to be upfront with the reader.
- Lines 183-185: Could you rephrase this sentence? I'm not really understanding what it is getting at.
- Lines 202-203: Is there a reason you show a graph for the redundancy predictor (Extended Data 4) but not for the FV predictor?
- Lines 213-221: As I alluded to above, I am skeptical that the future anthropogenic extinction analyses is set up correctly to allow the authors to arrive at the conclusion that the functional redundancy hypothesis (i.e., that intensive land uses will be less resilient because they have lower redundancy) is more supported than the functional vulnerability hypotheses (i.e., that intensive land uses will be more resilient because they already lost their extinction prone species). The way the authors compare these hypotheses is by sequencing deleting species from the community, ordered by response traits or population size (rarity). They then compare the extinction curves between different land use types, using both areas under the curve as well as half-lives of the extinction. Unless I am totally off-base, this approach bypasses the author's second hypothesis entirely. This is because the authors are forcing the species to be sequentially extirpated. The order is indeed determined by their vulnerability, but all species are eventually extirpated to generate the curve. There is thus no way that the intensive land uses can hold the species that remain (and thus their FD) for a longer time than the pristine habitats. As a result, the exercise is only allowing the functional redundancy hypothesis to operate. Off the top of my head, an alternative approach would be to probabilistically remove species according to their rarity or traits over a series of time of steps. Basically, each species could have an extinction probability that is proportional to its response traits or its population size. Then, through a series of sequential random draws applied to each species in the community (i.e., a series of time steps), communities in each land use could lose species, with rare species having a higher likelihood of extirpation than common species. FD could be assessed at each time step or at the end of the trial. This approach would allow for the possibility that intensive land uses are less likely to lose species because they have already filtered out the vulnerable ones (i.e., the authors' second hypotheses). The current approach, however, does not allow for this possibility; therefore, I do not believe the authors can conclude that intensive land uses are indeed less resilient to future functional diversity loss.
- Fig 3c (and figure 2): This is just my preference but I think these graphs would be a lot easier to understand if the y-axis graphed the actual metrics (i.e., FD, FR, FV, or AUC from the extinction trial) rather than the coefficient estimate with primary forest set as a baseline. Especially for Fig. 3, I had a hard time trying to understand what exactly was being graphed (before spending a long time reading the rest of the paper and the extended data). If figure 3c was clearly indicated as AUC, and you could see the curves above, then it would make more sense what was going on. I'd also consider including extended data 3 in figure 3 as this really helped me understand what the AUC was doing and why it was included.
- Lines 909-910: I would add, as a clause, that the actual analysis uses proportional removal (not raw species) so that the starting species richness does not drive the results. Indeed, the AUC for intensive sites would be lower if proportional removal did not occur, simply because there are fewer species to start with.
- Lines 226-229: As above, I would reconsider separating generalists from insectivores, for example, if you are looking at the robustness of the ecological process of insect removal (see rationale above).
- Lines 255-261: Again, I do not think you can make these statements based on the extinction simulation (see rationale above). Indeed, these statements are pervasive in the discussion (e.g., lines 267-271).
- Lines 262-264: I may have missed this, but where do you show that the species in pristine habitats are typically (and disproportionately) functionally unique? Wouldn't you need to do the richness-corrected analysis to show this?
- A general comment for the discussion. I would appreciate a brief paragraph on limitations. For example, I would mention the relative tenuous connection between FD and ecological processes provided by birds (as mentioned below). I would also discuss issues of detection (as outlined below).

- Lines 297-299: What version of the PREDICTs dataset did you use? This seems to indicate the original version that was published in 2014. Is there a more recent version with more data? I had thought so but maybe I am wrong.
- Lines 318-321: I think this discussion of temporal or geographical blocks could be written a bit more clearly. It makes sense in the extended data but is a bit hard to understand here. Could you rephrase, perhaps adding another sentence for clarification?
- Lines 322: There's a grammatical issue in this sentence (which there?)
- Lines 325-328: How did you determine what was close enough together to be aggregated into one assemblage? After aggregating points, how would you then account for variable effort across sites (i.e., some sites represent an aggregation of multiple surveys and others do not)?
- Lines 363-368: I understand the rationale for using PCA to collapse the trophic traits and locomotory traits, as well as the rationale for using the second PC axis (which didn't correlate with body size). However, why bother conducting another PCA on the first axes to get a body size axis? Why not use body size directly at that point? This would much more directly measure body size that a double PCA...
- Lines 373-388: I am a little confused about how the diet data were used. Did you collapse the diet data into a single PC axis like the morphology data? Were all variables without collapsing to generate the distance matrixes?
- Lines 384-388: How are the primary food type data used? Is it just to divide up the communities or are you actually using this information to calculate functional diversity as well? It would be good to clarify this here.
- Lines 390-420: The authors are far better experts about functional diversity calculations (and their nuances) than I am, so I am not going to review the technical dimensions of how functional diversity is calculated (I'll leave that to another reviewer). That said, it is widely known that functional diversity outcomes are quite sensitive to the types of traits included. I understand (and agree with) the rationale for focusing on morphological diversity and diet. However, I do think it would be nice if the authors did some sensitivity analyses to see if all their analyses are robust to choosing additional traits. For example, Etard et al (2021) used PREDICTS (i.e., the same database) and analyzed FD using body mass, trophic level, lifespan, litter/clutch size, diel activity, habitat breadth, and use of artificial habitats as traits. I would thus encourage the authors to include a few other traits along the lines of other studies (like Etard et al) and then see if their findings are robust to trait decisions.
- Lines 398-399: Again, I am not an expert on these metrics but I was taken aback a little by the description of functional redundancy, which is measured as "the proportion of this volume that is shared by multiple species." Consider two communities with identical fractions of the volume shared by more than 1 species, but in community A that volume is shared by 2 species and in community B that volume is shared by 10 species. According to the definition above, it would seem that both receive the same functional redundancy score even though community B clearly has more redundancy. Can you clarify this a bit?
- Lines 395-399: Are species abundances used in any way to calculate functional diversity or redundancy? From my read, it appears now. I wonder, however, if there might be a way to consider abundances as an even distribution of abundances across traits is probably much better for functioning than another community with equal trait diversity but that is dominated by a couple species.
- Lines 407-417: The approach for generating intraspecific diversity via a kernel probability density was particularly challenging for me to understand. Where did the data for estimating intraspecific variation come from?
- Lines 423-425: I am a little confused on how the functional diversity/redundancy metrics can be re-run on diet guilds if diet is also one of the key dimensions of in calculating functional diversity. Is it just that you are looking for variability in diet within the diet class? In this case, are less specialized insectivores, for example, more functionally diverse? If evaluating the potential impact of functional diversity within insectivores on insectivory, why should having a more diverse diet within the community contribute to a higher rate of insect predation if diet diversity isn't measuring the diversity of insects in a bird's diet but rather non-insect diet items. In short, I'm not sure the diet trait should be included in functional diversity/redundancy analyses of specific diet guilds.
- Lines 428-433: Is this the same sensitivity analyses that was just referenced above in lines 417-420?
- Lines 483-488: I am trying to wrap my head around the idea of standardizing FD to start at 1. Shouldn't functional diversity be on the same scale for all sites in the same study and wouldn't it thus make more sense to consider the absolute reduction of FD as a function of species loss? Imagine losing Species A causes a FD reduction of 1. In an intact community with an initial score of 10, losing Species A would represent a 10% decline (from 10 to 9). In a disturbed community with an initial score of 5, losing Species would represent a 20% decline (from 5 to 4). But I would argue the effect on ecosystem function should be equivalent in both communities. Therefore, I wonder whether it makes sense to standardize in this case (unless I am missing something).
- Statistical analysis: I identified a number of potential issues with the statistical analyses.
 - o First, the issue of imperfect detection was never raised in this manuscript. Bird detectability is known to vary a lot across different land uses, which can heavily bias findings about the impact of land use on individual bird species and diversity metrics. For that reason, many studies implement models that account for detection directly (i.e., occupancy models, distance sampling, etc.). I understand this isn't possible using the PREDICTs dataset (or really any other meta-analytic effort). Nonetheless, this is a major limitation of global analyses like this one and should be clearly articulated as a limitation in the main text and discussed as such.
 - o Second, whereas the calculations of FD, FR, and FV are very nuanced, the models used to analyze them are very simplistic, with land use as the only predictor and study/study block included as random effects. As mentioned above, I do not believe that it would be possible to formally model detection. However, could the authors not at least include some study-specific covariates known to influence bird detectability (for example, day of year, time of day, or any other readily available covariate known to influence detectability)?
 - o The authors provide no discussion of model fit. Do all the models satisfy model assumptions about normality, heteroskedasticity, and other considerations?
- Extended data: There is a lot of emphasis in the extended data focused on tropical/temperate comparisons but very little discussion of any of those findings in the main text. I wonder if it would be better to either include a little discussion of this in the main text or not provide all those analyses if they are not of interest.

(Remarks on code availability)

Version 1:

Reviewer comments:

Referee #1

(Remarks to the Author)

I thank the authors for their thoughtful and considered response and excellent paper – which presents strong results, including the finding that land use leads to species being spread thinly over trait space, making the assemblage functionally vulnerable to future change, and analytical methods which I expect to be influential.

The requested toning down of results relating to ecosystem functioning outcomes and foundational evidence for which traits respond to land use has been performed and presented, respectively, and I have no new major comments or suggestions.

Just a comment on the response really but I do feel that it's hard to say whether the predicted trends would really affect functioning, but wording is generally toned down to reflect this and caveats are given. Ecosystem functions tend to be strongly controlled by both the diversity and abundance of interacting species (not just the traits of a single group), as well as physiological rates/activity and, and there are of course organisms other than birds that provide very large proportions of these functions too, so it cannot be said with confidence that all these changes will result in an observable change in whole system functioning. While FD etc often correlates with function and sig relations can be found its predictive capacity is often weak- other factors are also at play. As I say, though, this is just my reflection on the authors response, and I feel the paper does not overstretch the conclusions it draws from the results in its current form.

I just had a few very small extra points:

52- rich>biodiverse (or just remove this word)

252- not sure what is meant by diversity of functions here when we are talking about individual functions- consider rephrasing

276-277 not sure if the 'demand' phrasing really works here as we have demand for services not functions and the link to this service aspect is not really covered at all in the paper – maybe simpler wording would be better here, e.g. “some of the FD lost under land use may be related to interacting species that are now absent and so no longer require their associated functions”

292- potentially leaving (the link to function cannot be assumed)

294 – FV (or write in full throughout, which might be better for general readers)

1124- where the positive effects of land use on granivores explained in the main text? Its an interesting result. I feel this is an important figure- would be great if there is space for it in the main paper

I did not review the methodology in detail this time round but find the analyses to be strong and well explained and justified. Maybe a little work is now needed to make the paper generally accessible, given complex responses to reviewer comments, but I will trust the editors to deal with this.

Thanks for the great work, which was a pleasure to read and review.

(Remarks on code availability)

Referee #2

(Remarks to the Author)

I have enjoyed reading this new version of the manuscript. It is very well written, and all the concerns that I had in the previous version have either been corrected or explained in a convincing way. The responses from the authors are comprehensive and precise; they have modified the procedure for bandwidth estimation (my main concern), supplied robustness tests on species richness standardisation, added the requested extinction simulation, and clarified the definition of redundancy. All my earlier methodological questions have therefore been fully resolved, and all clarifications I suggested now appear either in the main text or the Supplementary Information.

The revised paper is clear, methodologically sound and timely. It delivers important evidence that land use change erodes functional redundancy and resistance in bird assemblages worldwide, with direct relevance for conservation planning. I have no further requests for analysis or wording changes. I would like to congratulate the authors for an excellent paper.

(Remarks on code availability)

I tried to open the link provided but it didn't work. I was satisfied with the script in the previous version, and explanations from authors in the response letter suggest that my previous comments on code have been taken into account, so I have no further requests.

Referee #4

(Remarks to the Author)

I did not review the previous version of this manuscript by Weeks and colleagues, but was invited to review the revised version by a Nature editor, with a request that I review closely the original critiques and subsequent author responses to Reviewer 3. I have thus read the entire revised manuscript, as well as the response document.

In general, I agree with previous reviewers and commend the authors on a broad, interesting manuscript. Moreover, I feel that the authors have sincerely worked quite extensively to revise this manuscript in alignment with all three reviewers' demands as well as to defend their original choices where appropriate. While I have not seen the originally submitted manuscript, I can see (through the response) the many ways that the manuscript has been improved. To that end, I delightedly experienced multiple times the feeling of "Huh, I wonder if they considered issue X?" only to discover, "Oh, indeed! There's an entire sensitivity analysis on it." So thanks to the authors for making my job more pleasant.

As you might imagine from this preamble, my comments are mostly minimal. My overall assessment is that this is a thorough and well accomplished revision. They are, however, summarized below:

1. Although I have not seen the original, I can tell that Figure 1 has been improved in clarity; but does it give the wrong expectation? It seems to suggest that the study looks at how 'pristine' and modified communities respond to further degradation – i.e., the arrows from a/d to b/e/g imply processes that are being observed in this study. However, that is not empirically measured by the study; rather, this study predicts the disturbance process simply through extinction simulations. While the simulations are well considered (and more robust post-revision), simulations are not the same thing as empirical data. Since Figure 1 is largely a schematic for the study (hypotheses and predictions), is there a way to make it clear that the disturbance process is being simulated here, and not observed directly through time series?

2. For Figure 2, the study size bubbles imply the smallest number of assemblages per site is 50, when undoubtedly it is much smaller. I would add a size bubble that corresponds to the smallest unit on the map. It is typical for figure legends to show the full range of an important measure.

3. I appreciated Reviewer 3's concern about the role of imperfect detection in potentially biasing results and the dataset. I read the manuscript first, before the review, and when I got to the "Caveats and limitation" section I read it with careful interest, but was generally left unfulfilled. As written, the section on caveats tonally feels like a perfunctory section added to appease reviewers – it doesn't seem like the authors believe that this work is deserving of any 'caveats or limitations' at all! For example, most of the paragraph is taken up not with exploring potential limitations, but defending against them. Not only is this a waste of space, but it can be read as insulting to readers. For example, it is truly a limitation that the inferential strategy is a "space-for-time" comparison (pristine vs. modified) combined with a pure simulation of temporal processes (i.e., further disturbance); there is an extensive literature in ecology showcasing the limitations of space-for-time approaches and equally extensive examples showing how simulations fail to capture observed phenomena. Yet, the authors seem to dismiss these real limitations by touting the analysis as "probably the best space-for-time comparison available", which honestly made me laugh (not to be too mean, but it sounds like the things certain world leaders say). Similarly, the dismissal of potential concerns about detectability biases (which have been shown repeatedly to bias all sorts of diversity metrics, including functional diversity metrics) by asserting that "bird species are easily identifiable" and can be detected by "acoustic signals" almost absurdly misses the point that nearly all methodological studies documenting the extent of bias from imperfect detection have been based on field surveys of these same "easily identifiable" birds. Ultimately, I agree with the authors (on detectability) that the direction of bias would be such that they are likely underestimating the true effect. However, I do feel that this 'caveats and limitations' section would benefit from an increased dose of honest self-critique.

Additional Small Line Edits:

63. I get that there is too much here to be summarized, necessitating supplementary information. However, it seems a shame that none of the researchers who have done this valuable pioneering work will receive citation credit (since supplements aren't indexed). Perhaps cite 1-2 of the most important, most relevant studies, and then refer to the supplement?

131. With such a high p-value, can anything be concluded at all about FD in mature secondary vegetation? It seems inappropriate to conclude anything about this result.

Code availability: I was given a GitHub repository URL. However, the URL did not work for me so I was not able to review the code.

(Remarks on code availability)

The code, as cited above, was not available to me. The URL gave me an error. Moreover, GitHub is not a suitable archive for

code, as it can be changed post-publication. Code for published papers should have permanent archives.

Referee #1 (Remarks to the Author):

COMMENT: This study combines two large datasets AVONET, for bird traits and the land use biodiversity impacts dataset PREDICTS to great effect, and shows that land use change causes major shifts to the functional diversity and composition of bird assemblages. It also shows, via some rather advanced statistical analyses and simulations, that these changes could affect the stability of the functions that these communities provide.

The paper is impressive and the results are important, but as currently presented the emphasis is a little heavily statistical in presentation with not too much of the underlying biology discussed. I agree with the tight logical flow of arguments presented in the paper, these nicely underpin the analysis choices taken, but this does not prevent the paper from being based upon a series of contingent assumptions where fairly abstract metrics are related to each other e.g. that a simulated extinction will lead to a certain loss of trait space that might in turn affect function. This criticism does not mean that I am unsupportive of the paper but I do think it would benefit from two main things.

The first would be a more simple and fundamental description of which traits are changing in response to land use and why. This simple result – a description of the general trait changes imposed on bird communities worldwide by land use would be interesting in its own right, and enough to merit publication in a high impact journal, given the generality of the results. Further, it would make a good basic foundation for the more advanced analysis that could then follow. By going back to the basic natural history of these birds a bit it would also make the work less abstract and grounded in ecological reality.

RESPONSE: We thank the reviewer for their positive remarks.

Regarding their first suggestion, we don't think this study should focus in detail on the response of bird species traits to land-use change because a number of studies have previously investigated this question (e.g. Burivalova et al. 2015; Fenzel et al., 2016; Guerrero et al., 2024), many of them using trait data from AVONET (Bregman et al., 2016; Neate-Clegg et al., 2023; Weeks et al., 2023) and survey data from PREDICTS (Newbold et al. 2013; Etard et al. 2022; Etard & Newbold 2023). Thus, the descriptions requested by the reviewer have largely been established by previous research, providing a strong foundation for the current study.

While we therefore feel that it is unnecessary for this paper to revisit these findings, we take the point that some readers would appreciate this manuscript being less abstract.

To help ground our study, we now include a new analysis in the supplementary material showing how key species traits vary with land-use category (Extended Data Fig. 1). This shows, for example, that species inhabiting intensively modified landscapes are associated with higher dispersal ability, wider niche breadth, and larger range size. We also show the general connection with body and beak shape metrics used in other analyses, linked to changes in diets from insectivory to granivory, and from arboreal/aerial lifestyles to more generalist/terrestrial lifestyles. We now report these general findings in the introduction to our study (lines 125–127) and we hope this helps to place our study more firmly in the context of basic natural history.

COMMENT: The second would be a less certain wording when interpreting the results of the more advanced and assumption-rich analyses. While the logical flow of arguments is strong the paper does rely upon a series of complex stacked assumptions that requires a few 'leaps of faith by the reader'. The link between trait measures and real ecosystem level functioning measures is not clear or well established. It is a bit of a jump to go from a beak length distribution to an actual function such as the rate of invertebrate consumption or frequency of seed dispersal. How big an impact that these changes in functional composition would have on ecosystem functioning therefore semi-speculative and so wording should be less certain to reflect this. e.g.- 'stability likely reduced'.

RESPONSE:

The reviewer makes a reasonable point that the link between functional trait diversity and direct effects on function is not easily established at broad scales. Nonetheless, the connection between trophic traits and function makes intuitive sense (as noted by all reviewers) and is supported by a growing number of studies showing a link between trait diversity and ecological processes driven by plants (Hao et al., 2020; Huang et al., 2019; Looney et al., 2023; Ye et al., 2019) and animals (Dekanova et al., 2023; Gagic et al., 2015; Woodcock et al., 2019; Mosman et al., 2023). The link has also been shown through experimental studies in animal systems (e.g. Lefcheck & Duffy, 2015). Several previous analyses also confirm direct links between bird functional diversity and ecosystem-level functions (e.g. Saavedra et al., 2014, Barbaro et al., 2014, Barbaro et al., 2016, Martinez-Salinas et al., 2016). We have edited the text to strengthen this rationale (e.g. paragraph on lines 59-75), including a new section in the SI titled "Functional traits predict ecological processes" covering this topic in depth. We also follow the reviewer's advice by toning down the wording in some places (see below).

COMMENT:

Another key concern I have considering the true biological meaning of the results is whether the lost functioning of the community is still needed to maintain ecosystem

functions in the new environment? For example, does the loss of a fruit eating seed disperser matter if the new system is a city or monoculture that contains no naturally planted fruit producing trees? It would be nice to see some discussion of this. The paper is generally missing this type of biological discussion, particularly at the function level. Adding more information and insight in this area would make the paper more accessible to a wide readership, and thus increase its impact.

RESPONSE:

This is a common point made about changes in function after land-use change (i.e. maybe the functions are no longer needed anyway so what's the problem?).

Our first response is that this critique is largely irrelevant to our main analyses and conclusions. In these, we show that FD is not simply reduced after land-use change, but also “thinner”, with reduced redundancy. That means that the FD of human-modified assemblages will be vulnerable to any further species loss because there is limited back-up for each ecological role. In other words, we show a thinly spread functional space in modified systems, suggesting that **the demand for the function remains** but the biodiversity supplying this demand has low redundancy, leaving that function exposed to species losses through future disturbance.

We also think that the reviewer understates the need for FD in modified environments for three reasons. First, insects (including pests and disease vectors) as well as fruit-bearing plants (including many shrubs and trees) still exist in agricultural and urban environments – so insectivores and seed-dispersers are not superfluous and instead continue to play an important role.

Second, many animal species are sensitive to human pressures, and their local extinction often precedes the removal of their food, meaning that there is an unmet demand for the ecological service they provide. In birds, this happens when nectarivores (e.g. Anderson et al. 2011) or frugivores (e.g. Galetti et al. 2013; Terborgh et al. 2008; Sethi & Howe 2009) are extirpated before their food plants, leaving an unmet demand for pollination and seed dispersal, respectively.

Third, functional diversity is needed to maintain the potential for restoration (as this reviewer notes in a later comment, below). For example, a cleared forest with a larger diversity of surviving seed-dispersers has a higher potential for recovery (= resilience). Thus, while it makes sense that FD should decline where there is less demand for seed-dispersal or insect predation, any such decline can be accompanied by reduced resilience of the system.

Similar points are widespread in literature (e.g. Bregman et al. 2016; Hatfield et al. 2023). Nonetheless, we have followed the reviewer's advice and included some further discussion of these points (Lines 276-281), including a new section titled "Demand for ecological functions after land-use change" in the Supplementary materials. We hope that including this information makes the paper more accessible to general readers.

COMMENT: Finally, while the intro displays excellent grasp of and use of ecological theory, and does a good job of navigating a series of complex concepts in a few lines, it may benefit from adding the distinction of effect and response traits, as it's a somewhat key concept. E.g. in explaining why redundant species at the effects level can still stabilize function due to differences in response traits. While I understand the necessity of keeping things simple in a paper of this kind there are a few places where adding this distinction would make the arguments flow more logically. The concept does come in later (e.g. line 170, so maybe good to set it up from the onset).

RESPONSE:

We previously side-stepped the response and effect trait framework as the topic is quite specialised and hard to explain in an article for a general readership. However, we can see that drawing a distinction between response and effect traits helps to solidify our arguments in a few parts of the text and also in some responses to reviewer comments below, so we have now introduced them more clearly near the beginning of the manuscript (Lines 108-117).

COMMENT:

Overall, this is an accomplished study, that is well written, well thought through, and important. A few changes would make it more plausible, grounded and accessible. I feel these comments should be addressable and hope these and the more specific ones below are useful in revising the paper.

RESPONSE:

We are very grateful to the reviewer for this positive appraisal and their insightful comments, which have been useful in improving the clarity of the text.

COMMENT:

33- resistant or resilient? (or disturbance tolerant). Given the papers focus on stability these terms from the stability literature need to be used carefully.

RESPONSE: We agree. According to the dictionary, a general definition of resilience is ability "to withstand or recover quickly from" setbacks. So, resistance is clearly the 'first

line of defence' and an integral – perhaps predominant – factor at the heart of what may be conceptualised as 'resilience' in the context of ecological stability.

To clarify our terminology, we now distinguish between Resistance and Recovery, separating these elements as two components of Resilience. This terminology is well established in literature, based on the definitions of Hodgson et al. (2015), and now used widely in ecosystem science (e.g. Capdevila et al., 2021; Nelson et al., 2021; Steel et al., 2021; Webster et al., 2021). We have now highlighted these definitions as part of our conceptual framework (Fig 1) and in the main text (Line 86-90).

COMMENT: 43- likely destabilize

RESPONSE: Changed.

COMMENT:

46- reveal>indicate

RESPONSE: Changed.

COMMENT:

56- Le Provost 2021, Nat Comms may be a more appropriate reference from this project- more taxa, more directly linked to diversity responses

RESPONSE: We appreciate the suggestion and we agree it's a slightly better choice of paper. In this case, we haven't added the reference as it would mean adding a further citation to the already-long reference list (the existing paper works well enough and is used in a few other places in the ms).

COMMENT:

67-69- is this argument essential to the paper?

RESPONSE:

In these lines, we note that some previous papers have shown little or negligible change in FD in response to land-use change. Conclusions such as these can lead to an optimistic view of future assemblage function in response to environmental change. However, these studies only provide a snapshot of current function and do not account for many aspects of functional structure, particularly those related to functional resistance. Therefore, we retain some of this argument to highlight the importance of comprehensive analyses focusing on less-accessible dimensions of FD, including redundancy and resistance.

COMMENT:

84-88- This makes logical sense but how much empirical evidence is there for it, especially when direct measures of function are considered?

RESPONSE:

Numerous empirical studies have shown that functional redundancy maintains the stability of ecological communities in foodwebs (e.g. Sanders et al., 2018) and buffers ecological communities against perturbations (e.g. Pillar et al., 2013; McLean et al., 2019; Nunes et al., 2021). The reviewer is correct that few of these studies consider direct measures of function. However, the assumption that function will be impaired in less-redundant systems follows logically from observations of species loss and disrupted trophic interactions. We now point to more direct empirical evidence from a recent experimental study of ants showing that functional redundancy improves the stability of ecological functions through the expected “compensatory dynamics” which buffer the system against effects of species loss (Yeeles et al., 2025). This isn’t much of a surprise because the relationship between redundancy and the reliable performance of any given system is a general principle (Naeem, 1998). We have added further citations providing empirical and theoretical support for the link between redundancy and stability (Line 86-90) and provided further background in a new section in SI titled “Functional traits predict ecological processes”.

COMMENT:

93- effects groups?

RESPONSE:

We think functional group is more accessible to a wider audience than “effects groups”. We have retained the term functional group in a few places when referring more generally to dietary groupings. In general, however, we focus on “ecological roles” rather than functions, in line with comments elsewhere.

COMMENT:

104- really like this question

RESPONSE:

We are pleased to hear that the core question seems good. To accentuate this strong aspect of the paper, we have simplified our conceptual figure to zoom in more explicitly on this “key conundrum”.

COMMENT:

209- This might be a good place to discuss whether the lost functioning is 'needed' in the remaining ecosystem. Perhaps not in some cases, and this should be acknowledged, but it reduces options for sure- e.g. if future restoration to the original state was required.

RESPONSE:

The reviewer has provided a response to their point above. We agree that this is an important topic for discussion that needs to be tackled early and with greater depth. We now develop these themes in our main text (lines 276-281) and in a new section titled "Demand for ecological functions after land-use change" in the Supplementary materials.

COMMENT:

246- Allan et al 2015 did not study functional diversity

RESPONSE:

Citation removed.

COMMENT:

253- resilient to what? Again, resistance and resilience terms are not clear

RESPONSE:

We now update our definitions of Resilience, Resistance and Recovery based on recent papers (Hodgson et al., 2015; Capdevilla et al., 2021). Given that these are key concepts to our study we have edited our conceptual diagram (Fig. 1) to clarify the terms and also add a paragraph at the beginning of the Supplementary Materials (in a new section titled "Definitions and assumptions").

COMMENT:

276- ecosystem

RESPONSE

Now edited, thanks.

COMMENT:

298- Would be good to give a 1-2 line overview of predicts here to readers unfamiliar with it

RESPONSE:

Some text on PREDICTS now added to the Methods (Line 346-348).

COMMENT: 341- are these all effects traits? the classification of later traits as response suggests so, but it's not entirely clear.

RESPONSE: We include these traits as effect traits. Since this is a key point in our study, we have clarified this point and introduced the idea of response and effect traits early in the ms (Line 86). We have now included a rationale for the use of morphometric traits as effect traits in our supplementary materials (Table S1).

COMMENT:

363- I guess the net result is that all the measures are 'relative to body size'. Would be nice to have a little explanation here

RESPONSE:

We have added clarification that PCA 2 creates trait measures which are independent of body size (Line 423-434).

COMMENT:

367-368- might be nice to say a little more about what functioning these axes relate to

RESPONSE:

Good point. We have added Table S1 to define the traits we use, specify their link to functions, and a column for references that support these claims. We think the inclusion of this information will help readers understand the study.

COMMENT:

410-417- This is a particularly complex and difficult to understand section

RESPONSE:

Agreed. We have thoroughly edit this section and we think the methods relating to TPDs are much improved in the main Methods (whole section titled "Calculating of functional diversity and redundancy) and supplementary materials (whole expanded section also titled "Calculating functional diversity and redundancy").

COMMENT:

421- these are not functions, but community properties that may be related to them- this distinction is not clear in the paper in general

RESPONSE: We have changed the wording here to avoid giving the impression that we are dealing with specific functions, and instead we stick to the safer terrain of

“ecological roles” – we think that general readers will be comfortable with the idea that nectarivores, frugivores, granivores and predators have different ecological roles.

COMMENT:

487- Thus

RESPONSE:

Edited.

COMMENT:

520-523- This seems like cherry picking of data- why does its variability make it invalid, maybe it's just a variable community type?

RESPONSE:

We now retain these data to avoid the impression of cherry-picking. After modifying our analyses in response to reviewer 2, they are no longer outliers in any case. Overall, the new methods for generating trait-probability-densities and functional vulnerability (FV) produce a smoother distribution in our dataset with no outliers.

COMMENT:

535- was pristine vegetation present in all cases?

RESPONSE:

No. The pristine primary vegetation category is present in 50/98 (51%) of our study landscapes and accounts for 177 (13.8%) of all study landscapes analysed in this ms. We have added some text to explain our procedure and to clarify the methods (lines 687-689), including the sentence highlighted by the reviewer (now line 687-693). We also clarify that primary vegetation does not need to be present in all landscapes for the purposes of our linear mixed models (lines 685-687).

COMMENT:

Fig 1. Right side of figure is small and hard to read

RESPONSE:

We have redesigned this figure in line with comments from reviewers and we think it is now much easier to read and interpret.

COMMENT:

Figure 3. (and others) ‘coefficient estimate’ is not an intuitive label – it could be anything, so the reader has to rummage around to find out what they are looking at.

Please present a more intuitive label e.g. extinction-FD slope steepness. It took a few reads to get this figure, if I did at all. Important to get clear as it's the main results figure.

RESPONSE:

We have changed axes labels in this figure and some others to be more intuitive, generally replacing "coefficient estimate" with the explicit name of the response variable, in line with the recommendations of the reviewer. In this figure, we changed "coefficient estimate" to "Functional resistance" and explained this label further in the caption.

COMMENT:

Extended 1. Coefficient estimate could also be more intuitively labelled

RESPONSE:

We have removed the figure headings and added axis labels, which now read: Functional diversity, Functional redundancy and Functional vulnerability. These titles are now used in a standard way across the manuscript to improve clarity. We have also changed the figure format to improve the design.

COMMENT:

Extended 2- again, could axis be something like functional vulnerability (standardized coefficient)

RESPONSE:

We have followed this suggestion here and in all similar figures.

Referee #2 (Remarks to the Author):

COMMENT:

This is a very interesting manuscript that aims to understand the degree to which land use change affects the functional structure of avian communities worldwide. The central question is fundamental for conservation, the use of a complete trait database including all bird species is very compelling, as is the methodologically robust approach in which several aspects of functional structure and of its resistance to species losses are evaluated. The results are clear, and they are discussed in a convincing way (but note that I am not an expert in avian biology or ecology). In the methodological front, I would like to congratulate the authors for devising the functional vulnerability index, which I found very interesting; in particular, the estimation of redundancy at the species level looks like a very nice idea. Overall, I think that the manuscript is likely to be influential. However, I have identified a series of potential methodological issues that I explain below and would like the authors to clarify.

RESPONSE:

We appreciate the positive evaluation – thanks!

COMMENT:

 1. My main concern is related to the potential effects of species richness being the underlying driver of some (perhaps all?) functional patterns. Tables S3 and S2 strongly point this way, with redundancy being only slightly better than species richness as a predictor of resistance. All the functional diversity indices that the authors have chosen are related to a higher or lower degree (in general, I would say rather high) to species richness. The authors are aware of this, and rightly state that using null models to get corrected metrics that are not affected by species richness is not an ideal solution. I am however skeptical that including the study structure (study and study block) as random effects is enough to solve this issue. The authors state that “we include study and study block as random effects to ensure that functional metrics are exclusively compared among nearby assemblages within the same study”, but this doesn’t rule out that differences in functional structure among sites in the same study/study block are not due exclusively to differences in species richness: there are other factors beyond the regional species pool and sampling effort that are likely to affect species richness. In short, within the same study, an agricultural site generally will have lower functional richness than a primary vegetation site simply (or at least partly) because it has lower species richness, regardless of the traits of the species present in either site. A further complication is that the effect of species richness on the different functional metrics is likely to be non-linear, so that adding or removing a species in a species-poor

assemblage is likely to have a larger impact on its functional structure than in a species-rich assemblage.

RESPONSE:

We agree with the reviewer that our functional metrics are highly correlated with SR and that our analyses will therefore tend to show that assemblages with fewer species lose function faster (i.e. steeper extinction curves). This is, of course, expected because species richness is an integral part of ecosystem function with direct links to the number of functions that can be performed in an ecosystem. For this reason, the lower SR in anthropogenic landscapes is a fundamental component of declining ecological function in human-modified landscapes. We have now included some text in the main MS (line 549-557) to emphasise that species richness is an integral part of ecosystem function and to clarify why we use functional metrics unstandardized by species richness. (We note that Reviewer 3 agrees with this decision, see below).

To consider the problems associated with accounting for species richness, it is worth pointing out that FD is often inflated in species-poor habitats. This occurs because the reduced number of species are, on average, more functionally distinct from each than in primary vegetation. Nonetheless, it is generally assumed that this effect is spurious because a few highly distinct species in a human-modified habitat are unlikely to deliver a richer network of functions than a large assemblage of 50 species in primary vegetation. In line with this argument, we prefer to quantify overall function as the diversity of unique traits, estimated as the total volume of functional trait space (i.e. functional richness) without standardising by species richness.

The reviewer is correct that “adding or removing a species in a species-poor assemblage is likely to have a larger impact on its functional structure than in a species-rich assemblage”. However, we note that our functional resistance estimates are calculated using extinction curves that track FD of species remaining in the assemblage as proportions. In other words, they *are* standardised by SR. If one species is removed from a 5-species assemblage, it will remove a large chunk of FD, but the calculation will also move 20% along the x-axis. Meanwhile, if a species is removed from a 100-species assemblage, it will remove very little FD but only move the estimate 1% along the x-axis. Therefore, % loss of FD may differ, but the extinction curve for a species-poor assemblage and a species-rich assemblage should look similar, unless the redundancy per species or the distribution of redundancy across the trait space was substantially different.

To examine our results further, we have now included mixed effects models assessing how SR-standardized functional redundancy (the amount of niche overlap per species).

influences functional resistance (Figure S4). This new analysis indicates that there is a strong correlation between SR-standardized functional redundancy and functional resistance, but that declines in functional resistance are also strongly correlated with reduced overlap in functional trait space per species (that is, increased niche differentiation). This result is instructive because it indicates that reduced resistance in human-modified habitats is driven in part by species richness (as suggested by the review) but also because there is less redundancy to buffer the effects of species loss. We hope these textual edits and new analyses help to establish that the pattern mentioned by the review is an important part of the story and therefore does not need a solution.

COMMENT:

In the same line, regarding the correlation between redundancy and resistance, redundancy and resistance to extinction should be mathematically related, through their shared relationship with species richness. Because of this, I fail to see how this result is not somewhat trivial.

RESPONSE:

We had felt that the result helped to explain how different elements of our study were inter-related but on reflection we can see the reviewer's point that it doesn't add much. We have removed this finding from our ms.

COMMENT:

I think that a more useful estimation of resistance would be one that compares the expected trajectory of FD with species losses (trait-based or rarity-based) with a set of random trajectories for the same community (i.e. when species are lost in a random order). This way perhaps you could get a more robust indicator of FV. For example, one possibility would be to estimate worst-case and best-case extinction scenarios in which species losses are ordered following the species ranking (worst case: first species that are lost are the less redundant ones; the opposite for the best case. Perhaps you could recalculate species redundancy after each extinction to have more realistic worst and best cases). Then you can estimate FV_worst, FV_best and compare your observed FV value with them.

RESPONSE

We thank the reviewer for their appreciation of our functional vulnerability method, we believe that it strives to measure important aspects of functional structure which have been previously overlooked. We agree that comparing worst-case and best-case scenarios to the observed extinction curves could provide a robust assessment of functional vulnerability. However, this is not feasible. We spent many months attempting

to optimize this process which would indeed require the recalculation of species-specific redundancy at every time-step to account for changes to the patterns of redundancy when species are removed. Unfortunately, for some of our larger assemblages this process is extremely computationally heavy. Even with the use of state-of-the art high performance computing, we were unable to successfully derive functional vulnerability estimates for all assemblages using this method.

In this resubmission, we continue to use our original method, which has some advantages over the method suggested by the reviewer. Specifically, it needs far less computational power and therefore can be used as a simple tool to compare functional vulnerability across large assemblages, facilitating future comparisons between studies. Second, it focuses on a simple snapshot (the current assemblage) and does not rely on simulated extinctions to assess the current functional state. This simplifies the model and reduces the number of uncertainties and assumptions.

COMMENT:

 2. I had problems understanding what has been done in the TPD analyses (lines 414-427 in the main text). It is probably my fault, but the "reduced into smaller site-level TPD subsets" is a bit confusing to me.

RESPONSE:

The original wording was our attempt to explain that our FD and redundancy values were calculated at the assemblage level and generated from landscape-level TPDs. This is quite difficult to convey and we agree with the reviewer that we did not do a great job. We have changed our terminology throughout the ms to focus on study landscapes within which we have sampled species assemblages. We have largely removed the confusingly ambiguous term "site". Based on our new standardised definitions, we have re-written this section to explain that we calculated FD and redundancy at the assemblage level using in-built functions that effectively extract assemblage-level metrics from the landscape-level TPD (see line 531-541).

COMMENT:

I was also worried about what is written in lines 423-427: Four species (the number of dimensions plus one) would be needed to estimate the bandwidth, but not to estimate the TPD function of a site if you have already estimated the TPD of individual species. This looked worrisome to me, because if you are estimating a TPDs function for each site, then it is likely that you are using a different bandwidth estimation for each site. However, after superficially checking the script ("1. Simulations_kde_abundance_all.R"), it seems as if you have done the calculations correctly. But perhaps you should reconsider how these parts are written?

RESPONSE:

This section of text has been removed because we now use a new method, following the advice of this reviewer (see next response).

COMMENT:

Nevertheless, I think your choice of bandwidth estimation is not optimal, since you have considered the whole pool of species in the study (I think; Line 100 in the script). This would be the optimal approach if you were ultimately interested in estimating the TPD function for the full set of species (as in a “Global spectrum” approach).

However, here you are interested in estimating the TPD functions of individual assemblages. Using all species at once is likely to result in bandwidths that are too narrow for the scale of interest (the individual sites), so that the TPD functions of the sites will be too “peaky”, the overlap between species will be too small, and things like functional richness and functional redundancy will tend to mimic species richness much more strongly. In general, the approach to estimate TPD functions that makes sense to me would be the following:

--Step 1: Choose a common bandwidth for all species. You have done this using the `Hpi.diag` function, but applying it to the whole set of species, which is not optimal. I think the best way to solve this issue is to use `Hpi.diag` to estimate a bandwidth for each site (i.e. 1,281 bandwidths), then estimate an average of those bandwidths.

--Step 2: Combining the (square root) of that average bandwidth and the mean position of each species in the three-dimensional space, estimate a single TPD function for EACH INDIVIDUAL SPECIES, using the `TPDsMean` function (by the way, the function's name is misspelled in the main text); you have done this in L117 of the script, and it seems correct.

--Step 3: For each community (site), aggregate the (abundance-weighted) TPD functions of the species that are present. This can be done in the TPD package with the `TPDc` function; you have done this in Line 189, and it also seems correct.

RESPONSE:

We are very grateful for this detailed and insightful suggestion, which we have now implemented in full. Using this new method, we found that our previous approach likely underestimated the amount of overlap between species in three-dimensional trait space. Accordingly, the results generated from the updated method tend to increase redundancy in our TPDs. We have corrected the name of the function in the main text

and edited our methods section and Supplementary Information to clarify the rationale and details of the new approach.

COMMENT:

Other minor comments:

-Lines 126-127: “we calculated functional redundancy for each assemblage as the amount of shared niche overlap between co-occurring species.” I think this is not the best definition of functional redundancy under the TPD framework (see Box 2 in Carmona et al. 2016 TREE). FR does not only consider what proportion of functional space is occupied by multiple species, but also by HOW MANY SPECIES. This detail is important, since an assemblage that includes five species occupying the space in the same way should have higher redundancy than one including two species. Your description of redundancy here (and also in ED, Fig. 4) does not seem to account for this aspect. Rather, I would say something like “we calculated functional redundancy for each assemblage as the average number of species (across all parts of the functional space occupied by the assemblage) that could be removed without reducing the functional volume occupied.” Just a proposal; I am sure there is more elegant way and shorter way to write this.

RESPONSE:

Thank you for the clarification. We have edited the description of our redundancy calculation so that it is now in-line with the description in Carmona et al., 2016 (line 535-538). To ensure our main text remained concise we have included a shorter version in the main text and included the full description of redundancy (Carmona et al., 2016) in the Supplementary Information (see SI section titled “Calculating functional diversity and redundancy”).

COMMENT:

Referee #2 (Remarks on code availability):

I have reviewed parts of the code, which is very extensive. I haven't ran it, but I have been able to understand most of what I have seen. I think it will be a very useful resource for the community. My only concern is that some of the scripts use code from other authors, adapted in some cases, and I think this should be acknowledged in some way (either in the header of the script, or in the manuscript itself).

RESPONSE:

Good point. We have now edited the text to highlight that we followed previous methods, and added citations to acknowledge sources of code in the headers of our scripts. We have also added the relevant authors to the acknowledgements section.

Referee #3 (Remarks to the Author):

COMMENT:

Thank you for giving the opportunity to review “Land use change undermines the stability of avian functional diversity” for consideration in Nature. There are many things to love about this manuscript. The ideas are original, the dataset is large and thus adequate to address the questions, the writing is quite clear, the findings are very interesting, and the topic would be of interest to a broad swath of scientists and practitioners. I particularly like the alternative hypotheses that the authors pose for why disturbed land uses may be more (or less!) resilient to future species loss. I haven’t seen these hypotheses before and was really excited by the way the authors chose to address this topic. The finding that disturbed land uses lack functional redundancy and thus may be even more vulnerable to future biodiversity loss is an important and novel contribution to the ecological literature.

As mentioned, I have articulated all my thoughts below in a point-by-point manner. Again, thanks for giving me the opportunity to review this manuscript. I really enjoyed it and learned a lot from the authors’ great work.

RESPONSE:

Thanks for the encouragement – much appreciated!

COMMENT:

That said, I do have a number of concerns with this manuscript, articulated in depth below. Though I bring up a number of issues, my biggest concern centers around the way that the authors simulate future extinctions to evaluate the relative importance of functional redundancy versus functional vulnerability. In short (and as I articulate below), I do not believe that this approach allows for the mechanism underpinning their second hypothesis to operate and thus think that their core finding from that effort is an artifact of the way their analysis was conducted.

RESPONSE:

We thank the reviewer for really examining our work in depth and making us go the extra mile. These efforts have helped to strengthen the manuscript substantially, particularly in relation to the reviewer’s “biggest concern”. Having said that, some of the changes suggested were difficult to implement, extending the revision process by several months. In some cases, we feel that the additional material is a bit tangential and over-complicates the manuscript, so we have had to relocate some sensitivity analyses to the SI. Nonetheless, we hope that the 18-month revision process helps to confirm the robustness of our results!

COMMENT:

Introduction

Lines 66-70: I'm curious why the authors do not explicitly reference the Etard et al 2022 paper that they cite in other places, both here and more explicitly in reference to their findings. This paper uses the same dataset (PREDICTS) but different traits and a broader array of taxa to look at the effects of land use on functional diversity. Unlike the statement here (line 66-70), Etard et al 2022 do find strong declines in FD with land use, a finding that is echoed in Fig 2b. I would explicitly reference this paper when you talk about your findings regarding functional diversity (Lines 144-122), saying that your work aligns with prior analyses of the same dataset.

RESPONSE:

The reviewer correctly points out that our work is built on the foundation of earlier papers using PREDICTS, so we have followed their advice in citing Etard et al. 2022 (line 57) and emphasising that the first layer of introductory findings that we report broadly align with previous research. However, the suggestion that we use the same earlier version of the PREDICTS dataset as Etard et al. 2022 (and other papers using PREDICTS) is incorrect. We spent several months collecting and integrating new survey data, including major landscape-level studies, so our dataset is a much-expanded version of Hudson et al. (2017), the PREDICTS data used by Etard et al 2022. To clarify our additional work and the large increment in data available, we have added a new table (Table S1) explaining the different sources of data used in our analyses.

COMMENT:

Lines 106-110: Here and throughout the authors suggest that morphological diversity has well established links to ecological processes and functions in birds. As a result, the authors make strong statements about ecosystem functions and processes; for example, at the end of the abstract: "Our analyses reveal that land-use change may have major undetected impacts on the stability of key ecological functions, hindering the capacity of natural ecosystems to absorb further declines in functionality caused by ongoing perturbations." The evidence they cite to connect morphological diversity to ecosystem functions/processes is often Pigot et al 2020. The Pigot paper connects morphology quite strongly to other functional traits (i.e., trophic position, diet, etc.) and therefore morphology does seem to be a good surrogate for species' ecological roles in ecosystems. However, I do not see evidence in that paper or others cited here that functional diversity has been linked to actual ecosystem processes/functions in birds. Consider insect predation. One meta-analysis showed that a variety of functional diversity measures correlate with insect predation in tropical agroforests. However, no

FD measure performed better than species richness and, ultimately, the authors concluded that sampling effects are likely the driver, with a few key bird species providing the bulk of pest removal services (Philpott et al 2009 Ecological Applications). All this is to say that I do not believe the authors have established a firm connection between functional diversity and actual ecological processes in birds. That doesn't mean such a connection is absent- it would make intuitive sense for it occur. I would just recommend that the authors be more circumspect about this in the writing. I would also include a statement in a limitations section saying that the connection between functional diversity and actual ecosystem functions/services in birds has yet to be firmly established (that is, of course, unless the authors can cite papers firmly linking functional diversity/morphological diversity to measurements of ecosystem functions/processes).

RESPONSE:

These are good points and we agree that the text needs tightening up. The reviewer is correct that Philpott et al. (2009) published a meta-analysis suggesting that functional richness did no better than species richness in predicting pest consumption by insectivorous birds. However, the lack of clear finding in this study is no surprise given that it focuses on a local system with 106 bird species classified into four crude functional trait categories (e.g. foraging strata, diet, etc.). Fortunately, much progress has been made in the 16 years since the Philpott et al. study, capitalising on the recent accessibility of more comprehensive trait databases.

A wide range of empirical studies now support the view that functional diversity/richness within an assemblage predicts ecosystem multifunctionality (Huang et al., 2019; Lefcheck & Duffy, 2015), above-ground biomass in forests (Hao et al., 2020; Looney et al., 2023), and specific processes, such as biogeochemical cycling (Gagic et al. 2015), pollination and crop yield (Woodcock et al., 2019); carrion removal (Mosman et al., 2023) and decomposition rate (Dekanova et al., 2023). In birds, experimental studies reveal a correlation between functional richness and rates of insect predation (Barbaro et al., 2014) and pest control (Barbaro et al., 2016), similar to observational studies that also link functional trait diversity to pest-control (Martinez-Salinas et al., 2016) and seed dispersal (Saavedra et al., 2016). As the reviewers' note, these relationships are intuitive and supported by theoretical models (Ceulemans et al. 2019).

Using some of this information, we now strengthen the support for our main assumptions in the text, citing previous work where possible. We try to balance this growing evidence base for form-function relationships with some of the cautionary language advocated by the reviewers. In particular, we emphasise that the connection between functional diversity and specific functional outputs is nuanced, sometimes

inconsistent and requires further direct empirical demonstration. We have added some of this discussion to the main text (lines 59-66), and pointed from there to a more detailed section in the Supplementary Information (titled “Functional traits predict ecological processes”). We also discuss limitations in a caveats section at the end of the main text (lines 310-324).

COMMENT:

I am not sure that the conceptual diagram (i.e., Fig 1) is that helpful. There is a lot going on in this figure and it takes much more time to try to understand the figure than it does to understand the core concepts being communicated. For example, for panels b and c, bars need to be compared between the far left and far right sides of the graph to determine how land use transformations effect functional diversity/redundance. The reader intuitively wants to compare adjacent bars, but these are FD vs. FR and thus not intended to be compared. If the authors would like to keep a conceptual figure, I would consider simplifying it to only visually depict the two alternative hypotheses about how land use affects functional stability.

RESPONSE:

Agreed. We have taken the reviewer’s advice and remade the conceptual figure from scratch to explain our usage of the key terms, and to outline the hypotheses in much-simplified form. We think this is a more effective starting point for the study.

COMMENT: Lines 110-112: Please clearly define what an assemblage is upon first mention. Is it a list of species present at a given site? From one survey? Multiple surveys? Can multiple sites be aggregated (e.g., two point counts in the same area) or is it just one location?

RESPONSE:

We follow the growing trend in macroecology of using the term “assemblage” when no evidence is shown that species are directly interacting. Field survey data, such as that included in PREDICTS, produces lists of species but no evidence that they are interacting or interdependent “communities” in the ecological sense. See discussion in Blanchet et al. (2020). The term “community” is routinely used in publications like ours with barely any definition, despite the fact that such usage is, strictly speaking, inaccurate. We prefer the term “assemblage”, which refers to organisms occurring in spatial proximity (i.e. not necessarily interacting) within the spatial and temporal extent defined by the biodiversity survey.

We agree that further definition is required to clarify what we mean by “assemblage” in the context of our study, particularly because the scale of survey units vary across

published datasets, and even across data repositories such as PREDICTS. We define an assemblage as all the species detected across multiple sampling points that are (1) within the same study-block and (2) share the same land-use type. We processed data from many studies both within and outside PREDICTS by batching together different survey points into assemblages. This is often necessary because species richness is too low to calculate functional metrics for single points, and in many cases the survey points are so close together that individual birds may be double-counted (pseudo-replicated). Although we tried to standardise the scale of assemblages, it is impossible to align all studies perfectly because field methods and sampling resolution varied across different studies. However, we standardise carefully within studies. This means that the inconsistency across biodiversity surveys should not affect our results because differences in functional metrics are mainly compared between landscapes surveyed with the same protocol (this is the reason we use a mixed-effects modelling approach with study and study-block as random effects).

It is not possible to include all the details requested by the reviewer at the first mention of “assemblage”, so we now define our use of the term “assemblage” in the methods (line 382-385). We also include new sections in the Supplementary Information to explain this information in greater depth (sections titled “Species assemblages” and “Standardizing sampling units”).

Results and Discussion

COMMENT:

Fig 2: I would recommend running some post-hoc tests to identify differences among land-use categories and then denote significance between them with letters in the graphs. As is, the graph clearly shows differences relative to a ‘primary vegetation’ baseline. However, many readers might also be interested in knowing, for example, if there are significant differences between minimally and intensively used urban areas.

RESPONSE:

We appreciate the idea, but this is a non-trivial analysis which would take a large amount of time-investment for little gain. These are patterns that have been shown before whereas we are trying to ask a different and newer question with our analyses. Ultimately, we think the additional information would complicate the figures and narrative, and perhaps confuse the reader into thinking we are asking questions tackled by previous papers. For these reasons, we hope the reviewer will forgive us for skipping these post-hoc tests.

COMMENT:

Lines 125-140: I appreciate the author's explanations in the extended data for why they did not choose to use species-richness corrected measures of FD/FR. I agree that doing so would mask important effects on ecosystem function.

RESPONSE:

We are glad the reviewer agrees with this decision. We think correcting for species richness misses the point of our analyses and we now provide a clear explanation in the Supplementary materials (section titled "Role of species richness in assemblage function").

COMMENT:

That said, I'm still left feeling really curious about how doing so would affect the patterns in FD/FR across land uses. Were the most functionally unique species lost first from land-use change, causing declines in FD after accounting for changes in richness? Is it that the least unique species were lost first, causing a decline in FR but not FD after accounting for richness? Is it more of a random pattern of loss, such that FD and FR do not change after accounting for changing in richness? I'd suggest the authors include such an analysis as an addition to the manuscript, not to make statements about ecological functioning, but rather to make some statements about the pattern of functional diversity/redundancy loss with land-use change.

RESPONSE:

We agree that these patterns are inherently interesting to ecologists and that further details should be included. Two reviewers have suggested that we compare declines in functional resistance to species richness-adjusted methods, so we have now added a linear model analysing the relationship between SR-standardized functional redundancy and functional resistance (Figure S4). These analyses show that SR-standardized redundancy drives losses of functional resistance. In addition, SR-standardized redundancy declines more rapidly than species richness as land-use change often targets redundant species rather than specifically targeting the most unique species in the assemblage. These results support our argument that resistance is greatly reduced in human-modified landscapes as functional trait space is more thinly spread across the functional space.

We have not included the other suggested analyses focusing on standardized FD measures as these would require substantial additional time-investment and we don't think these extra analyses are necessary because the key points can be made without them. Although exploring the differences between raw FD and standardized FD seems only marginally relevant to the main thrust of our paper, we agree it's an interesting topic for future research.

COMMENT:

Lines 141-152: I wanted to bring up a potential issue with dividing birds into generalists, granivores, insectivores, and frugivores guilds. Are you not eliminating some key functional redundancy inherently by relegating birds into the generalist category? For insect predation, both generalists and specialist insectivores eat insects. They may do so in functionally redundant (or distinct) ways. So why would you not consider the generalists as well if you are interested in the particular ecosystem function of insectivory? The same statement could be made for seed dispersal and seed predation. Instead, to analyze redundancy/diversity in insectivory (for example), I'd subset to the fraction of community that eats insects (you could impose some sort of threshold; say, 25% or more of their diets is insects). Then conduct the analyses. The generalists would appear in multiple analyses but that's ok- they would still be there to provide (or not provide) the function. Also, the patterns exhibited here are entirely consistent with the observation that you often see strong declines in the richness of insectivores/frugivores but not generalists/granivores. Again, I'd be curious if the most functionally unique/redundant birds are lost first (a question that could be answered with richness-adjusted measures).

RESPONSE:

We agree with the reviewer that species with generalist diets can contribute to ecosystem function by performing similar roles to specialist species. However, it seems likely that their consumption of a wide variety of food types mean that generalist species have a more limited role in the delivery of specific functions/services than specialist species which focus mainly on single food types. Adding generalists would likely introduce a lot more noise into the analysis because some abundant generalists potentially contribute more to a particular process than rare specialists, while rarer generalists may contribute very little if anything. Overall, we agree that generalist species should be (carefully) included in ecosystem-level models, but we also emphasise that doing so weakens the hypothesis tests in this paper by adding a lot of uncertainty.

For our main analyses, therefore, we use the same >60% grouping as our previous manuscript. To address the reviewer's comment, however, we also include a more relaxed definition of trophic guilds in an additional analysis, assigning species to guilds if they consume >25% of their diet from a specific food source. This was a major undertaking in terms of time and computing power. The results are similar. We have edited the text to explain the rationale for both specialist (>60% of diet) and generalist (>25% of diet) groupings in the methods (lines 480-489) and to report findings from the additional analyses (lines 163-167; Extended Data Fig. 3).

COMMENT:

Lines 169-171: I have a few suggestions about additional analyses that could be done to robustly examine patterns of subsequent species loss.

First, it would be nice to do a little bit of sensitivity analyses surrounding the response traits you are choosing. As the authors are well aware from their own work (i.e., Hatfield et al 2017 Eco Apps), response traits can be especially fickle in their ability to predict species responses to disturbances across databases. It would thus be nice to see that the patterns exhibited here are robust to different choices about which traits they include. For example, beyond the four traits examined here, Etard and Newbold 2023 outline a few others that showed predictive ability in dictating species responses to land-use change (e.g., habitat breadth and specialization on natural habitats).

RESPONSE: We appreciate these suggestions. To provide some ecological realism to our extinction scenarios, we have used response traits predicting sensitivity to land-use change, prioritising traits with the strongest predictive power and which are also available at global scales for the 3696 bird species in our sample. The reviewer is correct that many such traits have been suggested and we agree that an exploration of the effects of our choices of response traits would be useful. To implement this, we have re-run analyses using an alternative set of traits: minimum elevation, temperature seasonality and generation length. These traits were selected to represent response traits which have specifically been identified for birds in response to climate change (Pacifiçi et al., 2017) as noted by the reviewer below.

The results are similar to our original analyses, suggesting that our findings are robust to trait choice and also giving insights to the ability of our assemblages to maintain function in response to climate change. We now explain our rationale for an expanded set of response traits in Table S2 and present the results in Fig. S2. Unfortunately, it is not possible to include all response traits as many are not available with sufficient robustness and resolution for our total species sample (increased sampling since earlier versions of PREDICTS).

COMMENT:

Second, I might also be explicit what sort of 'future disturbances' you are considering that could induce future species losses. As is, the disturbances are really vague and could represent anything. I see the value in that (we don't know what the next disturbance will be) but I also think an additional analysis of a more concrete stressor that could cause further extirpations would be interesting. One possibility could be climate change. You could use some traits that have been shown to be at least moderately predictive of species abilities to weather climate change and implement

those as the ones to determine species sensitivity. Some such traits are present in Pacifici et al 2017 Nature Climate Change, for example.

RESPONSE:

Future disturbances to ecosystems are uncertain, involving anything from changes in temperature, fires, storms, drought, disease, pollutants, hunting and human disturbance, all of which may exacerbate the impacts of land-use change. Nonetheless there are common patterns in response traits that tend to increase sensitivity to any disturbance. We therefore prefer to select a basket of traits with increased likelihood of predicting responses to most perturbations. We now explain this approach and list some of these possible future disturbances in our Methods (lines 580).

However, we take the reviewer's point that it would strengthen the paper to include a more explicit stressor. We follow their advice in using additional metrics thought to predict sensitivity of bird species to climate change (Pacifici et al. 2017), including higher elevational distributions, lower temperature seasonality, longer generations and reduced dispersal ability. We have re-run our simulated extinctions on post land-use change assemblages based on these metrics, which essentially tests possible impacts of the classic double-whammy of land-use change followed by climate change (a commonly proposed synergistic threat). The results indicate that land-use change erodes functional resistance to future climatic change. We have modified our text to clarify our approach to identifying response traits and to include this new sensitivity analysis (Figure S3; Table S2).

COMMENT:

- Lines 176-178: Just to make it extra clear to the reader, I'd add a clause saying that a decline in Functional Vulnerability with land use is, essentially, a good thing. That is, land use is causing the assemblages to be less functionally vulnerable. It took me a couple reads to get that.

RESPONSE:

Done.

COMMENT: Lines 178-180: While the two metrics do produce broadly similar trends, there are some key differences. The response trait version shows a much more confident decline in FV for intensive urban and plantation sites, whereas the rarity-based one shows stronger trends for pasture and cropland. I'd point out this difference to be upfront with the reader.

RESPONSE:

With our revised methodology, we now find different patterns in our results which are stated in our main text (Lines 197-205)

COMMENT:

Lines 183-185: Could you rephrase this sentence? I'm not really understanding what it is getting at.

RESPONSE:

We have edited this section for clarity and that sentence has been removed from the ms.

COMMENT:

Lines 202-203: Is there a reason you show a graph for the redundancy predictor (Extended Data 4) but not for the FV predictor?

RESPONSE:

Based on reviewers' comments, we have removed this result from the ms to make room for additional analyses and discussion relating to reviewer suggestions.

COMMENT:

Lines 213-221: As I alluded to above, I am sceptical that the future anthropogenic extinction analyses is set up correctly to allow the authors to arrive at the conclusion that the functional redundancy hypothesis (i.e., that intensive land uses will be less resilient because they have lower redundancy) is more supported than the functional vulnerability hypotheses (i.e., that intensive land uses will be more resilient because they already lost their extinction prone species). The way the authors compare these hypotheses is by sequencing deleting species from the community, ordered by response traits or population size (rarity). They then compare the extinction curves between different land use types, using both areas under the curve as well as half-lives of the extinction. Unless I am totally off-base, this approach bypasses the author's second hypothesis entirely. This is because the authors are forcing the species to be sequentially extirpated. The order is indeed determined by their vulnerability, but all species are eventually extirpated to generate the curve. There is thus no way that the intensive land uses can hold the species that remain (and thus their FD) for a longer time than the pristine habitats. As a result, the exercise is only allowing the functional redundancy hypothesis to operate. Off the top of my head, an alternative approach would be to probabilistically remove species according to their rarity or traits over a series of time of steps. Basically, each species could have an extinction probability that is proportional to its response traits or its population size. Then, through a series of sequential random draws applied to each species in the community (i.e., a series of

time steps), communities in each land use could lose species, with rare species having a higher likelihood of extirpation than common species. FD could be assessed at each time step or at the end of the trial. This approach would allow for the possibility that intensive land uses are less likely to lose species because they have already filtered out the vulnerable ones (i.e., the authors' second hypotheses). The current approach, however, does not allow for this possibility; therefore, I do not believe the authors can conclude that intensive land uses are indeed less resilient to future functional diversity loss.

RESPONSE:

We appreciate this insightful comment. We have now re-run the analyses using time-steps as described by the reviewer (several months of work and computation!). The approach we use is to assign a probability of extinction to each species at each timestep based on the sum of their response traits. Accordingly, species with small range sizes, large body sizes, poor dispersal abilities and specialised diets have a higher probability of being removed at each timestep than less sensitive species. In line with the reviewer's point, our method is designed to account for the likelihood that less vulnerable species will predominate in species assemblages in anthropogenic landscapes.

There are two major differences in the new approach. First, we allow zero extinctions at each timestep so that we are no longer "forcing species to be sequentially extirpated", and second, we set a maximum cap of 2 species going extinct at each time step. This capped approach reflects the reality that extinction scenarios often play out over an extended period. We assign a lower probability of extinction to species with less sensitive response traits, although this probability is always more than zero, even for species associated with urban habitats and heavily modified farmland. This because even those species well-adapted to highly modified habitats are sometimes locally extirpated by unexpected threats such as disease, pollutants etc., as illustrated by the fate of once-abundant birds (e.g. suburban populations of house sparrows and chaffinches disappearing in response to pollution and disease, respectively). We describe and justify our new approach in detail in the Methods (lines 641-648) and Supplementary Information (new section titled "*Simulations using passive (probability-weighted) extinction*").

The results – presented in Fig S2 and discussed in the main text (lines 231-245) – show that under this probabilistic method of local extinction assemblages in disturbed landscapes have significantly lower functional resistance than those in human-modified habitats. We agree that this method is useful to include, although the resistance metric is inherently related to species richness as more timesteps are required to eradicate the

entire assemblage. Therefore, we think it is best to present these results in conjunction with our original analysis which estimates changes in redundancy independently of species richness (because multiple highly vulnerable species can go extinct in a single time-step).

Given that the new resistance method suggested by the reviewer is highly sensitive to species richness, we also include an additional analysis assessing the effect of SR-standardized redundancy on functional resistance (Figure S4). This analysis shows that the average niche-space overlap per species is a clear driver of functional resistance losses across all species-loss scenarios.

COMMENT:

Fig 3c (and figure 2): This is just my preference but I think these graphs would be a lot easier to understand if the y-axis graphed the actual metrics (i.e., FD, FR, FV, or AUC from the extinction trial) rather than the coefficient estimate with primary forest set as a baseline.

RESPONSE:

Throughout the paper, we have removed the use of “coefficient estimate” as a y-axis label and replaced it with the name of the response variable. We think the figures are now clearer. In the specific case of Figure 2, we have maintained a common y-axis to allow comparison across the different functional metrics. Here we maintain plot titles and add a common y-axis title that reads “Functional metric (z-score)” with further explanation in the caption.

COMMENT:

Especially for Fig. 3, I had a hard time trying to understand what exactly was being graphed (before spending a long time reading the rest of the paper and the extended data). If figure 3c was clearly indicated as AUC, and you could see the curves above, then it would make more sense what was going on. I'd also consider including extended data 3 in figure 3 as this really helped me understand what the AUC was doing and why it was included.

RESPONSE:

We thank the reviewer for their advice on this figure, which is a key aspect of our paper. We have edited the names of the y-axis throughout the ms to increase clarity. As AUC will sound a bit obscure to general readers, we have labelled this axis “Functional Resistance” and then explain in the figure caption that this is calculated as AUC.

We are glad that extended data Fig 3 helped to explain what AUC is doing and why it is useful. We explored the reviewer's suggestion of adding extended data Fig. 3 to Fig. 3 but this didn't seem to work well design-wise and it's easy enough for readers to refer to the extended data figures. In addition, we think that our refined conceptual figure (Fig 1) now helps to clarify our approach and aids interpretation of Fig. 3.

COMMENT:

Lines 909-910: I would add, as a clause, that the actual analysis uses proportional removal (not raw species) so that the starting species richness does not drive the results. Indeed, the AUC for intensive sites would be lower if proportional removal did not occur, simply because there are fewer species to start with.

RESPONSE:

Done.

COMMENT:

Lines 226-229: As above, I would reconsider separating generalists from insectivores, for example, if you are looking at the robustness of the ecological process of insect removal (see rationale above).

RESPONSE:

We have run a sensitivity analysis expanding the species sample with a 25% cutoff level as suggested by the reviewer above. The results are similar and reported in our main text (Lines 163-166; Extended Data Fig 2). We have retained our previous analyses in the main paper because they are more streamlined and provide a more unambiguous test of the delivery of core functions (see response above).

COMMENT:

Lines 255-261: Again, I do not think you can make these statements based on the extinction simulation (see rationale above). Indeed, these statements are pervasive in the discussion (e.g., lines 267-271).

RESPONSE:

We agree that the reduced sensitivity of species in disturbed habitats should enhance the functional resistance of species assemblages to land-use change. The reviewer has correctly pointed out that a probabilistic extinction scenario would help to account for this. We have now included this supplementary analysis (Figure S3). The results show that assemblages in disturbed landscapes are less resistant than those in primary habitats, consistent with our previous analysis and conclusions. We find that although the effects of reduced species-sensitivities likely increase functional resistance as less-

vulnerable species are harder to remove, the fact that these landscapes consist of far fewer species with very limited redundancy has an overwhelmingly negative effect on functional resistance. In other words, the effects of reduced sensitivity are outweighed by reduced species richness and a thinning out of functional diversity (i.e. a lowering of redundancy).

The reasons for this finding are (1) there are still remnant populations of vulnerable species with unique traits in human-modified landscapes such as agriculture or suburban settings, and (2) we assign a low but non-zero probability of extinction to low-sensitivity species. This means that both residual primary-habitat species and relatively insensitive species are both lost from modified landscapes in our extinction scenarios, albeit at different rates. We think both these assumptions are valid and realistic.

The main drawback of this approach, as we mentioned above, is that the results are highly sensitive to species richness as larger assemblages inherently need more timesteps for the entire assemblage to be lost. In effect, this will artificially inflate the functional resistance of larger assemblages regardless of the functional structure of the community. As we cannot standardize functional resistance scores generated through this method by the starting species-richness, we elect to keep our original analysis in the main ms, supported by this more nuanced method as a supplementary analysis. To show that the reduced functional resistance in this supplementary analysis is driven by redundancy losses irrespective of species richness, we have added an additional analysis which assesses the effects of SR-adjusted redundancy on functional resistance across all three of our species-loss scenarios (Figure S4).

COMMENT:

Lines 262-264: I may have missed this, but where do you show that the species in pristine habitats are typically (and disproportionately) functionally unique? Wouldn't you need to do the richness-corrected analysis to show this?

RESPONSE:

The reviewer is right that we didn't explain this clearly. Based on our results, it is possible to infer that disturbance-sensitive species in pristine habitats have disproportionately high functional uniqueness for the following reason. Given that FV values are calculated as the inverse (i.e. negative) covariance between redundancy and sensitivity scores, it follows that high FV values indicate strong covariance between the uniqueness (= low redundancy) of species and their sensitivity to disturbance. Thus, the high FV of undisturbed (pristine) primary vegetation suggests that species in these habitats are, on average, functionally unique. These points are now explained more clearly in the main ms and Supplementary Information.

COMMENT

A general comment for the discussion. I would appreciate a brief paragraph on limitations. For example, I would mention the relative tenuous connection between FD and ecological processes provided by birds (as mentioned below). I would also discuss issues of detection (as outlined below).

RESPONSE:

The link between trait diversity and ecological processes is intuitive and increasingly well established with empirical support. We have added a section in the Supplementary Information to clarify this point with a more thorough set of supporting citations (see “Functional traits predict ecological processes” on the first page of the SI). In this section, we provide examples of empirical relationships between avian FD and ecological processes.

We have followed the reviewer’s advice and add a section on limitations in the main ms. In this section, we mention the possible role of detection in contributing to our results (lines 315-318). We point the reader to supplementary sections supporting our assumptions in more detail, including discussion of species detectability across different landscapes.

Methods

COMMENT:

Lines 297-299: What version of the PREDICTs dataset did you use? This seems to indicate the original version that was published in 2014. Is there a more recent version with more data? I had thought so but maybe I am wrong.

RESPONSE:

Good point. We use a more up-to-date version of PREDICTS (downloaded 24th Feb 2021) and we now cite Hudson et al., 2017 to reflect this. Our dataset is bolstered by further work by several members of our team who undertook literature surveys to access bird diversity data from additional sites worldwide, resulting in the inclusion of many additional studies (these are now incorporated into the latest internal release of PREDICTS: Contu et al. 2022). In addition, we have added data from two large landscape surveys in Amazonia and Borneo (not integrated with PREDICTS because of data ownership issues). We have thoroughly edited the methods and Supplementary Information, to clarify the improvements in our survey data compared with the original (2017) version of PREDICTS. To help clarify all the different sources of data, and their partitioning across continents, we now summarise the dataset in Table S1.

COMMENT:

Lines 318-321: I think this discussion of temporal or geographical blocks could be written a bit more clearly. It makes sense in the extended data but is a bit hard to understand here. Could you rephrase, perhaps adding another sentence for clarification?

RESPONSE:

We agree this was hard to understand in this compressed version. We have edited the text to unpack the methods (both in the main ms and the Supplementary Information) and we hope the procedure is now much clearer.

COMMENT: Lines 322: There's a grammatical issue in this sentence (which there?)

RESPONSE: Edited.

COMMENT: Lines 325-328: How did you determine what was close enough together to be aggregated into one assemblage? After aggregating points, how would you then account for variable effort across sites (i.e., some sites represent an aggregation of multiple surveys and others do not)?

RESPONSE:

Within each study-block we aggregate all survey sites with the same land-use and land-use intensity to form a species assemblage. All aggregated data represent multiple survey sites, so the reviewer is incorrect to imply that some assemblages are based on a single survey point. Nonetheless, sampling effort does vary somewhat across assemblages. To account for this, we follow methods implemented by Hudson et al. (2017) and used in many previous papers based on the PREDICTS database. Specifically, we first rescaled the sampling effort within each study by dividing by the maximum sampling effort in the study, and then we transformed abundance into effort-corrected abundance by dividing abundance measurements by the rescaled sampling effort. If species appear more than once in the aggregated sites, we sum the effort-corrected measurements as an abundance for each species. We have added new text to the Supplementary Information to clarify this procedure. The reason we use this method is simply to be consistent with the studies that PREDICTS had already aggregated into study blocks, and to allow us to contribute all data added by this study to the PREDICTS database.

We note that Newbold et al. (2015) – based on a smaller subset of PREDICTS data – used rarefied richness as a sensitivity analysis and this produced similar results to their main analyses.

COMMENT: Lines 363-368: I understand the rationale for using PCA to collapse the trophic traits and locomotory traits, as well as the rationale for using the second PC axis (which didn't correlate with body size). However, why bother conducting another PCA on the first axes to get a body size axis? Why not use body size directly at that point? This would much more directly measure body size than a double PCA.

RESPONSE: This is a fair point but we are mainly interested in trait size not body mass. (We assume the reviewer is referring to widely available body mass data rather than body length measurements, which are rather inaccurate and inconsistent for a number of reasons related to specimen preparation techniques and varying practices used in measurement protocols [e.g. beak tip to tail tip vs. beak tip to toe-tip]).

Overall trait size and body mass are different since birds can be relatively small in dimensions but much heavier in mass, and vice versa. A species can have long beak and wings while their body mass is small, and the signal of these elongated traits will be present in our estimate of body size, but not in body mass. Different hummingbirds are often tiny, for example, differing little in body mass, whereas those with longer wings and beaks will have a larger body size in our analyses.

In effect, our size axis is composed of the combined size axes from the trophic (beak) and locomotory (wings/tail/tarsus) traits. We think this connects more clearly with the trophic niche than body mass alone. For instance, in the hummingbird example mentioned above, our body size data will perform better than body mass in predicting likelihood of pollinating flowers with different corolla length. We now explain our approach more fully in the methods, including an explanation of why we are interested in the overall size axis extracted from trait measurements rather than body mass (line 435-443).

COMMENT:

Lines 373-388: I am a little confused about how the diet data were used. Did you collapse the diet data into a single PC axis like the morphology data? Were all variables without collapsing to generate the distance matrixes?

RESPONSE:

We apologise for the unclear text in the previous version of the ms. We have now edited the text to emphasise that the morphology data were not collapsed into a single PC axis. We used three morphometric axes generated from three different PCA: (1) a locomotory trait axis, (2) a trophic trait axis, and (3) a body size axis. We supply these three scores to the gawdis package along with two other datasets: (4) Hand-wing index

as a metric of dispersal ability and (5) the complete (uncollapsed) diet proportion data from Pigot et al. (2020) structured. *Gawdis* is designed to create a distance matrix from these mixed data types and can specifically account for our proportional diet data. *Gawdis* generates a single distance matrix from all supplied traits by applying a weighting algorithm which ensures the multivariate distance matrix is equally correlated to theoretical distance matrices generated from each of the individual traits (locomotory axis, trophic axis, body size axis, dispersal axis, dietary proportions). Therefore, *gawdis* effectively treats the proportional diet data as a single axis of variation (albeit not a PC axis, contrary to the suggestion of the reviewer).

The final distance matrix expresses how unique or divergent species are from each other based on trait similarity. We then back-transform this distance matrix into 3D coordinates. Space constraints mean that we can't fully describe the workings of *gawdis* in our ms, but we have added some text to explain the method in more detail, citing the *gawdis* software which contains a full explanation (main ms section titles "Calculating functional diversity and redundancy").

COMMENT: Lines 384-388: How are the primary food type data used? Is it just to divide up the communities or are you actually using this information to calculate functional diversity as well? It would be good to clarify this here.

RESPONSE:

We use the primary food type data (i.e. invertivore, frugivore, etc.) only to divide assemblages into trophic guilds. However, the underlying proportional dietary data is used to calculate functional diversity. We agree this is a slightly subtle distinction that was not properly explained in the previous version of the ms.

First, we use the underlying dietary proportions to assign a dietary guild to any species that consumes $\geq 60\%$ of its diet from a particular food source (e.g. $\geq 60\%$ invertebrates = invertivore; $\geq 60\%$ fruit = frugivore, etc.). In this new version of the ms, we have also added additional analyses based on dietary guilds assigned to species consuming only $\geq 25\%$ of their diet from a single food source (lines 163-168; Extended Data Fig.2). The method section now describes and explains these changes.

Second, the same proportional diet data is used to calculate functional metrics for our full assemblage analyses (all diet types together). We use the *gawdis* function to group the dietary proportions into a single trait axis of variation, as described above. We then add this derived trait axis to several other trait axes to create trait dissimilarity matrices which are used to create our trait hypervolumes (TPDs). It is not possible to do this when creating TPDs for particular trophic groups because they all share the same

diet, so we generate guild-specific TPDs without dietary data. We now clarify the use of dietary data in creating TPDs in main ms and in detail in the Supplementary Information document.

COMMENT:

Lines 390-420: The authors are far better experts about functional diversity calculations (and their nuances) than I am, so I am not going to review the technical dimensions of how functional diversity is calculated (I'll leave that to another reviewer). That said, it is widely known that functional diversity outcomes are quite sensitive to the types of traits included. I understand (and agree with) the rationale for focusing on morphological diversity and diet. However, I do think it would be nice if the authors did some sensitivity analyses to see if all their analyses are robust to choosing additional traits. For example, Etard et al (2021) used PREDICTS (i.e., the same database) and analyzed FD using body mass, trophic level, lifespan, litter/clutch size, diel activity, habitat breadth, and use of artificial habitats as traits. I would thus encourage the authors to include a few other traits along the lines of other studies (like Etard et al) and then see if their findings are robust to trait decisions.

RESPONSE:

We can see the logic of this suggestion, but we prefer not to follow the tendency for previous studies to treat functional diversity as a random basket of available traits, even if they have been shown to predict the sensitivity of species to land-use change.

The reviewer accurately points out that trait choice can have a profound effect on functional diversity estimates, yet also appears to be suggesting that we include response traits that influence how organisms respond to land-use change. These traits are useful to test a commonly posed question: what determines species sensitivity to land-use change? In this study, we are concerned with a different question: how well do species assemblages deliver functions after land-use change? Therefore, we use response traits as a foundation for our sensitivity scores, and then focus on effect traits for our main analyses, to align trait-usage with underlying assumptions. We are mainly interested in whether species assemblages drive a reduced diversity of trophic interactions, so we estimate the trophic diversity reflected in dietary niches and morphological traits (which also reflect body size).

Our approach follows numerous publications (e.g. Lefcheck et al., 2014; Zhu et al., 2017; Hatfield et al., 2018) proposing that careful selection of effect traits related to trophic processes is preferable to running functional diversity analysis on a host of different trait combinations with a variety of implications. We would also like to stress that expanding our study to include a basket of response traits – including lifespan,

clutch size, habitat breadth, or use of artificial habitats – would greatly complicate our re-analyses, taking weeks or months to run and producing results that are much harder to interpret in the context of the hypotheses.

To clarify our approach, we now include a brief explanation of response and effect traits in the introduction, and provide much more detail in explaining our trait choice rationale (Table S2 and S3).

COMMENT:

Lines 398-399: Again, I am not an expert on these metrics but I was taken aback a little by the description of functional redundancy, which is measured as “the proportion of this volume that is shared by multiple species.” Consider two communities with identical fractions of the volume shared by more than 1 species, but in community A that volume is shared by 2 species and in community B that volume is shared by 10 species. According to the definition above, it would seem that both receive the same functional redundancy score even though community B clearly has more redundancy. Can you clarify this a bit?

RESPONSE:

We had tried to simplify our methods to make them more digestible to general readers, and we agree that this led to an oversimplified description of this particular method (a similar criticism was made by Reviewer 2). We have edited the sentence highlighted by the reviewer and expanded our definition in the SI to clarify the definition and quantification of redundancy more clearly (see section titled “Calculating functional diversity and redundancy” and to clarify that redundancy per grid cell of the TPD is weighted by the number of species sharing each grid cell. Given this weighting, community B would indeed have a higher redundancy in the example described by the reviewer.

COMMENT:

Lines 395-399: Are species abundances used in any way to calculate functional diversity or redundancy? From my read, it appears no. I wonder, however, if there might be a way to consider abundances as an even distribution of abundances across traits is probably much better for functioning than another community with equal trait diversity but that is dominated by a couple species.

RESPONSE:

In our analyses, FD is linked to the total volume of trait space occupied, including any cell of the TPD in which probability of occupancy is >0 . Variation in abundance does not influence this metric. However, the abundance of a species does affect the probability

that a particular TPD-cell is occupied, and this feeds into the redundancy metric via an internal algorithm used in TPD calculations. In effect, the TPD method assigns higher probabilities of occurrence in hypervolume cells within the kernel of intraspecific variation for a species, reflecting a more extensive filling of trait space, slightly inflating redundancy values (which are averaged across all occupants of the cell). We now explain this contribution of abundance to our functional redundancy calculation in the Methods (line 539-541) and at length in the Supplementary Information (section titled “Calculating functional diversity and redundancy”).

We agree that unequal distribution of species abundances across traits may influence the functioning of assemblages and that incorporating species abundances may provide additional insights into ecological function at the species assemblage level. On the other hand, more extensive inclusion of abundance-weighting would require assumptions about how species' functional contributions scale with abundance across a wide range of geographical, taxonomic and ecological contexts, none of which is well understood. We prefer to align with our current conceptual framework which emphasizes the presence and distribution of traits, rather than variation in species abundance. A deeper consideration of abundance is not possible with the methods used in this study, but the question is interesting, and a promising direction for future research.

COMMENT

Lines 407-417: The approach for generating intraspecific diversity via a kernel probability density was particularly challenging for me to understand. Where did the data for estimating intraspecific variation come from?

RESPONSE:

The method estimates intraspecific variation using a plug-in kernel density estimator obtained from the *ks* package in R (Duong et al., 2007). The estimator uses TPDs to generate the dimensions of a kernel based on a smoothing algorithm to minimize empty space across the trait hypervolume. This is a standard metric used in previous studies (Carmona et al. 2016; 2019; 2021).

The method assumes that kernels are the same across all species in all assemblages. On one hand, this assumption is clearly simplistic as all species are unlikely to be equally variable. On the other hand, this method is a major improvement on other approaches that do not account for intraspecific variation in the context of hypervolume metrics. Using the kernel density method, assemblages containing more distinct species are likely to have less overlap between kernels and a more “gappy” functional trait space, allowing us to compare between assemblages. Importantly, by calculating redundancy as a function of the amount of overlap between density kernels,

redundancy values are only affected by the numbers and proximity of species with high trait similarity – effectively quantifying how much clustering is occurring in the functional space.

Other methods of calculating redundancy consider the pairwise distances of all species in the community (e.g. Ricotta et al., 2016). While these calculations help to identify average similarity of the assemblages, they are less sensitive to clusters of highly similar species. For instance, using pairwise distances, an assemblage consisting of two vulture species (scavengers) and a goose may have a very different redundancy score than an assemblage containing two vulture species and a sparrow, even though both assemblages are equally redundant. Using the kernel density method, neither a goose nor a sparrow will overlap with the vulture kernels, so both assemblages will be correctly identified as equally redundant.

We use the plug-in kernel density method for our main analyses because it allows us to include (1) information about diet (categorical variables) and (2) species with poor sampling of trait measurements. We have now added a new sensitivity analysis to assess whether direct estimates of intraspecific variation would alter our results. This step is currently only possible in birds and we believe we are the first study to attempt this approach at a global scale. Specifically, we use direct measurements from a sample of specimens/live individuals to generate direct estimates of trait variation within all species with >4 individuals measured in the AVONET dataset. The results are similar (Supplementary Fig 1).

To avoid confusion, we now explain these two different methods for estimating intraspecific variation in greater depth in the main text. We then provide more detailed explanations of our secondary analysis based on morphological intraspecific variation in the Supplementary Information (see “Robustness and sensitivity analyses – (i) Alternative measure of intraspecific variation”).

COMMENT:

Lines 423-425: I am a little confused on how the functional diversity/redundancy metrics can be re-run on diet guilds if diet is also one of the key dimensions of in calculating functional diversity. Is it just that you are looking for variability in diet within the diet class? In this case, are less specialized insectivores, for example, more functionally diverse? If evaluating the potential impact of functional diversity within insectivores on insectivory, why should having a more diverse diet within in the community contribute to a higher rate of insect predation if diet diversity isn't measuring the diversity of insects in a bird's diet but rather non-insect diet items. In short, I'm not sure the diet trait should be included in functional diversity/redundancy analyses of specific diet guilds.

RESPONSE: The reviewer is correct that this would be a problem. Fortunately, we don't use diet in these analyses and instead calculate guild-specific TPDs using only morphological data. We have now clarified this point in the main ms and Supplementary Information.

COMMENT:

- Lines 428-433: Is this the same sensitivity analyses that was just referenced above in lines 417-420?

RESPONSE:

Correct. We have removed this duplication.

COMMENT:

- Lines 483-488: I am trying to wrap my head around the idea of standardizing FD to start at 1. Shouldn't functional diversity be on the same scale for all sites in the same study and wouldn't it thus make more sense to consider the absolute reduction of FD as a function of species loss? Imagine losing Species A causes a FD reduction of 1. In an intact community with an initial score of 10, losing Species A would represent a 10% decline (from 10 to 9). In a disturbed community with an initial score of 5, losing Species A would represent a 20% decline (from 5 to 4). But I would argue the effect on ecosystem function should be equivalent in both communities. Therefore, I wonder whether it makes sense to standardize in this case (unless I am missing something).

RESPONSE:

We do not agree with the reviewer's appraisal, and we believe they are indeed "missing something" – specifically, they do not appear to have considered the implications for the method we use to measure resistance. The main problem with non-standardised FD is that the area under the curve (AUC) is automatically lower in the landscapes with lower FD. This does not make sense in the context of our hypotheses because landscapes with low FD may be resistant to declines in FD, since they can be occupied by less vulnerable species. Our goal is to determine how resistant the current level of function in any given assemblage is to further disturbance. To achieve this, it is necessary to standardise the start-point to 1 so that we can use the AUC method. The reviewer is concerned that the loss of a single species represents a larger proportional decline in available function in a 5-species assemblage than a 10-species assemblage. However, we don't see this as a problem as it more clearly aligns with our hypotheses. We add an analysis to evaluate the extent to which our results are driven solely by SR (see Figure S4).

COMMENT:

Statistical analysis: I identified a number of potential issues with the statistical analyses. First, the issue of imperfect detection was never raised in this manuscript. Bird detectability is known to vary a lot across different land uses, which can heavily bias findings about the impact of land use on individual bird species and diversity metrics. For that reason, many studies implement models that account for detection directly (i.e., occupancy models, distance sampling, etc.). I understand this isn't possible using the PREDICTs dataset (or really any other meta-analytic effort). Nonetheless, this is a major limitation of global analyses like this one and should be clearly articulated as a limitation in the main text and discussed as such.

RESPONSE:

Imperfect or biased detection is a problem for all field surveys of birds and other animals regardless of methods. Nonetheless, we agree that these limitations should be acknowledged and discussed. We have added a new section ("Caveats and Limitations") to the main ms, where we signpost to further discussion in the Supplementary Information (new section titled "Survey methods and limitations"). Apart from acknowledging the problem of detection biases, we also now emphasise that FD and redundancy of primary habitats surveyed in space-for-time comparisons is substantially underestimated because of historical extinctions and reduced detectability of many species, particularly in primary forests. Conversely, our methods exaggerate diversity in anthropogenic environments where visibility is better, and birds are often more abundant and conspicuous. With perfect and unbiased detection, the taxonomic and functional diversity of primary vegetation would almost certainly be much higher, thus accentuating our finding of reduced FD and resistance in anthropogenic landscapes. In other words, our main conclusions are conservative (see section in Supplementary materials titled "Estimating variation in functional diversity and redundancy").

COMMENT:

Second, whereas the calculations of FD, FR, and FV are very nuanced, the models used to analyze them are very simplistic, with land use as the only predictor and study/study block included as random effects. As mentioned above, I do not believe that it would be possible to formally model detection. However, could the authors not at least include some study-specific covariates known to influence bird detectability (for example, day of year, time of day, or any other readily available covariate known to influence detectability)?

RESPONSE:

We agree that variation in detectability is a common problem when comparing field survey data. As the reviewer notes, bird detectability often varies with time of year, at least in seasonal environments. In our models, we already account for seasonal distinctions between survey times by partitioning data into study blocks within which all sites were surveyed in the same field season. Similarly, time of day is irrelevant as our assemblages are never single points and always contain samples of points, typically collected over a morning or multiple mornings at different times. Moreover, all bird surveys are focused in the morning (dawn until midday) across all studies. In any case, our modelling approach essentially means we are comparing within study-blocks, and therefore comparisons are limited to surveys specifically designed to permit biodiversity comparisons between sites (using standardised methods etc.). As the reviewer notes, we include study block and study landscape as random intercepts, based on previous analyses (e.g. Newbold et al., 2015; 2016; 2020) that used an earlier version of the same dataset (“PREDICTS A”). We note that our dataset is more robust than this previous version because we have removed several studies weakened by pseudo-replication and incomplete sampling, and also nearly doubled the sample size of study landscapes by identifying many higher quality surveys in a literature review (PREDICTS B), and adding other independent studies, some of them outside PREDICTS (see Table S3).

These improvements will not fully address the problem of varying bird detectability across land-use types. Nor is it likely that any study-specific covariates in the PREDICTS dataset would help to account for this variation in detection probabilities. We avoid including any further random effects as these would be difficult to implement consistently in the PREDICTS database. Instead, we include deeper discussion of the issues, citing papers where we have covered the same issues at length (e.g. Hatfield et al. 2023). See previous response for further details.

COMMENT:

The authors provide no discussion of model fit. Do all the models satisfy model assumptions about normality, heteroskedasticity, and other considerations?

RESPONSE:

We assessed model fit for all analyses and found that assumptions of linear models are not violated. We have added this information to the main text (line 700-706).

COMMENT:

Extended data: There is a lot of emphasis in the extended data focused on tropical/temperate comparisons but very little discussion of any of those findings in the

main text. I wonder if it would be better to either include a little discussion of this in the main text or not provide all those analyses if they are not of interest.

RESPONSE:

We retain these results because they help to link our study to related research themes, including latitudinal gradients in the effects of land-use change, with implications for policy (i.e. policies need to be locally tailored). We have reduced emphasis on this point in the Extended Data (only Extended Data Fig. 6 presents this result). To further address the reviewer's point, we now mention the results and their implications in the main text (lines 384-420) and expand our discussion of this topic in the supplementary file (section titled "Latitude").

References

- Anderson et al. (2011) Cascading effects of bird functional extinction reduce pollination and plant density. *Science* 331: 1068-1071.
- Barbaro, L., et al. (2014), Bird functional diversity enhances insectivory at forest edges: a transcontinental experiment. *Diversity Distrib.* 20: 149-159.
- Barbaro, L., et al. (2017) Avian pest control in vineyards is driven by interactions between bird functional diversity and landscape heterogeneity. *J Appl Ecol* 54: 500-508.
- Blanchet, F.G., Cazelles, K. & Gravel, D. (2020), Co-occurrence is not evidence of ecological interactions. *Ecol. Lett.* 23: 1050-1063.
- Bregman, T.P., et al. (2016) Using avian functional traits to quantify the impact of land-cover change on ecosystem processes linked to resilience in tropical forests. *Proc Roy Soc B* 283: 20161289.
- Burivalova, et al. (2015) Avian responses to selective logging shaped by species traits and logging practices. *Proc. R. Soc. B.* 282, 20150164.
- Carmona, C.P., et al. (2019). Trait probability density (TPD): measuring functional diversity across scales based on TPD with R. *Ecology* 100: e02876.
- Carmona, C.P., et al. (2016). Traits without borders: integrating functional diversity across scales. *Trends Ecol. Evol.* 31: 382-394.
- Carmona, C.P., et al. (2021). Erosion of global functional diversity across the tree of life. *Sci. Adv.* 7: eabf2675.
- Ceulemans, R., et al. (2019). The effects of functional diversity on biomass production, variability, and resilience of ecosystem functions in a tritrophic system. *Sci. Rep.* 9: 7541.
- Dekanova, V., et al. (2023). Functional diversity of shredders, not species richness, drives the decomposition rate of leaf litter in ponds. *Front. Ecol. Evol.* 11: 1286672.

- Etard, A., Pigot, A.L. & Newbold, T. (2022). Intensive human land uses negatively affect vertebrate functional diversity. *Ecol. Lett.* 25: 330-343.
- Etard, A. & Newbold, T. (2024). Species-level correlates of land-use responses and climate-change sensitivity in terrestrial vertebrates. *Conserv. Biol.* 38: e14208.
- Fenzel et al. (2016) Bird communities in agricultural landscapes: What are the current drivers of temporal trends? *Ecol. Indic.* 65: 113-121.
- Gagic, V., et al. (2015). Functional identity and diversity of animals predict ecosystem functioning better than species-based indices. *Proc. R. Soc. B* 282: 8.
- Galetti, M., et al. 2013 Functional extinction of birds drives rapid evolutionary changes in seed size. *Science* 340, 1086-1090.
- Guerrero et al. (2024) Agricultural intensification affects birds' trait diversity across Europe. *Basic Appl. Ecol.*, 74, 40-48.
- Hao, M., et al. (2020). Functional traits influence biomass and productivity through multiple mechanisms in a temperate secondary forest. *Eur J Forest Res* 139, 959-968.
- Hatfield, J.H., Harrison, M.L., & Banks-Leite, C. (2018). Functional diversity metrics: how they are affected by landscape change and how they represent ecosystem functioning in the tropics. *Current Landscape Ecology Reports*, 3: 35-42.
- Hodgson, D. et al. (2015) What do you mean, 'resilient'? *Trends Ecol. Evol.* 30: 503-506.
- Huang, X., et al. (2019). Functional diversity drives ecosystem multifunctionality in a *Pinus yunnanensis* natural secondary forest. *Sci Rep* 9: 6979.
- Lefcheck, J.S. & Duffy, J.E. (2015). Multitrophic functional diversity predicts ecosystem functioning in experimental assemblages of estuarine consumers. *Ecology* 96: 2973-2983.
- Lefcheck, J.S., Bastazini, V.A., & Griffin, J.N. (2015). Choosing and using multiple traits in functional diversity research. *Environ. Conserv.* 42: 104-107.
- Looney, C.E., et al. 2023. Functional diversity affects tree vigor, growth, and mortality in mixed-conifer/hardwood forests in California, U.S.A, in the absence of fire. *Forest Ecol. Manag.* 544: 121135.
- Martínez-Salinas et al. (2016). Bird functional diversity supports pest control services in a Costa Rican coffee farm. *Agric. Ecosyst. Environ.*, 235, 277-288.
- McLean, M., Auber, A., Graham, N.A.J., et al. (2019). Trait structure and redundancy determine sensitivity to disturbance in marine fish communities. *Glob Change Biol.* 25: 3424-3437.
- Mosman, J.D., et al. (2023). Scavenger richness and functional diversity modify carrion consumption in the surf zone of ocean beaches, *ICES J Marine Sci.* 80: 2024-2035.
- Naeem, S. (1998). Species redundancy and ecosystem reliability. *Conserv. Biol.* 12: 39-45.

- Neate-Clegg, M., et al. (2023). Traits shaping urban tolerance in birds differ around the world. *Current Biology* 33, P1677-1688.
- Nelson D. et al. (2021). Energy pathways modulate the resilience of stream invertebrate communities to drought. *J Anim Ecol.* 90: 2053-2064.
- Newbold, T., et al. (2013). Ecological traits affect the response of tropical forest bird species to land-use intensity. *Proc. R. Soc. B.* 280: 20122131.
- Newbold, T., et al. (2016). Has land use pushed terrestrial biodiversity beyond the planetary boundary? A global assessment. *Science* 353: 288-291.
- Newbold, T., et al. (2015). Global effects of land use on local terrestrial biodiversity. *Nature* 520: 45-50.
- Newbold, T., et al. (2020). Global effects of land use on biodiversity differ among functional groups. *Funct. Ecol.* 34: 684-693.
- Nunes, C.A., Barlow, J., França, F., Berenguer, E., Solar, R.R.C., Louzada, J., Leitão, R., Maia, L., Oliveira, V.H.F., Braga, R.F., Vaz-de-Mello, F.Z., Sayer, E.J. (2021). Functional redundancy of Amazonian dung beetles confers community-level resistance to primary forest disturbance. *Biotropica* 53: 1510-1521.
- Pacifici, M., Visconti, P., Butchart, S., et al. (2017). Species' traits influenced their response to recent climate change. *Nature Clim. Change* 7: 205-208.
- Philpott, S.M. et al. (2009). Functional richness and ecosystem services: bird predation on arthropods in tropical agroecosystems. *Ecol. Appl.* 19: 1858-1867.
- Pillar, V.D., Blanco, C.C., Müller, S.C., Sosinski, E.E., Joner, F., Duarte, L.D.S. (2013). Functional redundancy and stability in plant communities. *J. Veg. Sci.* 24: 963-974.
- Saavedra et al. (2014) Functional importance of avian seed dispersers changes in response to human-induced forest edges in tropical seed-dispersal networks. *Oecologia* 176: 837-848.
- Sanders, D., Thébault, E., Kehoe, R., van Veen, F.J. (2018). Trophic redundancy reduces vulnerability to extinction cascades. *Proc. Natl. Acad. Sci. U.S.A.* 115: 2419-2424.
- Sethi, P. & Howe, H.F. (2009). Recruitment of hornbill-dispersed trees in hunted and logged forests of the Indian Eastern Himalaya. *Conserv. Biol.* 23: 710-718.
- Steel, Z.L., et al. (2021). Ecological resilience and vegetation transition in the face of two successive large wildfires. *J. Ecol.* 109: 3340-3355.
- Terborgh J., et al. (2008). Tree recruitment in an empty forest. *Ecology* 89: 1757-1768.
- Webster, C.L., et al. (2021) Population-specific resilience of *Halophila ovalis* seagrass habitat to unseasonal rainfall, an extreme climate event in estuaries. *J Ecol.* 109: 3260-3279.
- Weeks, T.L., et al. (2023) Climate-driven variation in dispersal ability predicts responses to forest fragmentation in birds. *Nat. Ecol. Evol.* 7: 1079-1091.

- Wilman, H., et al. (2014). EltonTraits 1.0: Species-level foraging attributes of the world's birds and mammals: Ecological Archives E095-178. *Ecology* 95: 2027-2027.
- Woodcock, B.A., et al. (2019) Meta-analysis reveals that pollinator functional diversity and abundance enhance crop pollination and yield. *Nat. Commun.* 10: 1481.
- Ye L, et al. (2019). Functional diversity promotes phytoplankton resource use efficiency. *J Ecol.* 107: 2353-2363.
- Yeeles, P., Lach, L., Hobbs, R.J. *et al.* (2025) Functional redundancy compensates for decline of dominant ant species. *Nat. Ecol. Evol.* **9**, 779-788.
- Zhu, L. et al. (2017). Trait choice profoundly affected the ecological conclusions drawn from functional diversity measures. *Sci. Rep.* 7: 3643.

We are very grateful to all the reviewers for their comments. These insights and critiques have been very helpful in refining the analyses and strengthening the paper in general.

Referees' comments:

Referee #1 (Remarks to the Author):

COMMENT: I thank the authors for their thoughtful and considered response and excellent paper – which presents strong results, including the finding that land use leads to species being spread thinly over trait space, making the assemblage functionally vulnerable to future change, and analytical methods which I expect to be influential.

The requested toning down of results relating to ecosystem functioning outcomes and foundational evidence for which traits respond to land use has been performed and presented, respectively, and I have no new major comments or suggestions.

RESPONSE: Thank you.

COMMENT: Just a comment on the response really but I do feel that it's hard to say whether the predicted trends would really affect functioning, but wording is generally toned down to reflect this and caveats are given. Ecosystem functions tend to be strongly controlled by both the diversity and abundance of interacting species (not just the traits of a single a single group), as well as physiological rates/activity and, and there are of course organisms other than birds that provide very large proportions of these functions too, so it cannot be said with confidence that all these changes will result in an observable change in whole system functioning. While FD etc often correlates with function and sig relations can be found its predictive capacity is often weak- other factors are also at play. As I say, though, this is just my reflection on the authors response, and I feel the paper does not overstretch the conclusions it draws from the results in its current form.

RESPONSE: Fair points. We appreciate that the reviewer was not necessarily suggesting we have overstretched the findings, but in view of this reasonable critique we have now edited the text in the Caveats and Limitations section of the main ms to emphasise that our analyses cannot capture the functioning of whole ecosystems. We now make the point that our data inevitably overlook aspects such as physiological rates/activity as well as the contributions of non-avian taxa. We think this helps to ground the discussion with a more realistic tone so we thank the reviewer for pointing us in this direction.

COMMENT: 52- rich>biodiverse (or just remove this word)

RESPONSE: edited.

COMMENT: 252- not sure what is meant by diversity of functions here when we are talking about individual functions- consider rephrasing

RESPONSE: To avoid any ambiguity, we have edited "diversity of functions' to "high FD" which for some readers may link more clearly with our definition of functional resistance.

COMMENT: 276-277 not sure if the 'demand' phrasing really works here as we have demand for services not functions and the link to this service aspect is not really covered at all in the paper – maybe simpler wording would be better here, e.g. "some of the FD lost under land use may be related to interacting species that are now absent and so no longer require their associated functions"

RESPONSE: We think that the concept of roles such as pollination and seed dispersal being ecosystem services (or at least ecological services/supporting services) is so widespread that we have retained the word “service”. The economic concept of service and associated demand is useful here because it helps to convey the idea of lost interaction partners. We believe this wording gets the message across better than the reviewer’s suggested text, which seems to us a bit opaque and also a mouthful.

COMMENT: 292- potentially leaving (the link to function cannot be assumed)

RESPONSE: Agreed. Changed to “potentially leaving”

COMMENT: 294 – FV (or write in full throughout, which might be better for general readers)

RESPONSE: Changed to “FV”. We share the reviewers preference for terms written out in full, but after trying various formats we think it works better to use FV throughout.

COMMENT: 1124- where the positive effects of land use on granivores explained in the main text? It’s an interesting result. I feel this is an important figure- would be great if there is space for it in the main paper

RESPONSE: We agree the variable responses of different guilds to land-use change are worth mentioning so we have highlighted the result here and elsewhere in the main text, along with an explanation (particularly in section titled “Functional stability varies across ecological processes”). We don’t think we should overstretch that finding in the main ms because the pattern in granivores is quite well established (e.g. Etard et al., 2022). We also discuss variation in responses to land-use change, and how this may be driven by diet-related mechanisms, in the SI subsection titled “Trophic groups”.

COMMENT: I did not review the methodology in detail this time round but find the analyses to be strong and well explained and justified. Maybe a little work is now needed to make the paper generally accessible, given complex responses to reviewer comments, but I will trust the editors to deal with this.

Thanks for the great work, which was a pleasure to read and review.

RESPONSE: We have tried to simplify the methods where possible to make the paper accessible to a general audience.

Referee #2 (Remarks to the Author):

COMMENT: I have enjoyed reading this new version of the manuscript. It is very well written, and all the concerns that I had in the previous version have either been corrected or explained in a convincing way. The responses from the authors are comprehensive and precise; they have modified the procedure for bandwidth estimation (my main concern), supplied robustness tests on species richness standardisation, added the requested extinction simulation, and clarified the definition of redundancy. All my earlier methodological questions have therefore been fully resolved, and all clarifications I suggested now appear either in the main text or the Supplementary Information.

The revised paper is clear, methodologically sound and timely. It delivers important evidence that land use change erodes functional redundancy and resistance in bird assemblages worldwide, with direct relevance for conservation planning. I have no further requests for analysis or wording changes. I would like to congratulate the authors for an excellent paper.

RESPONSE: We are grateful for all the feedback which helped us to improve the paper.

Referee #2 (Remarks on code availability):

COMMENT: I tried to open the link provided but it didnt work. I was satisfied with the script in the previous version, and explanations from authors in the response letter suggest that my previous comments on code have been taken into account, so I have no further requests.

RESPONSE: Unfortunately, we shared the github link associated with the previous submission (which has expired). We have now replaced this with an updated link to a Zenodo site (see last comments below).

Referee #4 (Remarks to the Author):

COMMENT: I did not review the previous version of this manuscript by Weeks and colleagues, but was invited to review the revised version by a Nature editor, with a request that I review closely the original critiques and subsequent author responses to Reviewer 3. I have thus read the entire revised manuscript, as well as the response document.

In general, I agree with previous reviewers and commend the authors on a broad, interesting manuscript. Moreover, I feel that the authors have sincerely worked quite extensively to revise this manuscript in alignment with all three reviewers' demands as well as to defend their original choices where appropriate. While I have not seen the originally submitted manuscript, I can see (through the response) the many ways that the manuscript has been improved. To that end, I delightedly experienced multiple times the feeling of "Huh, I wonder if they considered issue X?" only to discover, "Oh, indeed! There's an entire sensitivity analysis on it." So thanks to the authors for making my job more pleasant.

RESPONSE: Thank you!

COMMENT: As you might imagine from this preamble, my comments are mostly minimal. My overall assessment is that this is a thorough and well accomplished revision. They are, however, summarized below:

1. Although I have not seen the original, I can tell that Figure 1 has been improved in clarity; but does it give the wrong expectation? It seems to suggest that the study looks at how 'pristine' and modified communities respond to further degradation – i.e., the arrows from a/d to b/e/g imply processes that are being observed in this study. However, that is not empirically measured by the study; rather, this study predicts the disturbance process simply through extinction simulations. While the simulations are well considered (and more robust post-revision), simulations are not the same thing as empirical data. Since Figure 1 is largely a schematic for the study (hypotheses and predictions), is there a way to make it clear that the disturbance process is being simulated here and not observed directly through time series?

RESPONSE: We hadn't intended to give the impression we were observing the changes in community structure under different land-use change pathways, so we have made changes to the figure and caption. Specifically, we have changed the wording in the disturbance section of the figure to emphasise that we are dealing with conceptual changes in species assemblages undergoing potential future disturbance. We also make this very clear in the caption, using phrases like "Hypothetical future disturbances".

We think this figure is needed because it illustrates the core "conundrum" laid out in the introduction and provides the rationale underpinning our simulations. That is, human-modified assemblages may be either more or less resilient, but the outcomes are currently unknown.

In making the suggested changes to Figure 1, it struck us that some relevant concepts are not well conveyed. Rather than complicate the figure further, we have added a new Extended Data Figure 1 to clarify some terms and methods. This new figure helps to show that (1) there is turnover (new species arriving in human-modified assemblages, not just species lost) and (2) there can be changes in internal structure of functional trait space such as holes. Also, the new figure clarifies how we convert functional trait space into metrics of redundancy, vulnerability and resistance (=resilience).

COMMENT: 2. For Figure 2, the study size bubbles imply the smallest number of assemblages per site is 50, when undoubtedly it is much smaller. I would add a size bubble

that corresponds to the smallest unit on the map. It is typical for figure legends to show the full range of an important measure.

RESPONSE: We have now extended the figure legend to visualise the size of studies containing 5 sites. We could not extend the lower limit of the legend any further without affecting the aesthetics of the figure, particularly as it created odd spacing within the legend. We think that adding the smaller samples <5 sites has helped to convey the range of survey depth in our sample.

COMMENT: 3. I appreciated Reviewer 3's concern about the role of imperfect detection in potentially biasing results and the dataset. I read the manuscript first, before the review, and when I got to the "Caveats and limitation" section I read it with careful interest, but was generally left unfulfilled. As written, the section on caveats tonally feels like a perfunctory section added to appease reviewers – it doesn't seem like the authors believe that this work is deserving of any 'caveats or limitations' at all! For example, most of the paragraph is taken up not with exploring potential limitations, but defending against them. Not only is this a waste of space, but it can be read as insulting to readers. For example, it is truly a limitation that the inferential strategy is a "space-for-time" comparison (pristine vs. modified) combined with a pure simulation of temporal processes (i.e., further disturbance); there is an extensive literature in ecology showcasing the limitations of space-for-time approaches and equally extensive examples showing how simulations fail to capture observed phenomena. Yet, the authors seem to dismiss these real limitations by touting the analysis as "probably the best space-for-time comparison available", which honestly made me laugh (not to be too mean, but it sounds like the things certain world leaders say). Similarly, the dismissal of potential concerns about detectability biases (which have been shown repeatedly to bias all sorts of diversity metrics, including functional diversity metrics) by asserting that "bird species are easily identifiable" and can be detected by "acoustic signals" almost absurdly misses the point that nearly all methodological studies documenting the extent of bias from imperfect detection have been based on field surveys of these same "easily identifiable" birds. Ultimately, I agree with the authors (on detectability) that the direction of bias would be such that they are likely underestimating the true effect. However, I do feel that this 'caveats and limitations' section would benefit from an increased dose of honest self-critique.

RESPONSE: We thank the reviewer for flagging up the limitations to our limitations! We have largely rewritten this section to address the comments. We think it is much better now, but unavoidably longer.

Additional Small Line Edits:

COMMENT: 63. I get that there is too much here to be summarized, necessitating supplementary information. However, it seems a shame that none of the researchers who have done this valuable pioneering work will receive citation credit (since supplements aren't indexed). Perhaps cite 1-2 of the most important, most relevant studies, and then refer to the supplement?

RESPONSE: We agree with the reviewer and we would very much like to cite more of the underlying biodiversity survey research. We have successfully argued for this before, eg Benitez-Lopez et al. 2021 <https://www.nature.com/articles/s41559-021-01426-y> which ended up with 640 references in the main ms! This solution seems reasonable in an online journal like NEE but it is not practical for a hardcopy journal like Nature. Unfortunately, there is no room to add citations to the underlying research, and including 1-2 key papers seems a little unfair as it raises difficult choices of who to acknowledge, which could be annoying to those "missing the cut".

COMMENT: 131. With such a high p-value, can anything be concluded at all about FD in

mature secondary vegetation? It seems inappropriate to conclude anything about this result.

RESPONSE: We have re-written the text to clarify that this result was non-significant.

COMMENT: Code availability: I was given a GitHub repository URL. However, the URL did not work for me so i was not able to review the code.

RESPONSE: We apologise for the old link. We have now published all data and code on Zenodo. The link to these materials is [10.5281/zenodo.17161425](https://doi.org/10.5281/zenodo.17161425). We have now added this link to the main ms (line 1269).

Referee #4 (Remarks on code availability):

COMMENT: The code, as cited above, was not available to me. The URL gave me an error. Moreover, GitHub is not a suitable archive for code, as it can be changed post-publication. Code for published papers should have permanent archives.

RESPONSE: See previous response.